# Proteogenetic drug response profiling elucidates targetable vulnerabilities of myelofibrosis

Mattheus H. E. Wildschut [1,2,3], Julien Mena [1], Cyril Dördelmann [4], Marc van Oostrum [2], Benjamin D. Hale[1], Jens Settelmeier [2,5], Yasmin Festl[1], Veronika Lysenko [3], Patrick M. Schürch [3], Alexander Ring[3], Yannik Severin[1], Michael S. Bader[6], Patrick G. A. Pedrioli[2,5,7], Sandra Goetze [2,5,7], Audrey van Drogen[2,5,7], Stefan Balabanov[3], Radek C. Skoda[6], Massimo Lopes [4], Bernd Wollscheid [2,5,8] ✉, Alexandre P. A. Theocharides [3,8] ✉ & Berend Snijder [1,5,8] ✉

Myelofibrosis is a hematopoietic stem cell disorder belonging to the myeloproliferative neoplasms. Myelofibrosis patients frequently carry driver mutations in either *JAK2* or Calreticulin (*CALR*) and have limited therapeutic options. Here, we integrate ex vivo drug response and proteotype analyses across myelofibrosis patient cohorts to discover targetable vulnerabilities and associated therapeutic strategies. Drug sensitivities of mutated and progenitor cells were measured in patient blood using high-content imaging and single-cell deep learning-based analyses. Integration with matched molecular profiling revealed three targetable vulnerabilities. First, *CALR* mutations drive BET and HDAC inhibitor sensitivity, particularly in the absence of high Ras pathway protein levels. Second, an MCM complex-high proliferative signature corresponds to advanced disease and sensitivity to drugs targeting pro-survival signaling and DNA replication. Third, homozygous *CALR* mutations result in high endoplasmic reticulum (ER) stress, responding to ER stressors and unfolded protein response inhibition. Overall, our integrated analyses provide a molecularly motivated roadmap for individualized myelofibrosis patient treatment.

Myelofibrosis (MF) is a chronic blood cancer belonging to the family of myeloproliferative neoplasms (MPN). Compared to the other MPN, polycythemia vera (PV) and essential thrombocythemia (ET), MF is characterized by a more aggressive disease phenotype and a worse prognosis due to increased risk of progression and transformation to acute myeloid leukemia (AML)[1]. The disease occurs in patients either as primary MF (PMF) or as a natural evolution from pre-existing PV or ET disease (PPV-MF and PET-MF, respectively)[2]. At the molecular level, MF is driven by mutations occurring in the hematopoietic stem and progenitor cell (HSPC) compartment that deregulate downstream

[1]Institute of Molecular Systems Biology, Department of Biology, ETH Zurich, Zurich, Switzerland. [2]Institute of Translational Medicine, Department of Health Sciences and Technology, ETH Zurich, Zurich, Switzerland. [3]Department of Medical Oncology and Hematology, Division of Hematology, University Hospital Zurich, Zurich, Switzerland. [4]Institute of Molecular Cancer Research, University of Zurich, Zurich, Switzerland. [5]Swiss Institute of Bioinformatics, Lausanne, Switzerland. [6]Department of Biomedicine, Experimental Hematology, University Hospital Basel and University of Basel, Basel, Switzerland. [7]ETH PHRT Swiss Multi-Omics Center (SMOC), Zurich, Switzerland. [8]These authors jointly supervised this work: Bernd Wollscheid, Alexandre P. A. Theocharides, Berend Snijder. ✉e-mail: bernd.wollscheid@hest.ethz.ch; alexandre.theocharides@usz.ch; bsnijder@ethz.ch

hematopoiesis[3]. Over 90% of MF patients carry defined disease-driving mutations in either one of two genes: *JAK2*, a tyrosine kinase, and Calreticulin (*CALR*), an endoplasmic reticulum (ER)-residing chaperone of glycoproteins[4, 5]. Both *JAK2* and *CALR* driver mutations cause cytokine-independent overactivation of the JAK/STAT pathway[6]. Secondary mutations associated with disease progression can occur in epigenetic regulators, the spliceosome complex, and a variety of tumor suppressors[7]. Despite the genetic homogeneity driving MF, the clinical presentation is heterogeneous.

JAK inhibitors (JAKi) that target the common JAK/STAT pathway hyperactivation have revolutionized the treatment landscape for MF patients by inducing symptom relief and improving quality of life[8]. However, with the exception of hematopoietic stem cell transplantations, currently approved MF therapies are not curative[9] and do not eliminate the malignant MF clone[10]. Furthermore, therapies including JAKi are limited by pre-existing or acquired resistance[11,12], and subsequent treatment alternatives are at present limited[13]. Thus, the complex disease biology of MF drives significant patient heterogeneity, requiring a better understanding of molecularly targeted and individualized therapeutic strategies.

Despite cell lines being essential in preclinical research, their drug response and molecular profiles commonly do not recapitulate patient heterogeneity and clinical response. For example, while JAKi efficiently and specifically targets cell lines harboring MF driver mutations[14], this targeted effect on the malignant MF clone is absent in both mouse models[15,16] and patients[14,16]. To ensure optimal clinical translatability of our findings, we set out to perform ex vivo drug response profiling and mass spectrometry-based proteotyping directly on MF patient blood cells. Utilizing pharmacoscopy, previously shown to be clinically predictive across a variety of hematological malignancies[17,18], we developed a single-cell multiplexed and deep learning-based MF drug response profiling platform that captures patient heterogeneity and quantifies drug efficacy on the malignant MF clone. Cellular proteotyping enabled us to capture molecular interpatient heterogeneity using isolated blood cell populations across large clinical MF, MPN, and healthy donor (HD) cohorts. This integrative pan-cohort analysis provides direct insight into the molecular drivers of clinical heterogeneity, linking targeted therapeutic sensitivities to molecularly stratified MF patient profiles.

In this work, we perform integrated clinical image-based drug response and proteotyping profiling of MF patients to reveal three distinct targetable vulnerabilities in defined patient subgroups. First, MF patients with *CALR* driver mutation and low Ras signaling pathway expression responded well ex vivo to BET and HDAC inhibitors. Second, patients characterized by increased DNA replication and cell proliferation displayed increased sensitivity to idasanutlin, vosaroxin, and navitoclax. Third, homozygous *CALR*-mutated patients were characterized by particularly high levels of ER stress, resulting in sensitivity to ER stressors and UPR inhibitors. Overall, our integrated functional drug response landscape yields molecular insights into MF pathogenesis and concurrent therapeutic strategies, providing a roadmap for improved personalized treatment of MF patients.

## Results
### Image-based single-cell oncogenic readouts in MF samples
HSPCs in the peripheral blood of MF patients contain the disease-driving cells[15,19], are characterized by clonal dominance[20,21], and display molecular features that might allow their detection by immunofluorescence. We therefore set out to establish immunofluorescence readouts reflecting the oncogenic nature of each cell in both CALR and JAK2 mutant-driven MF compatible with pharmacoscopy-based drug response analysis.

For *CALR*-mutated MF, the *CALR* driver mutations result in the formation of a common neoantigen[4,5]. Making use of a recently reported antibody that specifically binds this mutant CALR protein-

specific sequence (CALRm)[22], we established high-throughput immunofluorescence of mutant CALR expression in MF cells, compatible with automated microscopy and single-cell image analysis. First, we validated this image-based readout using cell lines, showing that the CALRm antibody stained CMK11-5 cells CRISPRed to express mutant CALR[23] as well as the *CALR* mutant patient-derived MARIMO cell line[24] (Supplementary Fig. 1a). Confirming antibody specificity and sensitivity of the approach, CALRm staining was neither observed in the same CMK11-5 background expressing wild-type CALR, nor in a CRISPRed CALR knockout CMK11-5 line by both imaging and flow cytometry analysis (Fig. 1A, Supplementary Fig. 1b–d). Using an image-based pooled cell line screening approach, we next confirmed that the *CALR* mutation specifically resulted in sensitivity to JAK/STAT signaling inhibition by ruxolitinib and fedratinib (Supplementary Fig. 1e). Lastly, we used the image-based CALRm readout to analyze peripheral blood mononuclear cell (PBMC) samples isolated from MF patients diagnosed with either *CALR* or *JAK2* driver mutations (MF CALR and MF JAK2, respectively). CALRm-positive cells were uniquely present in MF CALR PBMCs, and their proportion correlated quantitatively to mutation burden as determined by the patient variant allele frequency (VAF) (Fig. 1B). Strikingly, CALRm levels displayed considerable single-cell heterogeneity in MF PBMCs, even within PBMCs of patients that genetically carry a 100% VAF (Fig. 1B).

For *JAK2*-mutated MF, we established a pSTAT5 antibody-based readout that enables image-based quantification of JAK/STAT pathway activation at the single-cell level in patient samples. As expected, pSTAT5 levels were significantly elevated in MF driven by either *JAK2* or *CALR* mutations compared to HD blood (Supplementary Fig. 1f). Confirming the approach, pSTAT5 levels were efficiently and dose-dependently reduced by a variety of JAK inhibitors in a patient sample (Supplementary Fig. 1g), although considerable differences in efficacy were observed between different JAK inhibitors possibly relating to their unique pharmacological properties[23]. We further confirmed that mutant CALR expression at the single-cell level correlated with hyperactivity of the JAK/STAT pathway within MF CALR PBMCs (Supplementary Fig. 1h). Despite the correlation of CALRm and pSTAT5 levels at the single-cell level, their subcellular localizations differed (Supplementary Fig. 1i). Whereas CALRm was in close proximity to the ER and *cis*-Golgi system, confirming previously reported tagged CALR mutant construct localization[25], pSTAT5 had a general cytoplasmic localization as similarly found for myeloid leukemias[26] and *JAK2*-mutated cell lines[27]. Thus, the pSTAT5-based readout is a suitable proxy for the presence of oncogenic driver mutations of MF, amenable to quantification by automated microscopy.

### Pharmacoscopy elucidates drugs specifically depleting the malignant MF clone present in patient blood ex vivo
Utilizing the established oncogene readouts, we set up a pharmacoscopy-based drug response screen to measure oncogene and HSPC depletion upon ex vivo exposure of MF patient PBMCs to a library of 79 clinical-grade drugs (Fig. 2A). We screened 43 MF patients of which 16 carried *CALR* and 27 carried *JAK2* driver mutations with diverse secondary mutation profiles (MF PBMC cohort; Fig. 2B and Supplementary Data 1). In short, after 20 hours of drug treatment, cells were fixed and stained with a multiplexed antibody panel and imaged by high-content automated microscopy. Small compound drugs were tested at 1 and 10 μM, while biologics were tested at 0.1, 1, and 10 μg/ml. Single-cell image analysis and deep learning-based classification (by convolutional neural network; CNN; Supplementary Fig. 2a) quantified the number of viable HSPCs (CD34+), T-cells (CD3+), monocytes (CD14+), and other cells (marker negatives) in each condition, outperforming classification based on average features per cell (Supplementary Fig. 2b). Cell viability was scored by CNN based on cellular and nuclear morphology (Supplementary Fig. 2c), and conventional image analysis was used to

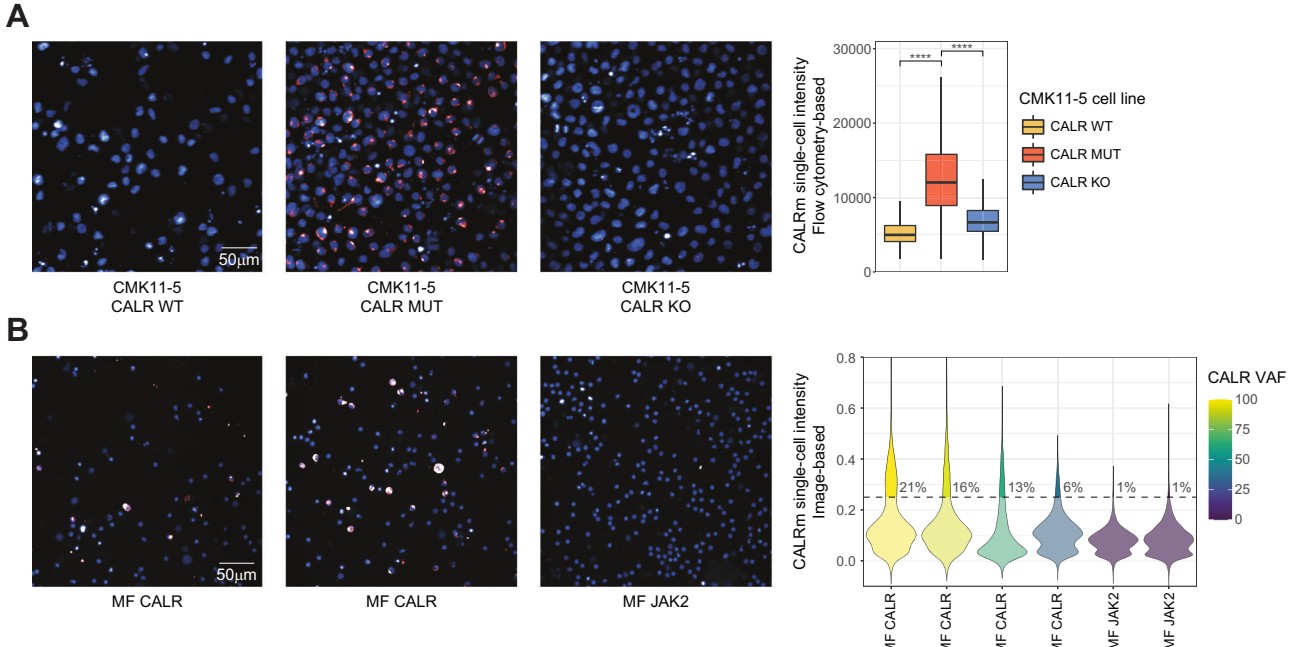

**Fig. 1 | Single-cell analysis of mutant CALR protein expression. A** Staining of a cell line panel with the CALR mutant-specific antibody CALRm. Representative images of the included cell lines are shown (left panels; blue: DAPI; red: CALRm). Boxplots (right) represent single-cell CALRm intensities for the panel as quantified by intracellular flow cytometry analysis (right panel). *p*-Values indicate Student's *t*-test significance: **** = *p* < 0.0001. Exact *p*-values are reported in Source Data. Box plots indicate the median (horizontal line) and 25% and 75% ranges (box), and whiskers indicate the 1.5× interquartile range above or below the box. **B** Single-cell immunofluorescence imaging of MF PBMCs stained with CALRm. Representative images of different MF PBMC samples are shown (left panels; blue: DAPI; red: CALRm). Single-cell mean CALRm intensities of four replicate wells with 25 images per well are shown as a violin plot with percentages of thresholded CALRm-positive cells annotated as text (right panel). Representative results of two independent repeats are shown. Violin colors indicate the *CALR* VAF of the corresponding MF patient. See also Supplementary Fig. 1.

quantify the CALRm and pSTAT5 oncogenic levels for MF CALR and MF JAK2, respectively, for each CNN-classified cell, for a total of 34,750,749 cells.

We first analyzed the cellular composition in non-drug-treated conditions, revealing vast heterogeneity both in cell population sizes (Fig. 2C) and in cell phenotypes (Fig. 2D) across the 43 patient samples. Image-based HSPC fractions showed highly significant concordance with the clinically determined peripheral blood (PB) blast percentages (*p* < 1.06⁻⁶; Supplementary Fig. 2d, e). Furthermore, elevated monocyte fractions were observed for MF CALR patients, consistent with their elevated clinically determined PB monocyte counts (*p* < 3.01⁻⁴; Supplementary Fig. 2d, f). Drug response results for 3 of the 43 patients were excluded from further analysis due to HSPC fractions below 0.3% (Fig. 2B). Quantification of oncogenic CALRm and pSTAT5 readouts across cell types and patient samples demonstrated that levels are highest in monocytes and HSPCs and, particularly for CALRm, lowest in T-cells (Fig. 2D, E). Within the MF CALR subcohort, the highest CALRm levels were seen across cell types for patients carrying predominant homozygous mutations, which we define as *CALR* VAF >= 75% and refer to as 'homozygous CALR' (Fig. 2F).

Next, we quantified and analyzed the relative changes of cell types and oncogenic readouts upon ex vivo drug treatment across the cohort. We prioritized drugs and drug classes with significant on-target effects depleting HSPCs and/or the respective oncogenic CALRm and pSTAT5 readouts across MF CALR and MF JAK2 patients. Despite extensive interpatient heterogeneity in drug responses (Supplementary Fig. 3a, Supplementary Data 2), we observed several recurring drug sensitivities across the cohort, including the BCL-2 family protein inhibitor navitoclax and BET inhibitor pelabresib, both drugs of current clinical interest for MF (Fig. 3A)[27]. Globally, we found that drugs belonging to the same drug class elicited similar responses, reflecting their class-specific mechanism of action (Supplementary Fig. 3b).

Integration of the drug response profiles with patient-matched clinical and cellular proteotype information (Fig. 2A) allowed us to infer the main determinants of the observed drug response variability in MF. We first globally assessed which clinical factors (Supplementary Data 1) explain the observed drug response individuality, revealing MF driver mutation status and PB blast counts as the strongest covariates (Fig. 3B). Unsupervised clustering of the drug response profiles indeed indicated significant similarity in the mutation-related vulnerabilities (Supplementary Fig. 3c). As a striking example, both tested BCL2 inhibitors elicited considerably stronger responses in both the HSPC- and oncogenic readout for *JAK2* mutant-driven MF (Supplementary Fig. 3d). Second, we analyzed the associations between drug sensitivity and protein expression measured by sample-matched proteomic analysis of isolated CD34⁺ HSPCs across the cohort (40/43 patients; Fig. 2B, Supplementary Fig. 4a–d, Supplementary Data 3). To get a top-level view, we first calculated the number of times we observed a significant pathway-level association with HSPC drug sensitivity or resistance across all tested drugs. Resistance-associated pathways showed a higher recurrence across drugs, indicating the presence of shared resistance mechanisms, whereas sensitivity-associated pathways were more drug-specific (Fig. 3C). Top pathways enriched in drug sensitivity-associated proteins included ribosome, spliceosome, cell cycle, and DNA repair pathways. Inversely, the MAPK-Ras pathway levels associated with HSPC resistance to the majority of tested drugs, in line with previous findings[28].

**CALR driver mutations sensitize MF HSPCs to BET inhibition**

Exploring mutation-associated drug sensitivities deeper, we next focused on the observation that HSPCs of MF CALR patient samples were particularly sensitive to the tested BET and HDAC

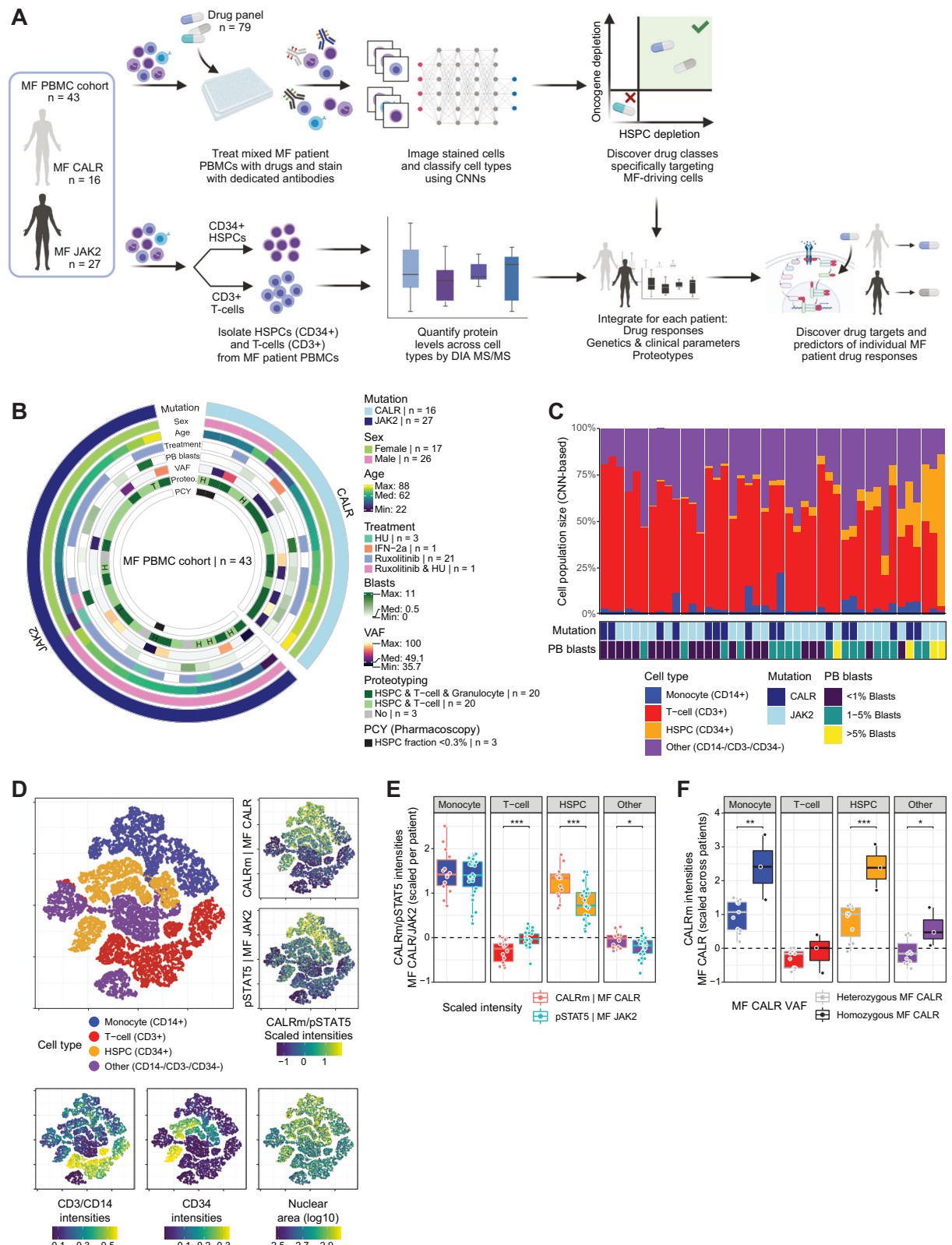

inhibitors (BETi and HDACi; *n* = 7 drugs; Fig. 4A, Supplementary Fig. 3d). Importantly, the BETi association held when taking potential confounders into account (Fig. 4B). We validated that the presence of the *CALR* mutation results in BETi sensitivity utilizing the CMK11-5 cell line panel. In mixed cultures of *CALR* wild-type, mutant, and knockout cells, *CALR*-mutated cells were specifically depleted by BETi treatment (Fig. 4C).

As an additional stratifier of the observed ex vivo BETi and HDACi drug response heterogeneity across driver mutations, we assessed the HSPC proteomic signatures related to drug sensitivity and resistance. This revealed that BETi and HDACi-sensitive HSPCs were characterized by low expression levels of Ras signaling pathway members (Fig. 4D), including CDC42 and MAP2K1/MEK (Fig. 4E). These results are in line with previous reports on Ras pathway-mediated resistance to BETi in

**Fig. 2 | Clinical and molecular determinants of cellular heterogeneity in MF.**
**A** Outline of the integrative pharmacoscopy and clinical proteotyping workflow of MF patients as performed in this study. **B** Circos plot of clinical annotations of the MF PBMC cohort. Legend indicates color codes for the included variables. For discrete variables, patient numbers are included, and for continuous variables, the range and median are represented. Proteotyping outliers are annotated by H and T for HSPC and T-cell proteotypes, respectively. See also Supplementary Data 1. **C** Cell population sizes as defined by the CNN-based cell classifier across the DMSO-treated conditions of the MF PBMC cohort. Annotations indicate the mutation status and PB blast counts of the respective patients. **D** t-SNE embedding of the CNN class probabilities for DMSO-treated cells across the cohort of 43 MF patients ($n = 41,286$ patient cells; up to 250 cells randomly selected from HSPC, T-cell, monocyte, and other cells for each MF patient). Top left panel: CNN-based cell types. Top right panels: single-cell CALRm (upper) and pSTAT5 (lower)

intensities, scaled by $z$-score normalization. Bottom panels: single-cell CD3/CD14 and CD34 intensities and nuclear area. **E** CALRm/pSTAT5 intensities per cell type and driver mutation. Intensities are scaled by $z$-score normalization per patient. Boxplots indicate the range of scaled intensities across MF CALR and JAK2 patients, and dots represent individual patient values (MF CALR, $n = 14$; MF JAK2, $n = 26$). Fill colors indicate cell types as in Fig. 1C. **F** CALRm intensities per cell type and homozygous (VAF > = 75%) or heterozygous MF CALR (VAF < 75%). Intensities are scaled by $z$-score normalization across all MF CALR cells (MF CALR heterozygous, $n = 11$; MF CALR homozygous, $n = 3$). Boxplots indicate the range of scaled intensities, and dots represent individual patient values. Fill colors indicate cell types as in Fig. 1C. Asterisks indicate non-adjusted two-sided Student's $t$-test significance: **** = $p < 0.0001$, *** = $p < 0.001$, ** = $p < 0.01$, * = $p < 0.05$; exact $p$-values are reported in Source Data. Boxplots as in Fig. 1A. See also Supplementary Fig. 2.

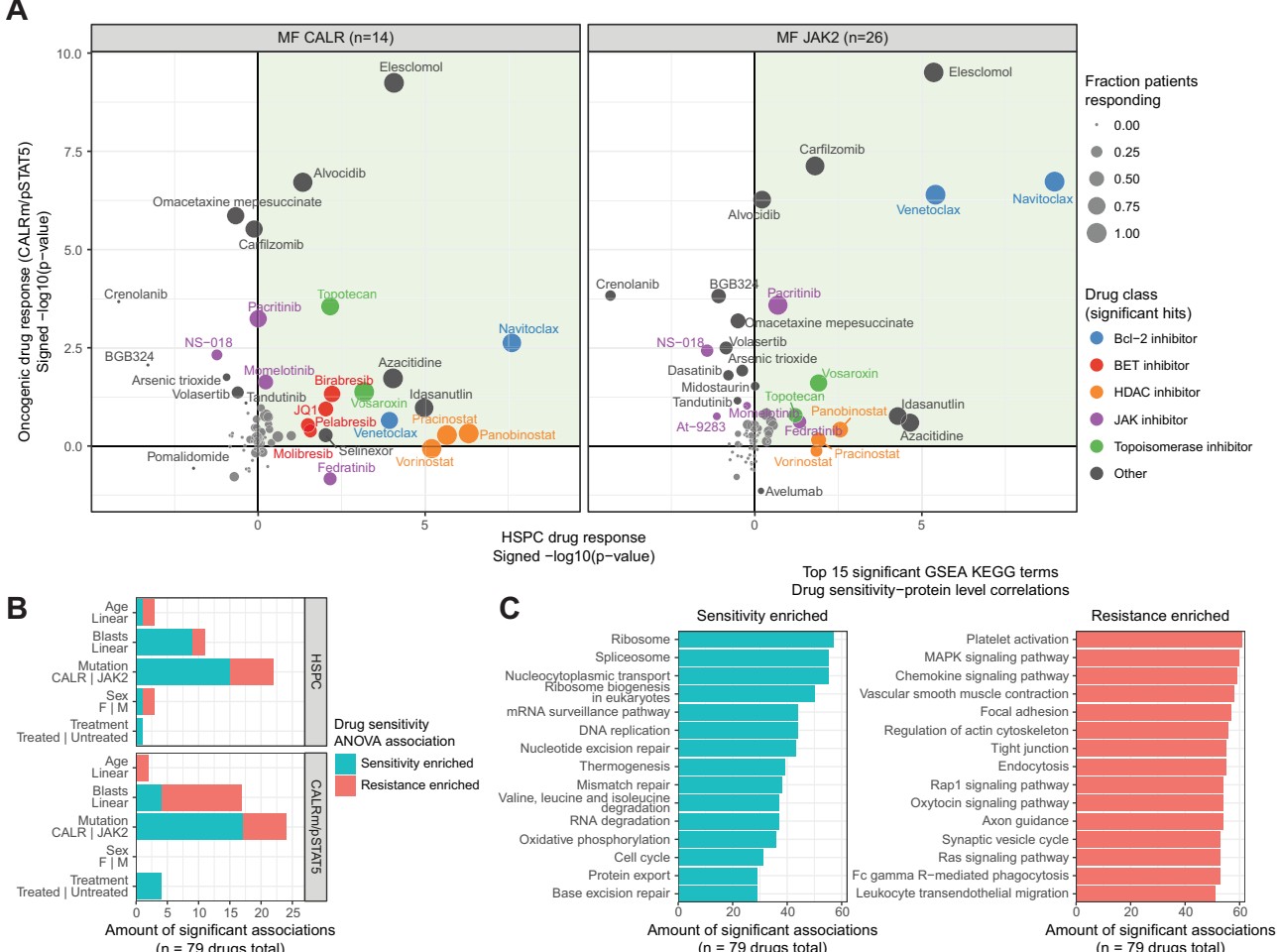

**Fig. 3 | Pharmacoscopy elucidates drugs specifically depleting the malignant MF clone.** **A** Summary of pharmacoscopy-based drug responses, split by MF CALR (left; $n = 14$ patients) and JAK2 (right; $n = 26$ patients). The $x$-axis shows the signed significance of HSPC drug responses per patient, averaged across the genetically stratified subcohorts. The $y$-axis shows the signed significance of oncogenic CALRm and pSTAT5 drug responses per patient for MF CALR and JAK2, respectively, averaged across the genetically stratified subcohorts. $p$-values indicate Student's $t$-tests of summarized drug responses across replicates and concentrations compared to the respective control condition (DMSO, PBS, or isotype control). Values indicate $-\log10$ $p$-values, signed positively or negatively for on- or off-target response, respectively, averaged across the CALR MF and JAK2 MF cohorts. The dot color indicates drug class; the dot size is the fraction of patients that have a

significant on-target effect averaged across the two drug scores. **B** Association of MF patient drug responses with clinical factors. For every drug and for both drug response readouts, an ANOVA was performed for selected factors. Significant drug response-factor associations (ANOVA $p < 0.05$) were counted ($x$-axis) per factor ($y$-axis) and are displayed as either associating with sensitivity (blue) or resistance (red). **C** Protein pathway-level associations to MF patient drug responses. For every drug, the HSPC drug responses were correlated (Spearman) to HSPC protein levels across patients. GSEA was performed on the ranked correlations, calculating the enrichment of positive or negative protein–drug response correlations. Significant GSEA enrichment ($p < 0.05$) was counted ($x$-axis) per KEGG pathway term ($y$-axis), and the top 15 positives (blue; left panel) and negative (red; right panel) enriched terms are displayed. See also Supplementary Figs. 3 and 4.

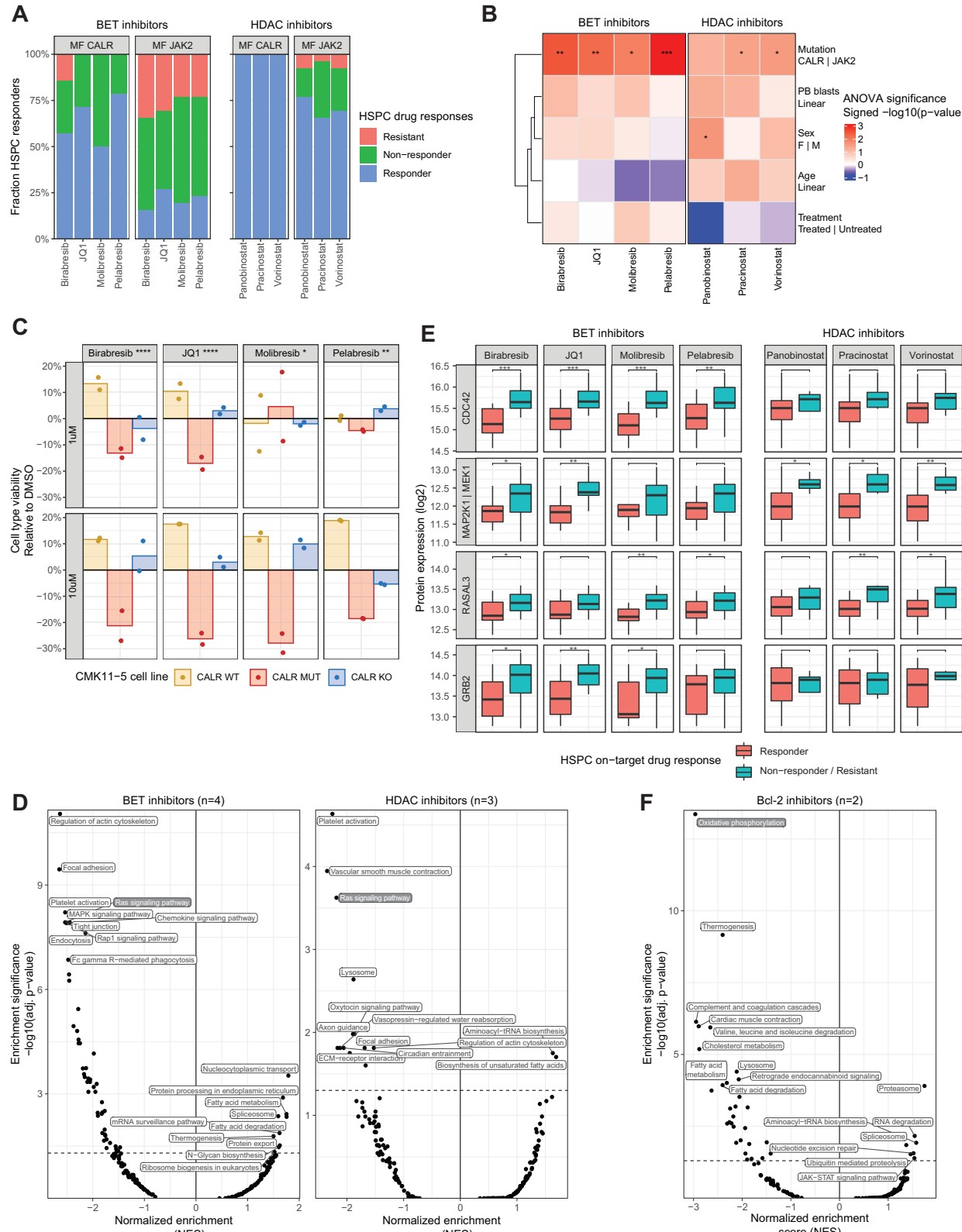

other cancer types[29]. We further repeated these analyses for BCL-2 inhibitors (BCL-2i) sensitivity, another drug class of clinical interest for MF, which revealed a strong proteotype signature correlating to drug sensitivity (Fig. 4F): HSPCs with high levels of oxidative phosphorylation proteins were strongly resistant to BCL-2 inhibition, corresponding to the described role of this metabolic state in BCL-2i drug responses[30].

**Large-scale clinical proteotyping elucidates the MF-specificity of targetable protein alterations**

To interrogate if targetable protein alterations are specific to MF, we performed cellular proteotype analysis of samples from an orthogonal clinical cohort (n = 113) comprising granulocytes from MF and ET patients and age- and gender-matched HDs (Fig. 5A, B). Given the potential contamination of erythrocytes in density-isolated

**Fig. 4 | *CALR* driver mutations sensitize MF HSPCs to BET inhibition. A** BET and HDAC inhibitor HSPC responses across the cohort split by driver mutation (MF CALR, *n* = 14 patients; MF JAK2, *n* = 26 patients). Responder = significant (*p* < 0.05) on-target response, non-responder = non-significant on-target, resistant = off-target response. **B** Signed ANOVA significance of BET and HDAC inhibitor HSPC drug responses to clinical parameters. **C** BET inhibitor drug responses of a co-cultured *CALR* CRISPRed cell line panel analyzed by pharmacoscopy. A representative experiment of two is shown, with two technical replicates across two conditions with 25 images per well. The *y*-axis depicts the change in relative fraction per cell line in the drug-treated condition compared to DMSO. *p*-Values indicate significance as determined by the Student's *t*-test of CALR MUT depletion across the different replicates and concentrations compared to DMSO-treated cells. **D** Analysis of protein processes significantly up- or downregulated in BET (left) and

HDAC inhibitor (right) responding patients. *t*-Tests were performed for each protein, comparing protein levels in HSPCs of responding MF patients compared to the non-responding and resistant patients. GSEA was performed on proteins ranked according to signed *t*-test significance. GSEA normalized enrichment scores (NES; *x*-axis) and enrichment significance (*y*-axis) are shown. **E** HSPC protein levels of selected Ras signaling pathway members. Patients are grouped according to HSPC response to BET and HDAC inhibitors (Responder, *n* = 28; resistant/non-responder, *n* = 22). Protein expression (*y*-axis) of proteins shown is significantly different for responders for at least 3 BET/HDAC inhibitors. *p*-Values indicate the Student's *t*-test significance. Boxplots as in Fig. 1a. **F** Analysis of protein process–drug response associations of Bcl-2 inhibitors as in (**D**). Asterisks indicate non-adjusted two-sided significance: **** = *p* < 0.0001, *** = *p* < 0.001, ** = *p* < 0.01, * = *p* < 0.05; exact *p*-values are reported in Source Data. See also Supplementary Figs. 3 and 4.

granulocyte fractions, we performed linear regression to remove the influence of this possible contaminant on granulocyte protein quantification (Supplementary Fig. 4e–h). Granulocytes were analyzed for their high mutational penetrance in MPNs, high abundance in both healthy and diseased blood, and routine use in clinical diagnostics.

Comparing the similarity in granulocyte proteotypes of HDs and MPN patients revealed that MF, the most aggressive MPN, is the most distinct from HD (Fig. 5C, D, Supplementary Data 3). Proteins underlying these differences included known oncogenic signaling proteins as well as members of the spliceosome and ribosome complexes (Supplementary Fig. 5a), which we identified as recurring drug sensitivity-associated pathways in HSPCs (Fig. 3C). We employed a machine learning approach to elucidate the minimal set of proteins that define the distinct cellular proteotypes of disease and mutation (Fig. 5E). This strategy identified 15 key proteins that lead to excellent separation of the five subcategories (Fig. 5F, G, Supplementary Fig. 5b).

The 15-protein signature included both known and novel protein alterations, which included MPN general, ET/MF-specific, or mutation-specific protein expression patterns (Fig. 6A, Supplementary Fig. 5c). Elevated CALR protein levels in MF JAK2 and reduced levels in MF CALR are in line with previous reports[31, 32]. Similar trends in ALPL levels can relate to differences between the two driver mutations in the clinical neutrophil/leukocyte alkaline phosphatase (NAP/LAP) score[33,34]. Protein alterations similarly described in other malignancies include reduced levels of CD59 in AML[35] and cancer-promoting loss of fumarate hydratase (FH) in a variety of cancers[36,37]. Of note, the 15-protein signature included MCM4, MCM7, and RFC2, all involved in DNA replication and specifically upregulated in MF (Fig. 6A, Supplementary Fig. 5c).

We complemented this reductionist approach with global disease- and driver mutation-specific protein and pathway level associations (Fig. 6B). Network-based visualization indicated pathway and protein levels that were associated with patient characteristics, enabling a systems biology view on the protein-level rewiring underlying MPNs. For example, the ribosome was found as a central node positively associated with MPN, MF, higher *CALR* VAF, and higher *JAK2* VAF. In contrast, ALPL protein levels are associated specifically with *JAK2* mutation status and *JAK2* VAF. The network connectivity further highlighted the significant association of MF with elevated DNA replication, which we found as a recurring drug sensitivity-associated pathway in MF HSPCs (Fig. 3C). Lastly, the network analysis expanded on the 15-protein signature with additional protein complex members, including the physically interacting replication factors RFC2 and RFC3, as well as MCM3, MCM4, and MCM7, members of the MCM helicase complex involved in unwinding DNA for DNA replication (Fig. 6B).

### Proliferative MCM-high MF responds to the targeting of survival signaling and DNA replication

We explored the MCM proteins as part of a signature of elevated DNA replication in MF patients and its association with drug sensitivity.

Across the HD/MPN granulocyte cohort, the levels of signature proteins MCM4 and MCM7 correlated strongly and increased in MPN, most significantly for MF patients (Fig. 7A). However, considerable heterogeneity was present among MF patients, in which a subset of MF patients could be found with markedly elevated MCM4 and MCM7 abundance (Fig. 7A, Supplementary Fig. 5c).

MCM levels per patient showed a significant positive association with PB blast percentages, consistently for all three cell types (Fig. 7B, C, Supplementary Fig. 6a). Furthermore, prototype-wide, MCM4 and MCM7 levels correlated to a 50-protein signature of physically interacting proteins and protein complexes (Fig. 7D, Supplementary Fig. 6b). This signature included the MCM2-7 helicase complex and other proteins involved in DNA replication, cell cycle regulation, and DNA repair (Fig. 7E). Analysis of an independent MPN cohort confirmed elevation of MCM complex levels in MF patients also on a transcriptional level[38] (Supplementary Fig. 6c). Next, making use of the subcohort of 20 MF patients present in both the MPN granulocyte and MF PBMC cohort (Fig. 2B), we evaluated co-regulation of MCM abundance across cell types. This analysis revealed that MCM4 and MCM7 levels were strongly correlated between all three cell types, allowing a robust categorization of MF patients into MCM-low, -medium, and -high classes (Fig. 7F).

Given the function of the MCM complex in DNA replication, we measured the functional impact of high MCM protein abundance on cell proliferation and DNA damage using canonical immunofluorescence readouts (Ki-67 and γH2Ax, respectively) on HD and MCM-category stratified MF samples (Supplementary Fig. 6d). Across these categories, we found a significant correlation between MCM protein levels and cell proliferation, yet this higher proliferation rate did not associate with increased DNA damage (Fig. 7G). As proliferative HSPCs drive MF, we next measured the impact of different MCM protein levels on the rate of DNA replication fork progression of isolated HSPCs in vitro. While ET and MCM-low MF consistently displayed increased fork speed compared to HD, MCM-medium and -high MF HSPCs had fork speeds comparable to that observed in HDs (Fig. 7H). The high proliferation rates of MCM-high MF, therefore, surprisingly coincided with normalized DNA replication fork speeds.

We next assessed if these elevated MCM levels led to unique therapeutic opportunities by correlating expression levels of the 50-protein replicative signature with ex vivo drug responses across the MF PBMC cohort (Fig. 7I). HSPCs with a high replicative signature were particularly sensitive to the p53-stabilizer idasanutlin, the BCL-2/BCL-XL inhibitor navitoclax, and the topoisomerase 2 inhibitor vosaroxin (Fig. 7J). We validated the effect of vosaroxin, a potential therapeutic option for MF, on isolated HSPCs from patients found to be sensitive by pharmacoscopy. Upon vosaroxin treatment, depletion of viable HSPCs coincided with simultaneous induction of apoptosis and reduction of proliferation (Fig. 7K, Supplementary Fig. 6e). Thus, specific adaptations of the DNA replication process and pro-survival pathways appear to be of particular importance for

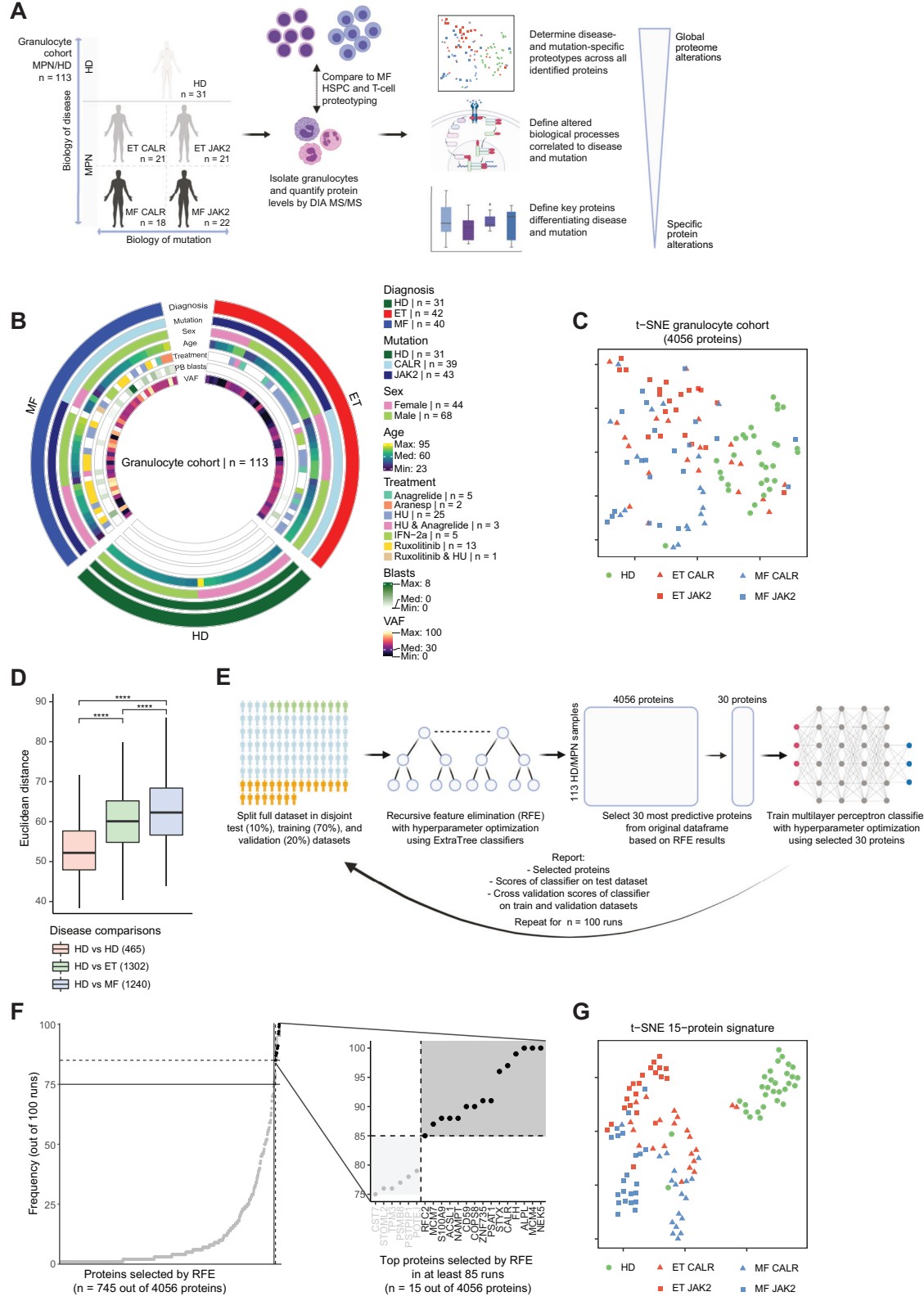

the MCM-high replicative phenotype, possibly representing a therapeutic strategy for this advanced MF patient subset.

**Homozygous *CALR* mutations lead to ER stress-associated vulnerabilities**

Our global cellular proteotype analysis further identified MF-specific upregulation of proteins involved in protein processing in the ER, which were additionally correlated positively to *CALR* VAF

across MPN samples (Fig. 6B). These proteins comprised a network of ER chaperones and co-chaperones including HSPA5 that physically interact with CALR. In contrast, intracellular CALR protein abundance itself negatively scaled with *CALR* VAF, likely due to secretion of mutant CALR protein[37]. These results indicate that molecular differences between MF patients reflect not only their driver mutation but also the respective mutation burden (Supplementary Fig. 7a).

**Fig. 5 | Large-scale clinical proteotyping elucidates the MF-specificity of targetable protein alterations. A** Workflow for the proteotyping of myeloproliferative neoplasm (MPN) samples. Proteotypes of granulocytes isolated from healthy donors (HD; *n* = 31) and a clinical cohort of essential thrombocythemia (ET; *n* = 42) and MF patients (*n* = 40) from two different medical institutions were analyzed using DIA MS/MS. HD are age- and gender-matched to the MPN cohort. **B** Circos plot of clinical annotations for the HD/MPN granulocyte cohort. Legend indicates color codes for the included variables. For discrete variables, patient numbers are included, and for continuous variables, the range and median are represented. Extended clinical annotations can be found in Supplementary Data 1. **C** t-SNE embedding of granulocyte proteotypes. Dimensionality reduction is based on all quantified proteins (n = 4056). MPN patients and HD are labeled by disease and mutation status. **D** Proteotype similarities between HD and MPN subcohorts. Boxplots show distributions of Euclidean distances between proteotypes of selected subcohorts (self-comparisons were excluded). Asterisks indicate non-adjusted two-sided Student's *t*-test significance: **** = $p < 0.0001$; exact p-values are

reported in Source Data. Boxplots as in Fig. 1A. **E** Workflow overview of machine learning-based approach to identify the minimal protein signature that classifies HD and MPN patients by disease and mutation status. The dataset is split into disjoint test, train, and validation cohorts, after which a recursive feature elimination (RFE) model is applied to prioritize the proteins most discriminatory for disease and mutation status. Based on the selected proteins, a multi-layered perceptron classifier is trained, of which the scores and selected proteins are reported. The final protein signature is derived from the most frequent scoring proteins. **F** Protein selection frequencies across 100 RFE runs. The main panel shows protein selection frequencies (*y*-axis) are shown for all proteins selected at least once (*x*-axis). Insert shows the same data for the most frequently selected proteins. The gray box indicates the 15-protein signature of proteins selected in at least 85 runs. **G** t-SNE embedding of the 15-protein signature for MPN patients and HD. MPN patients and HD are labeled by disease and mutation status. See also Supplementary Figs. 4 and 5.

We expanded on this finding by analyzing the correlations of protein levels with mutant allele burden within MF patient granulocytes, stratified by driver mutation. Whereas ribosomal proteins commonly associated with both *CALR* and *JAK2* VAF, proteins involved in protein processing in the ER were specifically correlated to *CALR* VAF (Fig. 8A, Supplementary Fig. 7b). The highest correlating proteins to *CALR* VAF (Fig. 8B, C) yet not *JAK2* VAF (Supplementary Fig. 7c) constituted a co-regulated network of physically interacting ER stress-related proteins. An intrapatient comparative analysis of genetically stratified single-cell mRNA data[39] confirmed increased levels of the ER stress signature in *CALR*-mutated HSPCs also on a transcriptional level (Supplementary Fig. 7d). The strongest negatively correlating proteins with *CALR* VAF do not physically interact yet are all glycoproteins (Fig. 8B). These included MPO and EPX, both glycoprotein chaperone clients previously reported to be absent from homozygous *CALR*-mutated granulocytes[40].

As we defined the CALR MF patients with high VAF (VAF ≥ 75%) as homozygous CALR, we next compared the cellular proteotypes of homozygous CALR granulocytes to all other MF samples. This consistently revealed strong upregulation of a chemical ER stressor-induced proteotypic signature[41] (Fig. 8D). Upregulation of ER stress proteins in homozygous MF CALR was not only detected in granulocytes but also in T-cells and HSPCs (Fig. 8E, left panel). We confirmed causality by genetic introduction of the *CALR* mutation in cell lines, which led to upregulation of the ER stress protein signature, not observed upon introduction of the *JAK2* mutation (Fig. 8E, right panel). ER stress upregulation related to a loss of function upon *CALR* mutation, as the phenotype was reproduced upon *CALR* knock-out in the same genetic background (Fig. 8E, right panel). Analyzing patient PBMC samples using the image-based CALRm readout further confirmed a significant correlation at the single-cell level of mutant CALR and HSPA5 expression levels (Fig. 8F). Thus, the presence of the *CALR* mutation relates to ER stress both across and within patient samples.

We next measured the functional consequences of the observed ER stress signature using a dedicated drug panel that includes investigational ER stressors and inhibitors of the unfolded protein response (UPR), the cellular survival pathway activated upon ER stress[42]. Within two MF PBMC samples, these two drug classes led to a specific depletion of CALR mutant-expressing cells (Fig. 8G). Furthermore, we investigated the drug response differences of the homozygous CALR patient samples compared to all others in the PBMC cohort (Fig. 8H). Homozygous CALR led to striking sensitivity to the protein translation inhibitor omacetaxine mepesuccinate, the CDK9 inhibitor alvocidib, and, in line with our molecular findings, the proteasome inhibitor carfilzomib (Fig. 8H, I, Supplementary Fig. 7E). As reduction of CALRm signal could either indicate a killing of CALRm-expressing cells or a reduction of CALRm expression in otherwise viable cells, we investigated this further by assessing the effect of carfilzomib treatment on

HSPCs isolated by flow cytometry. Comparing MF CALR heterozygous and homozygous HSPCs, we validate the highest reduction of CALRm intensity in homozygous MF CALR and find that this coincides with a reduction of cell viability and induction of apoptosis (Fig. 8J, Supplementary Fig. 6e). Notably, next to the aforementioned specific sensitivities, homozygous CALR patient samples showed resistance to BCL2-family inhibitors navitoclax and venetoclax (Fig. 8H, I). In conclusion, homozygous CALR present in a subset of MF patients leads to ER stress associated with sensitivity and resistance to clinically relevant drugs.

## Discussion

Here we identified and investigated targetable molecular vulnerabilities underlying MF using integrated drug response profiling and cellular proteotype analyses across clinically annotated patient cohorts. We established a high-throughput readout of mutant CALR expression and JAK/STAT hyperactivation at the single-cell level based on deep learning-driven image analysis, which enabled on-target drug sensitivity profiling of MF patient blood samples at scale. We explored specific drug sensitivities associated with (1) MF driver mutations, (2) elevated proliferation characterized by high MCM complex levels, and (3) elevated ER stress induced by homozygous *CALR* mutations (Supplementary Fig. 8).

The single-cell resolution of our image-based readouts allows us to investigate the functional consequences of oncogene expression in the context of cellular heterogeneity. We found considerable variability of mutant CALR expression correlated with JAK/STAT activation and ER stress, both within and between cell types in the same sample. We further observed lineage-dependent heterogeneity, as HSPCs and monocytes showed high CALRm levels across the cohort, while T-cells showed low levels. Part of this cell type-specificity could potentially be explained by differences in mutation penetrance, as MF driver mutations are commonly dominant in HSPCs and myeloid cells such as granulocytes and monocytes, yet rare in T-cells for both MF CALR[43] and MF JAK2 patients[44]. However, both the single-cell heterogeneity and lineage expression pattern were maintained in patients in which genetically all cells homogeneously carry homozygous mutations. Therefore, non-genetic regulation both across and within cell populations likely contributes to heterogeneous mutant protein expression and resulting biology.

Our drug response results indicated that *CALR* and *JAK2* driver mutations lead to divergent cellular proteotypes and associated drug sensitivities in MF. MF patients are commonly genotyped in clinical routine, yet recurrent driver mutations currently do not provide predictive information to guide treatment decisions. We found genetic and molecular stratifiers of response to both BET and BCL-2 inhibitors, two drug classes currently under investigation in clinical trials for the treatment of MF[45,46]. Subgroup analyses of clinical trial results for these

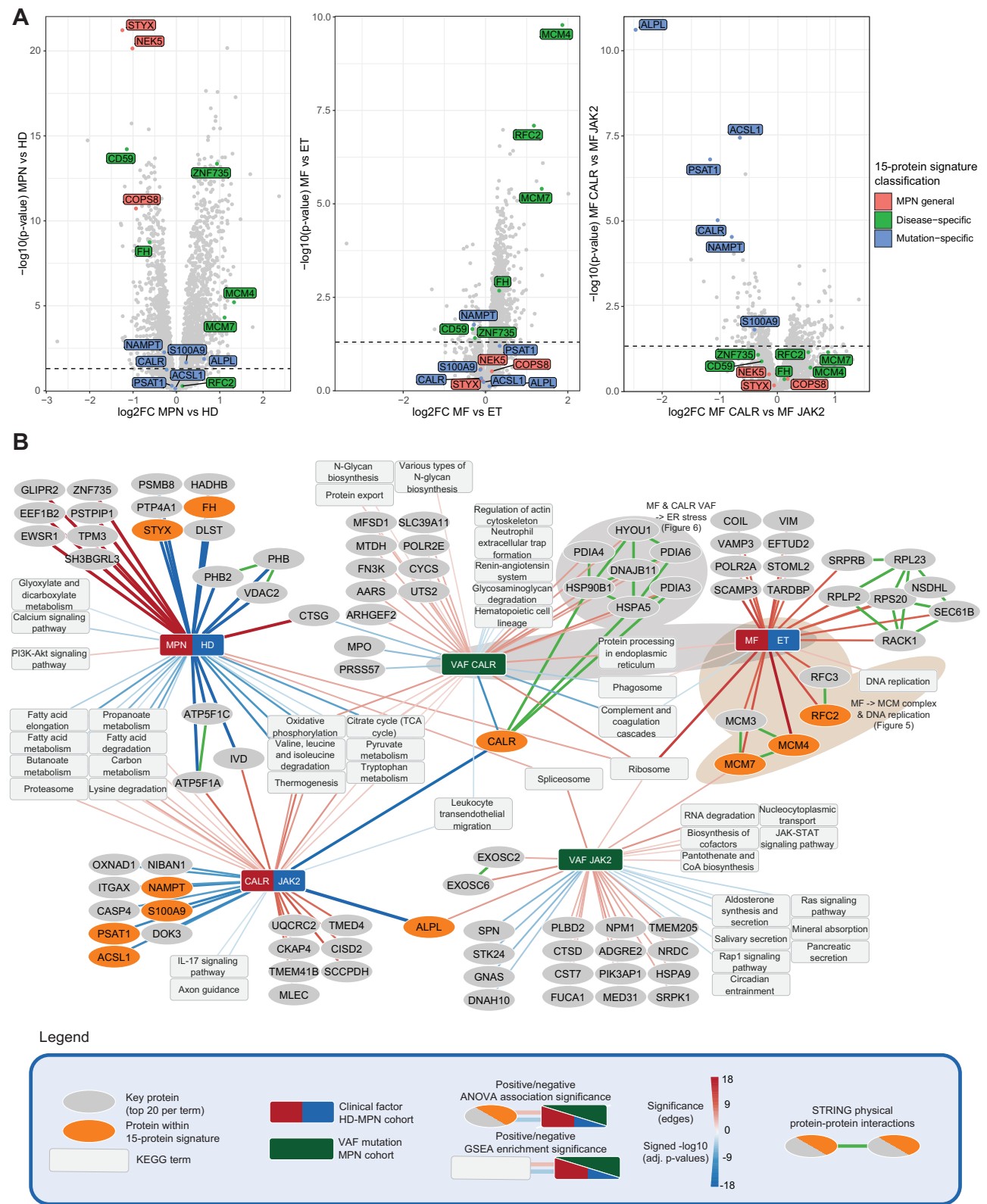

treatment options can strengthen our findings and contribute to a refined personalized treatment of MF. In light of our successful identification of driver mutation-specific MF biology, larger cohort studies following the here-established strategy will be instrumental in elucidating the functional influence of the multitude of reported secondary mutations[7]. The platform could further be developed to investigate cell-extrinsic drug responses of myelofibrosis by analysis of spatial and multicellular phenotypes ex vivo, a strategy recently proven successful for delineating immunotherapy and immunomodulatory responses in varying contexts[47–49].

The strong association we report between MCM expression levels, replication fork speed, and proliferative potential across MF patients suggests that specific adaptations of the DNA replication process are instrumental in sustaining pathological HSPC proliferation in MF. Interestingly, replication fork acceleration was recently reported to support induced HSPC proliferation upon simulated viral infection in

**Fig. 6 | Protein and pathway-level alterations linked to disease and driver mutation status. A** Volcano plots of comparisons underlying the classification of the RFE-selected protein signature (n = 15). Volcanoes indicate log2FC (x-axis) and significance (y-axis) of Student's t-test results comparing granulocyte protein levels for the indicated groups. Members of the 15-protein signature are highlighted and colored by reaching significant differences only in the HD vs. MPN comparison (MPN general alterations), in the MF vs. ET comparison (Disease-specific alterations), or also in the MF CALR vs. MF JAK2 comparison (Mutation-specific alterations). **B** Association network of protein and pathway-level alterations linked to disease and driver mutation status. For every protein quantified in the granulocyte proteotypes, expression levels were associated with general disease (MPN vs. HD), diagnosis (MF vs. ET), and mutation status (CALR vs. JAK2) by ANOVA analysis. For all clinical factors, the top 20 significantly associated proteins with an adjusted

ANOVA p-value < 0.001 are shown. The edges connecting these proteins to factors represent the corresponding signed ANOVA significance. For a pathway-level view, all signed ANOVA p-values were ranked per factor, after which GSEA enrichment analyses were performed. Significantly enriched KEGG pathway terms are linked to the respective factors with edges representing signed GSEA p-values (adjusted p < 0.05). For discrete clinical factors, red edges indicate association with the first term and blue with the second term. For VAF, red edges indicate positive and blue negative association. Orange proteins indicate those proteins within the network that are also members of the 15-protein signature. Green edges represent STRING-defined physical protein-protein interactions (physical interaction STRING score > 0.7). Brown and gray backgrounds indicate subnetworks of interest that are experimentally followed up on. See also Supplementary Fig. 5.

mice[50] and is modulated in human cells by excess chromatin loading of the MCM complex and the associated factor MCMBP[51]. This, therefore, could be in line with the high replication fork speeds as determined in HSPCs obtained from ET patients and MF patients with low MCM levels. In contrast, the normalized DNA replication fork speed of MCM-high MF patients may be a required adaptation to specific DNA replication constraints associated with disease progression, posing a promising target in advanced MF.

We showed that ER stress levels in MF patients increase with *CALR* mutation burden and that HSPA5 expression strongly correlated with mutant CALR expression, independently of *CALR* mutation type. Further, either knockout or introduction of the frameshift mutation induced ER stress, indicating it results from CALR loss-of-function and does not require mutant-induced JAK/STAT hyperactivation. Upregulation of ER stress and concurrent activation of UPR were first described at the transcriptional level in granulocytes of *CALR*- compared to *JAK2*-mutated ET patients[52]. Cell line studies have, however, reported impairment of the UPR upon *CALR* mutation[53] and UPR activation specifically in response to type I over type II *CALR* mutations[54]. Finally, at a single-cell level, transcriptional UPR activation was found in *CALR*-mutated compared to WT HSPCs within individual ET and MF patients[39] and in mouse models of MF[55]. We found the ER stress phenotype to be strongest in homozygous CALR patient samples. Here, despite the highest *CALR* VAF and mutant CALR expression levels by immunofluorescence, total intracellular CALR levels detected by proteomics were lowest, consistent with a loss-of-function phenotype of wild-type CALR. Consistently, we found this patient subcohort to be particularly sensitive to the ER-stressor carfilzomib, UPR inhibition by experimental compounds, and ribosome inhibition by omacetaxine mepesuccinate.

Our study thus provides a rich resource of clinically annotated integrated molecular and functional data that enables further exploration of fundamental insights into MF biology. The genetic, molecular, and functional MF patient stratification outlined here can inform future clinical trials and provide a concrete roadmap for improved personalization of MF treatment.

## Methods

### Granulocyte isolation and processing

Peripheral blood samples were collected upon written informed consent according to the Declaration of Helsinki from MF and ET patients diagnosed according to the World Health Organization (WHO) classification[56]. Patients were included who visited the University Hospital Zürich in the period of April 2018–June 2019 or the University Hospital Basel in the period March 2015–May 2019. The study was approved by the local ethics committee (KEK-ZH-NR: 2009-0062/1 and BASEC-NR: 2018-00539) and the Ethik Kommission Beider Basel. Healthy donor samples were collected from coded blood donors by the Blutspende Zürich under a study protocol approved by the Cantonal Ethics Committee, Zürich (KEK Zürich, BASEC-Nr 2019-01579). The HD cohort was age- and gender-matched to the patient cohort,

and the size (n = 31) was chosen to allow for averaging out the biological variation between healthy individuals.

As input for granulocyte isolation, per sample, either 10–40 mL of MPN patient peripheral blood or one HD buffy coat (~50 mL of erythrocyte- and plasma-depleted blood) was used. Density gradient centrifugation purification was performed using Ficoll–Paque PLUS (GE Healthcare), after which the remaining erythrocytes were removed from the granulocyte pellet by double ACK hydrolysis for the discovery cohort and single ACK hydrolysis for the validation cohort. Granulocyte cell pellets counted using a hemocytometer were snap-frozen in liquid nitrogen immediately after isolation and stored in liquid nitrogen until processing. All granulocyte pellets were processed in duplicate and fully randomized in a single batch. In short, 1 million granulocytes were lysed in 0.5% SDS containing TCEP and CAA. SDS was removed with 8 M Urea using the FASP procedure[57] in 96-well plates (PALL). Protein concentrates were solubilized and processed, digested, and cleaned in 96-well S-Trap plates (ProtiFi) according to the manufacturer's protocol. Dried peptides were resuspended in 5% acetonitrile and 0.1% formic acid supplemented with iRT peptides (Biognosys), and peptide concentrations were normalized.

### Granulocyte proteotyping

For spectral library generation of all patient and HD samples, 1 μg peptide was pooled and subjected to high pH-RT fractionating using an HPLC 1260 (Agilent). Seventy-two fractions were collected and pooled into twelve samples for subsequent MS analysis. Furthermore, 1 μg peptides of patients grouped by disease state and driver mutation were pooled. Unfractionated total cohort pooled samples were injected between every 10 runs to monitor LC–MS/MS performance. Total and subtype-specific pooled fractionated and unfractionated were run on a Fusion Lumos mass spectrometer (Thermo Scientific) using a 2 cm Acclaim™ PepMap™ 100 C18 HPLC trap column (Thermo Scientific) and 25 cm EASY-Spray™ HPLC analytical column (Thermo Scientific) set-up connected to an EASY-nLC 1200 instrument (Thermo Scientific). Peptides were loaded in 100% buffer A (98% $H_2O$, 2% acetonitrile, 0.15% formic acid) and eluted at a flow rate of 250 nL/min with a segmented 2-h gradient from 1 to 58% buffer B (80% acetonitrile, 0.15% formic acid). The method for data-dependent acquisition (DDA) was Orbitrap-based and consisted of a 3 s cycle time with an MS1 scan over a scan range of 350–1650 m/z with 60,000 resolution and maximum injection time of 60 ms and default charge state 2. The isolation window for MS2 was set to 1.6 m/z and collision energy to 27% HCD, and fragments were measured in the Orbitrap with a resolution of 15,000 and dynamic exclusion of 60 s. MS performance was monitored using QuiC (Biognosys). A spectral library was built integrating the fractionated and unfractionated DDA runs using ProteomeDiscoverer 2.4 (Thermo Scientific). The spectral library workflow included a first MSPepSearch scoring using the ProteomeTools HCD28 spectral library, after which low-scoring peptides by Percolator was subjected to a second Sequest HT search.

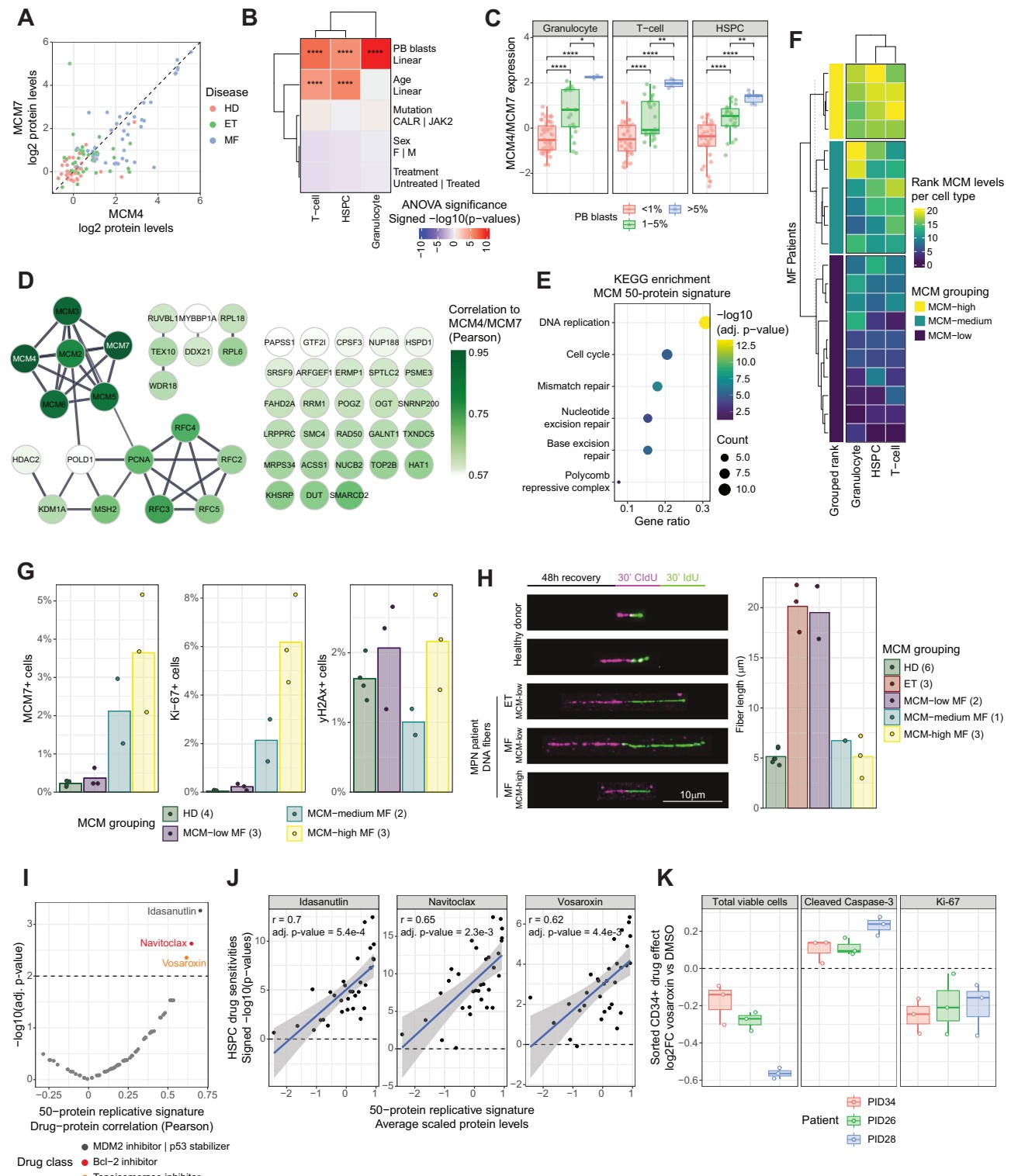

Individual patient and HD peptide samples were run on the same LC–MS/MS set-up and gradient as for DDA library generation runs but in data-independent acquisition (DIA) mode. The Orbitrap-based method used contained 40 dynamic windows over a scan range of 350–1650 m/z with 30,000 resolution and 27% HCD collision energy and a survey scan with 120,000 resolution and a maximum injection time of 100 ms and default charge state 2. Raw files were converted to HTRMS using HTRMSConverter (Biognosys) and analyzed in Spectronaut 13.9 (Biognosys) applying the "only protein group-specific" proteotypicity filter, and otherwise, the standard manufacturer's settings, after which common contaminants were removed, and Spectronaut output files were exported.

## MF PBMC isolation

Peripheral blood samples were collected from MF patients visiting the University Hospital Zürich in the period of December 2012–May 2021 upon written informed consent according to the Declaration of Helsinki. The study was approved by the local ethics committee (KEK-ZH-NR: 2009-0062/1 and BASEC-NR: 2018-00539). Patient peripheral blood mononuclear cells

**Fig. 7 | Proliferative MCM-high MF responds to the targeting of survival signaling and DNA replication. A** Correlation of MCM4 and MCM7 protein levels across the granulocyte cohort. Levels normalized by subtracting the median HD expression. **B** ANOVA significance of averaged MCM4 and MCM7 protein level associations with clinical parameters across the MF cohort. The red color indicates a positive association with continuous parameters (PB blasts and age) or to the first term of discrete factors (mutation, sex, and treatment). **C** *Z*-score normalized MCM4 and MCM7 protein levels across MF patients stratified by PB blast counts (<1%, $n = 64$; 1–5%, $n = 39$; >5%, $n = 6$). *p*-Values indicate the Student's *t*-test significance. **D** Protein–protein interaction network of the top-50 proteins correlating to MCM4 and MCM7 protein levels across T-cell, HSPC, and granulocyte proteotypes. Edges: STRING score > 0.7. Proteins colored by average Pearson correlation. **E** Pathway enrichment analysis results of the 50-protein signature. Significant positive KEGG term enrichments ($p < 0.05$) are shown. **F** Ranking of averaged MCM4 and MCM7 protein levels per cell type and MF patient for the 20 MF patients present in both the granulocyte and MF PBMC cohorts. **G** Quantification of the percentage of MCM7, Ki-67, and yH2Ax positive PBMCs. Dots summarize the staining quantification of 2–4 patients (brackets) per MCM group averaged across three replicate wells and 25 images per well. Bars indicate the average per group. **H** Representative images of in vitro HSPC DNA replication fork fibers (left panel) and corresponding quantification (right panel). Dots summarize average fiber lengths of HSPCs isolated from 1 to 6 patient samples (brackets) per MCM group with 93–370 fibers per sample. Bars indicate the average per group. **I** On-target HSPC drug responses correlated to the average expression of the 50-protein replicative signature. *x*-axis, Pearson correlations; *y*-axis, corresponding significance. **J** Scatterplots of the three highest correlating drug responses of (**I**). Linear fit (blue line) and 95% confidence intervals (gray area) are shown. Pearson correlation and significance are indicated. **K** Vosaroxin responses of FACS-isolated CD34 + MF patient HSPCs from three vosaroxin-sensitive patients. Asterisks indicate non-adjusted two-sided significance: **** = $p < 0.0001$, *** = $p < 0.001$, ** = $p < 0.01$, * = $p < 0.05$; exact *p*-values are reported in Source Data. Boxplots as in Fig. 1A. See also Supplementary Fig. 6.

(PBMCs) were isolated by density gradient centrifugation purification using Ficoll-Paque PLUS (GE Healthcare) and viably stored in FCS supplemented with 10% DMSO in liquid nitrogen until further use. For selected samples, CD34+ cells were isolated from patient PBMCs using MACS bead isolation (Miltenyi), which were similarly viably stored in FCS with 10% DMSO in liquid nitrogen until further use. Upon use, frozen PBMCs were rapidly thawed at 37 °C, washed, and treated with DNAse for ten minutes at 37 °C to prevent clotting. Afterward, recovered cells were counted, and for pharmacoscopy and immunofluorescence, PBMC-based assays were used directly.

### MF T-cell and HSPC proteotyping
For HSPC and T-cell proteotyping, cells were enriched from thawed PBMCs using CD34+ and CD3+ MACS isolation kits (Miltenyi), respectively, according to the manufacturer's protocol. We have found this protocol to yield a median CD34+ enrichment purity of 98% across 73 independent isolations for the included MF patients. In the case of low viable cell counts after the thawing of PBMC samples, pharmacoscopy-based drug response screening was prioritized over proteotyping (<4E6 viable PBMCs, 3/43 samples). Next, samples were fully randomized for batch processing, accounting for driver mutation, MF subtype, age, gender, and sample date. Peptides were isolated from up to 1e6 cells of the respective samples using the iST 96× kit (PreOmics), using the manufacturer's protocol and a 3-h digestion. Dried peptides were resuspended in LC-LOAD (PreOmics), and peptide concentrations were determined using the Pierce Quantitative Colorimetric Peptide Assay (Thermo Scientific) according to the manufacturer's protocol. Peptide concentrations were normalized and supplemented with iRT peptides (Biognosys). LC–MS/MS was performed in DIA mode on a Q Exactive HF-X mass spectrometer (Thermo Scientific) using the standardized Moonshot DIA protocol[58]. Raw files were directly imported into Spectronaut 15.0 (Biognosys), and features were extracted in a library-free method using directDIA. Specificity filtering was set to "protein group-specific", Lys-C was added as a digestion enzyme, a regular q-value cutoff was used for detection, and other settings were kept to the standard manufacturer's settings. The Spectronaut output files were exported.

### Protein-level identification and quantification
Protein quantification and subsequent statistical analyses on DIA features extracted from Spectronaut were performed using MSStats[59] version 3.16. Features were log-transformed, normalized, and summarized per protein. In case of availability of technical replicates, analyses were run twice: on the single MS run level for QC analyses and otherwise summarized per patient or cell line for further analyses and plotting. In all datasets, outliers were determined on a level of (a) correlation to all other samples of the same isolated cell type (Supplementary Fig. 4a, d) and (b) missing values after peptide q-value cutoff filtering (Supplementary Fig. 4b, d). Most outliers in the HSPC and T-cell cohorts had low protein levels of the respective markers used for isolation, CD34 and CD3, indicating outlier status relating to isolation impurities (Supplementary Fig. 4c). On the resulting dataset with missing values, we applied stringent filtering to remove all proteins not found in 90% of all patients of at least one group. In this way, we can account for group-specific protein expression in, for example, only MPN patients but not HDs, while removing all proteins that have random missingness and thus suffer from less robust quantification. After filtering, for the remaining missing values, imputation was performed, replacing the missing value with the lowest detected value for that protein and up to five percent added noise.

### Granulocyte proteotype 5-class classifier with RFE
The granulocyte proteotypes are split into five groups: HD, ET CALR, ET JAK2, MF CALR, and MF JAK2. 100 runs are performed, in which, in each run, classifiers are trained on selected proteins. In short, the data is first split into disjoint training, validation, and test sets of 70%, 20%, and 10%, respectively. Next, the training dataset is split six times for cross-validated recursive feature elimination (RFE) using grid-optimized hyperparameters until 30 features are left. The expression values of the RFE-selected proteins are used to train a grid-optimized and cross-validated multilayer perceptron classifier (MLPC). For all 100 runs, the performance of the resulting MLPC is tested on the initial test dataset, of which the test scores (accuracy, weighted F1, and macro F1) are saved together with the cross-validation scores and selected protein features.

### Hemoglobin regression
To combat inter-cohort differences leading to data batch effects despite randomized peptide isolation and LC–MS/MS analysis, red blood cell (RBC) contaminations were computationally resolved (Supplementary Fig. 4e, g). Average intensity of the major RBC proteins hemoglobin A, B, and D was calculated (Supplementary Fig. 4f). With this average level, a linear regression model was calculated for every protein present in the dataset. Correlation with the hemoglobin contents was regressed, after which the original protein values were replaced by the residuals that were used in all subsequent analyses (Supplementary Fig. 4h).

### Granulocyte variant allele frequency (VAF) quantification
Genomic DNA was isolated from -1E6 granulocytes per sample using the QIAamp DNA Micro Kit (QIAGEN) according to the manufacturer's protocol. VAF quantification of isolated granulocyte DNA was

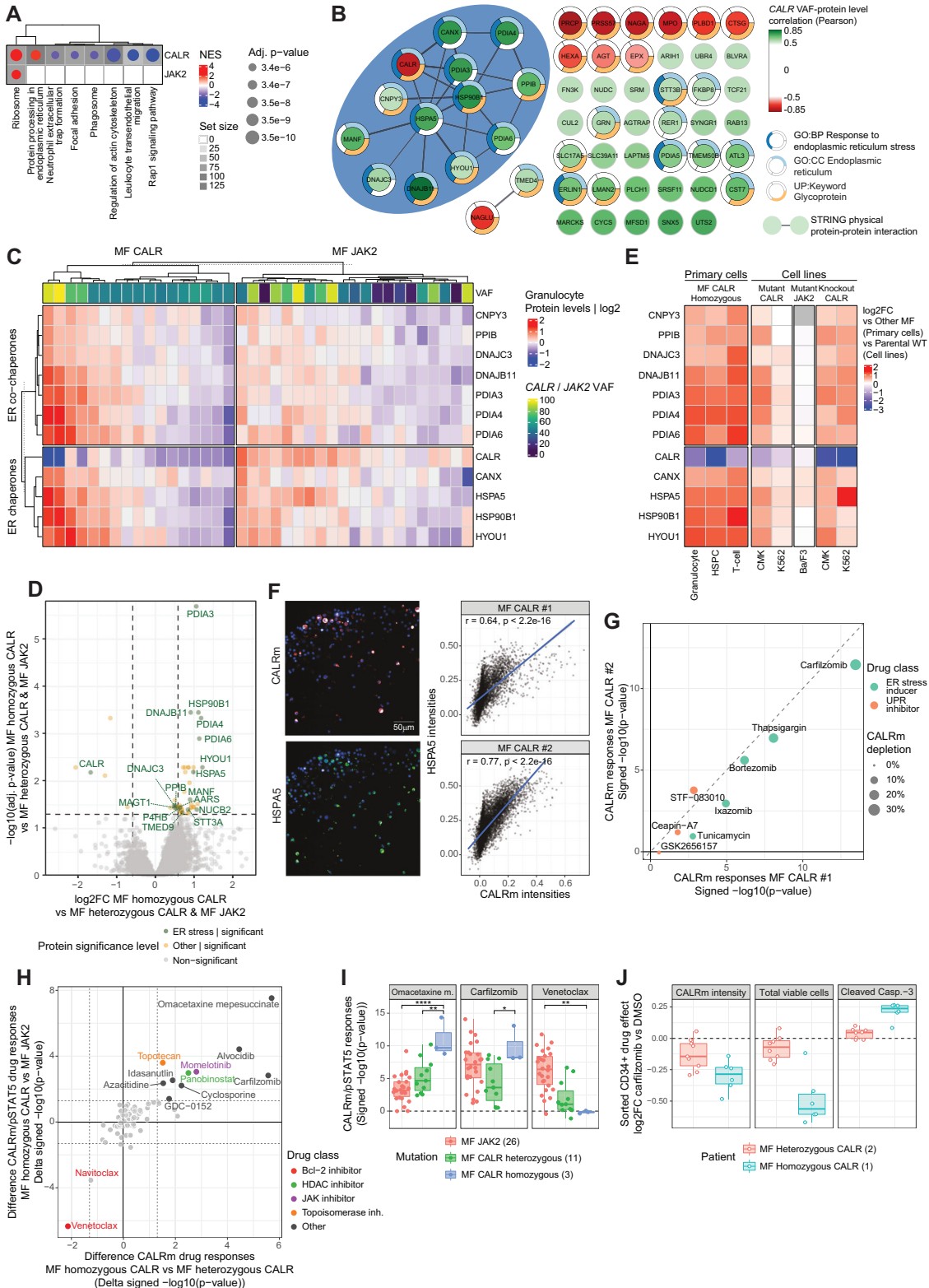

subsequently performed using ddPCR[60] for the determination of *CALR* type 1, type 2, and *JAK2* V617F mutations. For the other CALR mutations, a fragment analysis PCR was performed[61]. When there was no cell material available for genomic granulocyte DNA isolations but mutation quantification was available from clinical NGS data, those were included. ADVIA cytometry measurement of MPO activity was further used to assess CALR homozygosity status, in line with previous findings[40].

**Granulocyte proteotype association network analysis**
For every protein, expression levels across the HD and MPN granulocyte cohort were associated with general disease (MPN vs. HD), diagnosis (MF vs. ET), and mutation (CALR vs. JAK2) by ANOVA analysis. For VAF associations, mutation-stratified MPN subcohorts (*CALR*-mutated ET and MF; *JAK2*-mutated ET and MF) were analyzed by ANOVA analysis integrating VAF and diagnosis as a confounding factor to negate diagnosis-associated VAF differences.

**Fig. 8 | Homozygous *CALR* mutations lead to ER stress-associated vulnerabilities. A** Gene set enrichment analysis (GSEA) of ranked correlations between granulocyte protein levels and *CALR/JAK2* granulocyte VAF. KEGG terms with an adjusted enrichment $p < 0.0001$ are shown. **B** Protein–protein interaction network of proteins significantly correlating to *CALR* VAF. Proteins with significant Pearson correlations are shown ($p < 0.01$). Edges: physical interaction STRING score $> 0.7$. Node color: Pearson correlations; Outer rings: selected protein pathway memberships. Blue shading indicates the interaction cluster of CALR and associated ER proteins. **C** Heatmap of ER stress cluster protein expression levels across MF patient granulocytes. Patient VAF and protein class are indicated. **D** Volcano plot of MF CALR homozygous versus all other MF CALR proteotypes. Proteins are colored green and annotated if significantly different and previously described to be upregulated in ER-stressed cell lines[41]. **E** Heatmap of ER stress cluster protein expression levels across cell types and cell lines. Protein levels normalized as log2FC as indicated in the legend. **F** Single-cell expression levels of CALRm correlated with HSPA5 across PBMCs of two MF CALR patients. Representative images (left panel) and quantification (right panel) of four replicate wells with 25 images per well are shown. Representative results of two independent repeats are shown. Pearson correlation and significance are reported. **G** Significance of depletion of CALR mutant expressing cells by ER stress inducers and UPR inhibitors across two MF CALR patient samples. The dashed line represents the diagonal. **H** Comparison of oncogenic drug responses between MF CALR homozygous and either MF CALR heterozygous (*x*-axis) or MF JAK2 (*y*-axis). Axes indicate an averaged difference in signed −log10 significances. Selected drugs are labeled and colored by drug class. **I** Oncogenic drug responses correspond to selected MF CALR homozygous-specific drug responses shown in (**H**). Boxplots indicate the range of drug responses across the MF cohort, dots represent individual patients. **J** Carfilzomib responses of FACS-isolated CD34+ MF patient HSPCs. Boxplots represent log2FC differences between DMSO- and drug-treated conditions. Asterisks indicate non-adjusted two-sided Student's *t*-test significance: **** = $p < 0.0001$, ** = $p < 0.01$, * = $p < 0.05$; exact *p*-values are reported in Source Data. Boxplots as in Fig. 1A. See also Supplementary Fig. 7.

## Mutant CALR and pSTAT5 antibody conjugation

The monoclonal antibody SSI-HYB 385-06 Anti Frameshift Mutated Calreticulin (Anti Crtfs), IgG1/Kappa, clone SSI-4F10 (Art. No. 100808) (from now on CALRm) was developed and manufactured by Statens Serum Institut, Copenhagen, Denmark. The CALRm and pSTAT5 antibodies were both conjugated using an in-house developed conjugation procedure. Briefly, the antibody storage solution was exchanged to coupling buffer (PBS with NaHCO3 and NaOH) by Zeba Spin Desalting Column (Thermo Scientific)-mediated buffer exchange. Antibodies were incubated with NHS-reactive Alexa Fluor 647 or 488 (both Thermo Scientific) for one hour at room temperature. Afterward, using another Zeba Spin-mediated buffer exchange, unconjugated fluorophores were removed, and the antibody was eluted in an antibody storage solution (BSA with NaN3). Antibodies were aliquoted and stored at 4 °C until further use.

## Detection of mutant CALR by intracellular flow cytometric analysis

To investigate the detection of mutant CALR by the CALRm antibody, CMK wild-type, knockout, and mutant cells were harvested, washed with phosphate-buffered saline (PBS), and stained with the Zombie Aqua™ Fixable Viability dye (Biolegend) for 20 min at room temperature in the dark. After washing, cells were fixed and permeabilized using the Intracellular Fixation & Permeabilization Buffer Set (eBioscience) according to the manufacturer's instructions. After fixation and permeabilization, cells were incubated with CALRm for 60 min at room temperature in the dark. Cells were subsequently washed and analyzed on a BD LSRFortessa™ flow cytometer.

## MF PBMC immunofluorescence and pharmacoscopy

Thawed PBMCs were counted, and 10,000 cells per well were seeded in RPMI medium supplemented with 10% human serum on top of 384-well plates with pre-dispensed drug libraries. The drug library was composed to consider MF patient eligibility (i.e., no chemotherapies), maximize pathway diversification, include pathways targeted by drugs currently used and in development for MF, and possess pathway redundancy. All drugs were randomized across the plate layout with duplicates of 1 μM and 10 μM final concentrations, except for antibodies that had three replicates and 0.1, 1, and 10 μg/ml final concentrations. Cells were incubated with drugs for 20 h at 37 °C and 5% $CO_2$. Drug and antibody concentrations were based on previous experience[17,18,48,49,62,63]. Drug concentrations were validated as clinically predictive in two separate interventional clinical trials, treating patients with leukemias or lymphomas based on their ex vivo drug responses to drugs tested at these concentrations (EXALT-1 and DARTT-1[17,18,63]) and shown to be predictive of clinical response in a multi-year observational study on multiple myeloma[49].

The next day, cells were fixed with 1% PFA for 20 min. After that, cells were blocked, and permeabilized, and nuclei were stained with 10% BSA, 0.1% Triton, and 0.25 μg/uL DAPI for 30 min in the dark. Subsequently, plates were stained overnight at 4 °C with an antibody mix (Table 1). Throughout, liquid handling steps that interact with non-adherent cells in the well are kept to a minimal pipetting speed to prevent loss of cells. Single wells were used to demultiplex the CD3/CD14 stains in the green channel. Plates were imaged by confocal microscopy using an Opera Phenix (PerkinElmer) at 20× magnification with 25 imaged sites per well. For test stains, the same procedures were followed without exposing the cells to drugs and with additional sets of antibodies (Table 1).

## Image immunofluorescence and pharmacoscopy analysis

Cells were identified from images using CellProfiler (Broad Institute of Harvard and Massachusetts Institute of Technology) based on nuclear detection by DAPI signal and size extrapolation for cellular cytoplasm[62]. Dead cells, dying cells, and incorrectly segmented cells or debris were removed using a CNN-based approach training on manually curated cell crop images. This live/dead CNN classifier was trained on a total of 86,000 manually annotated image crops of individual cells (comprising 1000 live and 1000 dead cells from each of the 43 MF patients). Manual labeling was performed by transfer learning and expert labeling based on nuclear (DAPI) and cell (brightfield) morphologies. This allowed for the training of a CNN-based classifier that detects live cells with an accuracy of 90.3% and a sensitivity of 90.5%.

For pharmacoscopy, 4-class cell type identities were determined similarly using a ResNet CNN trained on a total of 6600 manually curated cell crop images (150 per cell type from each of the 43 MF patients). For comparison, a feature-based CNN was trained in parallel by segmenting the cells in the images and analyzing mean channel features as opposed to raw images. This feature-based network has not been used further; cell type classification and all further analyses have been performed using the higher-performing image-based CNN. As selected MF samples had very low HSPC cell type fractions, a minimum threshold of 0.3% HSPCs was required for further inclusion in drug response analyses (40/43 samples). Drug pharmacoscopy scores were calculated as the relative reduction of cell population fractions of interest compared to the corresponding fractions in DMSO-control wells for small molecules, PBS-control wells for biologicals, and isotype-control wells for antibodies[17].

## High-resolution imaging

Cells were stained with a similar protocol as above, yet instead of in 384-well plates, they were seeded on μ-Slide VI imaging slides (Ibidi). Imaging was performed using a Nipkow spinning disk microscope

**Table 1 | Antibodies used in this study and their respective applications**

| Antigen | Company | Clone | Conjugation | Application |
|---|---|---|---|---|
| CALRm | Statens Serum Institut | 385-06 | In-house: AF647 | Pharmacoscopy MF CALR patients and cell lines |
| pSTAT5 | Invitrogen | Polyclonal | In-house: AF647 | Pharmacoscopy MF JAK2 patients |
| CD3 | BioLegend | UCHT1 | AF488 | Pharmacoscopy all patients |
| CD14 | BioLegend | HCD14 | AF488 | Pharmacoscopy all patients |
| CD34 | BioLegend | 581 | PE | Pharmacoscopy all patients |
| CALR | Abcam | FMC 75 | PE | Pharmacoscopy cell lines |
| HSPA5 (GRP78) | Invitrogen | C38 | AF488 | Immunofluorescence (IF) imaging |
| pSTAT5 | Invitrogen | Polyclonal | In-house: AF488 | IF imaging |
| CALX | Abcam | EPR3632 | AF488 | IF imaging |
| GM-130 | Abcam | EP892Y | AF594 | IF imaging |
| IdU | BD Biosciences | B44 | Unconjugated | DNA fibers |
| CldU | Abcam | BU1/75 (ICR1) | Unconjugated | DNA fibers |
| IgG1 (mouse) | Invitrogen | Polyclonal | AF488 | DNA fibers |
| IgG (rat) | Jackson ImmunoResearch | Polyclonal | Cy3 | DNA fibers |
| Ki-67 | Cell Signaling Technology | D3B5 | AF647 | IF imaging and CD34+ drug response analyses |
| Cleaved caspase-3 | Cell Signaling Technology | D3E9 | AF647 | CD34+ drug response analyses |

(Visitron) with a W1–T2 Confocal Scanner Unit (Yokogawa), on which, for selected fields, 100 Z-stack cell images were captured. These images were processed using ImageJ[64], in which the 25–30 Z-stacks that captured the cell best were flattened to a single image by average intensity representation.

## Cell line CRISPRing and proteotyping
For comparison to patient cells, the following cell lines carrying *CALR* mutations were included: CMK11-5, from which the wild type and the CRISPRed clone 751[23] were kindly provided by K. Shide, and K562, from which the wild type, CRISPRed mutant and knockout clone were kindly provided by A. Vannucchi. To complement the CMK background with a knockout counterpart, CMK11-5 wild-type cells were CRISPRed using a ribonucleoprotein (RNP)-based strategy[65]. After confirmation of CRISPRing in bulk isolated DNA, a single-cell sort was performed. Various clones could be confirmed to be homozygous knockouts on both a DNA and protein level. For cell line proteotyping, for the CMK dataset, the same unlabeled DIA quantification workflow as for the granulocyte proteotyping was used. For the K562 dataset, isolated peptides were TMT-labeled and measured on a Fusion mass spectrometer (Thermo Scientific) using a DDA MS3-based quantification approach as previously described[66]. The dataset of the proteomic analysis of *JAK2*-mutated compared to wild-type Ba/F3 cell lines of Pearson et al.[67] was additionally included. Given the differences in quantification, all protein quantities were only analyzed as log2FC relative to the corresponding parental wild types present in the respective datasets.

## Cell line pharmacoscopy
CMK and MARIMO cell lines were cultured in RPMI supplemented with 10% FBS and antibiotics. Cells were seeded on 384-well plates at a concentration of 5000 cells/well and incubated, processed, and imaged, using the same protocol as for PBMCs, except for fixation and the antibody panel used. For fixation, cells were incubated for one hour in 4% PFA, followed by antigen retrieval using incubation with Urea and Tris for 10 min at 80 °C. The antibody panel used consisted of CALRm and CALR WT antibodies. For test stains, the same procedures were followed without exposing the cells to drugs. For the computational analyses, a similar strategy as for patient PMBCs was followed with an adjusted CellProfiler pipeline and CNN for the cleanup of images. WT, MUT, and KO cell lines were distinguished based on the gating of single-cell CALR WT and CALRm intensities.

## DNA replication fork fiber assays
Frozen CD34+ cells from a liquid nitrogen biobank were rapidly thawed at 37 °C, washed, and treated with DNAse (ITW Reagents) for ten minutes at 37 °C to prevent clotting. Afterward, the recovered cells were counted and cultured in complete X-vivo media (Lonza) for 48 h containing GlutaMAX (Thermo Scientific), SCF, TPO, and Flt3 (all PreProTech). After recovery, cells were incubated with subsequent CldU and IdU (both Sigma)-containing X-vivo media, both for 30 min at 37 °C. Nucleotide incorporation was stopped by addition and washing with ice-cold PBS, after which cells were concentrated, resuspended in the remaining volume, pipetted onto imaging slides, and lysed in an SDS lysis buffer. After 5 min of lysis, slides were tilted to spread the DNA fibers. The slides were dried and fixed in methanol:acetic acid overnight at 4 °C, after which DNA was denatured using HCL and subsequently blocked using BSA. The DNA was stained using primary antibodies against IdU (BD Biosciences) and CldU (Abcam) and respective secondary antibodies (anti-mouse IgG1 AF488, Invitrogen; anti-rat Cy3, Jackson ImmunoResearch). The DNA was imaged using a fluorescence microscope at 63X magnification. Finally, the length of the fibers was measured using imageJ[64].

## Drug response analyses of flow-sorted CD34+ cells
Thawed PBMCs were resuspended to $1 \times 10^8$ cells/mL in PBS supplemented with 2% FBS. Cells were stained with CD34 antibodies (Table 1) and LIVE/DEAD™ Fixable Near-IR Dead Cell Stain Kit (Invitrogen) in the dark on ice for 30 min to enable sorting of viable CD34+ cells. Stained cells were washed with PBS and resuspended at a final concentration of $1 \times 10^7$ cells/ml in 2% FBS. Cell sorting was performed with an Aria Fusion flow cytometer (BD Biosciences) at 4 C. Purified CD34+ viable cells were collected in 10% FBS in RPMI prior to seeding in drug-loaded 384-well plates and incubated for 20 h at 37 °C and 5% CO$_2$. Finally, cells were fixed, blocked, and stained as described above in "MF PBMC immunofluorescence".

## Data analysis, visualization, and statistics
Data from proteotype and validation experiments were analyzed and visualized in R 4.2.0. Pharmacoscopy analyses were performed in MATLAB R2021a and visualized in R 4.2.0. Flow cytometry data was analyzed using FlowJo software (FlowJo Enterprise, version 10.0.8, BD Biosciences).

No statistical method was used to predetermine the sample size. Unless otherwise stated, significance values were calculated with a Student's t-test. Where significance is not shown, it did not reach

*p* < 0.05. Spearman's rank correlation or Pearson correlation coefficients and significances are reported for all scatterplots. Unless stated otherwise, data distributions were assumed to be normal, but this was not formally tested. Where applicable, data distributions are shown. Drug screening plate layouts were randomized across the wells of each 384-well plate. For proteotype analyses, the sample preparation and measurements were randomized for disease and mutation status. For the HSPC and T-cell proteotype analyses, additional randomization was included for common confounders such as age and sex. Otherwise, no randomization was performed as part of this study. Data collection and analysis were not performed blind to the conditions of the experiments. Further information on research design is available in the Nature Research Reporting Summary linked to this article.

## Reporting summary

Further information on research design is available in the Nature Portfolio Reporting Summary linked to this article.

## Data availability

Source data are provided in this paper. The mass spectrometry proteomics data have been deposited to the ProteomeXchange Consortium via the PRIDE[68] partner repository with the dataset identifier PXD036075. Additionally, Supplementary Data 1 contains the full clinical annotations of the different included cohorts, Supplementary Data 2 the processed pharmacoscopy drug response matrices, and Supplementary Data 3 the processed proteotype matrices. Source data are provided in this paper.

## Code availability

Image analysis was performed using the open-source CellProfiler package available at https://www.cellprofiler.org. All other analyses were performed using standard MATLAB R2021a and R 4.2.0 code.

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

## Acknowledgements

The authors thank all the members of the Snijder, Theocharides, and Wollscheid laboratories for their assistance and helpful discussions. We thank Tina Friis and further involved employees of the Statens Serum Institut Copenhagen for providing us with the CALR mutant-specific antibody. The authors acknowledge the Scientific Center for Optical and Electron Microscopy (ScopeM) at ETH Zurich for their support with the spinning disk confocal imaging. We thank the patients whose samples were included in this study. Grant support: This work was supported by the Swiss National Science Foundation (SNF project grant 310030_170237, to A.P.A.T. and B.W.) and the Personalized Health and Research Technologies (iDoc grant PHRT-326) strategy focus area (SFA) of the ETH domain (to M.H.E.W., B.S., A.P.A.T, and B.W.). A.P.A.T. was supported by the Professor Dr. Max Cloëtta foundation. Figs. 2A, 5A, E, and Supplementary Fig. 8 were created using a licensed version of BioRender.

## Author contributions

Conceptualization: M.H.E.W., B.S., A.P.A.T. and B.W.; Methodology: M.H.E.W., M.v.O., J.M. and C.D.; Software: M.H.E.W., J.M., J.S., Y.S.,

P.G.A.P. and B.S.; Formal analysis: M.H.E.W., J.M., C.D., J.S. and Y.S.; Investigation: M.H.E.W., J.M., M.v.O., C.D., Y.F., V.L., P.S., S.G., A.v.D and B.D.H.; Resources: S.B., R.C.S., B.S., A.P.A.T. and B.W.; Data curation: A.R., M.S.B., R.C.S and A.P.A.T.; Writing—original draft: M.H.E.W., B.S., A.P.A.T. and B.W.; Writing—review & editing: M.H.E.W., B.S., A.P.A.T., B.W., M.L., R.C.S., M.v.O., P.S., Y.F., A.R., S.G. and C.D.; Visualization: M.H.E.W.; Supervision: B.S., A.P.A.T., B.W. and M.L.; Funding acquisition: M.H.E.W., B.S., A.P.A.T. and B.W.

## Competing interests

B.S. was a co-founder of Allcyte GmbH, which has been acquired by Exscientia. B.S. is a shareholder of Exscientia. The remaining authors declare no competing interests.
