## [Peer Review File · Nature Communications]

REVIEWER COMMENTS

Reviewer #1 (Remarks to the Author): expertise in drug screening methods

Wildschut, et al. provide an in-depth analysis of the drivers of myelofibrosis (MF) stemming from a combined molecular profiling and functional analysis of patient-derived blood cell populations and CRISPR-edited cell lines. The authors identify three distinct vulnerabilities in MF patient samples, each of which can be targeted by existing drugs currently in trial or clinically available. The novelty of this manuscript lies within the application of an existing functional screening approach in tandem with proteomic characterization and molecular profiling to develop new treatment paradigms for patients with MF. Though this study focuses on a myeloproliferative neoplasm, the approach could be generalized to other types of blood cancers.

Overall, this is an exciting work that provides novel therapeutic avenues and robust evidence that functional drug screening approaches in tandem with -omics data can be leveraged to personalize treatment selections for patients with cancer. The article nicely complements similar studies demonstrating the utility of functional drug screening, particularly for blood cancers. Yet, it extends well beyond published work by integrating several data types to uncover MF biology and devise therapeutic strategies.

Specific Comments and Questions:

1. How are the drug concentrations selected for screening, do they match PK/PD clinical data?
2. It is highly advisable to include some additional methodological details to favor reproducibility, such as fixation and staining process, adjustments that are made to ensure cells are not lost etc
3. Development of specific antibody-based readouts is integral to the paper and well described in the results. However none of the data is included in the main figures – it would make sense to do so.
4. How large was the dataset used for training the CNN to distinguish live and dead cells? What is the accuracy and sensitivity of this system? Was manual labeling performed by semi-quantitative measures or by an expert in evaluating cell fitness based on morphology?
5. It is unclear whether there are limitations related to sample collection that need to be considered, such as minimum bounds for the prevalence of each cell type. It may be useful to include a description of which sample criteria must be met for patient samples to undergo this analysis
6. Which data was used to perform the t-SNE embedding in Figure 1B?
7. In Figure 2D, low expression of Ras signaling pathway members is identified as a driver of sensitivity. However, several other pathways show similar NES. Are these relevant? If not, what are the criteria for exclusion
8. The DNA replication fork speed of MCM-high MF patients remains similar to that in the HD population. What is the relevance of this?

Minor: the first paragraph of the results section is missing a heading. Page 7, Line 241 – the text currently reads Figure 5A and 3H. 3H appears to refer to 3G. Reference to Jacobs et al, submitted should be eliminated – either a preprint is cited, or data is shown.

Reviewer #2 (Remarks to the Author): expertise in myelofibrosis genomics and treatment response

In the present work the authors describe a powerful high-throughput in vitro screening system for the evaluation of drug activity in PBMCs from myelofibrosis patients. The strength of this systems relies on the capacity to discriminate between cells harboring CALR or JAK2 mutations. The presence of CALR variant is evidenced by the detection of the mutated protein using a specific antibody, while the presence of JAK2 mutation is determined by evaluating JAK/STAT pathway activation through phospho-STAT5 antibody. The proposed tool allows to prioritize the selection of drugs based on their effect on HSPCs and mutated cells. By coupling this system with parallel proteotyping of HSPCs and T-cells and proteotyping of granulocytes from an independent patients' cohort authors were able to describe specific therapeutic vulnerabilities correlated with patients'

molecular profile.

Even if the present work is sound, interesting and well conducted, there are some criticisms that needs to be addressed by the authors.

Major

1. In supplementary table 1 primary and secondary myelofibrosis cases are distinguished according to column "specific diagnosis". It would be helpful for a general readership to specify the distinction between primary and secondary myelofibrosis progressing from ET and PV in the introduction. Furthermore, I did not find any caption for supplementary tables where abbreviations are defined, it would be helpful for comprehension.
2. Line 111: what do the authors mean with homogeneously? Based on boxplots shown in Figure S2A single-cell mean intensity of mutant CALR signal display high dispersion. Does the antibody stain all CMK11-5 cells, which is the frequency of stained cells in each image? In reference 22, the antibody used was able to detect intracellular mutant CALR in white blood cells from MPN patients by means of flow cytometry. Can the authors evaluate its expression also by means of flow cytometry both in cell lines and patients' samples to confirm antibody staining?
3. Line 119: Frequency of mutated cells shown in Figure S2D does represent mean or median values of frequency readout of how many replicates/images per sample? Can the authors include this information in figure legend and rename bars accordingly to the related images?
4. Line 120: Since the authors showed CALRm intensity for Cell lines it would be useful to represent patients' heterogeneity by showing signal intensity also for MF PBMCs shown in Figure S2D.
5. Line 150: Did the authors looked for a direct correlation between the size of image-based HSPC fractions and blast percentage detected in patients? Can they perform a correlation analysis (e.g. Pearson or spearman correlation)? Did the authors evaluate blast immunophenotype for the studied patients? Did the blasts express CD34 antigen?
6. Figure 1D: CALR mutant patients were classified as heterozygous or homozygous according to CALR VAF. According to Table S1, CALR VAF was established only in 9/16 patients, nevertheless it seems that patients with unknown VAF was included in heterozygous group. Why did the authors make this decision? Would it be possible to perform the analysis by including only patients with known VAF?
7. Why z-score normalization was performed per patient in Figure 1C and across patients in Figure 1D. Would it be possible to harmonize z-score normalization, or, if not, why does the authors believe that the different normalization does not impact the results?
8. Is there any correlation between pSTAT5 levels and JAK2 VAF?
9. Figure 1B: CALRm and pSTAT5 signals seems to be detected only in monocytes and HSPC, not in lymphocytes and other cells. Would it be possible to evaluate whether lymphocytes from analyzed patients harbor CALR or JAK2 mutations?
10. Figure 2C: it is not clear the comparison to which the p-value refers to? In this panel two different drug concentration were used but only one p-value is indicated for each drug. Moreover, Student's t-test is used to compare the mean values of two groups, in each sub-panel there are represented the mean values of three groups (CALR WT, CALR MUT, CALR KO). Can the Student's t test be applied when each group is composed by only two replicates? Does each group represent a normal distribution? The authors should be more precise in describing the statistical methods they adopted for their analysis including an appropriate methods section and providing more specific information in figures captions.
11. Figure 2: Did the authors evaluated the effect of HDAC inhibitors on CMK11-5 cell line as it is shown for BET inhibitors in Figure 2C?
12. Figure S5B: in main text authors say that MF represents the condition with the highest dissimilarity with HD, nevertheless, in figure S5B they show a network where the color of each node represents the fold change between MF and ET. Are these proteins differentially expressed also when considering the comparison between MF and HDs? Why did they decided to show the comparison between MF and ET? The maximum FC detected is only 1.6 ($\text{Log}_2\text{FC} = 0.68$), considering tSNE representation in Figure 5B it is not surprising since MF and ET samples are almost superimposable. It is expected that differences between MF and HDs are more pronounced. Does this difference involve the same proteins and pathways?
13. Figure 3G: It is not clear why, for the mutation-specific proteins only MF CALR and MF JAK2 samples are included, and it is not shown the expression in ET CALR and ET JAK2 samples. In methods sections authors says that they evaluated differential protein expression using ANOVA

analysis. Why are there shown Student's t-test results?

14. Figure 4: represented factors were selected only based on adjusted p-value or FC was used to select significantly deregulated proteins? Would it be possible to represent the results of ANOVA analysis, in a supplementary figure showing a volcano plot for each analysis performed, or in a supplementary table? It is not clear how ANOVA analysis was used to determine the correlation between protein expression and mutation VAF, why a Pearson or Spearman correlation analysis wasn't used instead of ANOVA?

15. Line 230: according to authors results, MF diagnosis was associated with the expression of proteins involved in DNA replication. This result came from the proteotype analysis of granulocytes which represent terminally differentiated post-mitotic cells. Can the authors comment on this? Did the authors evaluate granulocyte purification efficiency (e.g. morphologic evaluation or flow cytometry)?

16. Would it be possible to validate MCMs expression in granulocytes, T-cells and/or HSPCs by means of western blot analysis and at transcript level by means of real time qRT-PCR? Authors should validate the expression of deregulated proteins, like 15-protein signature and HSPA5 by means of other techniques such as western blot analysis or real time qRT-PCR?

17. Line 288: According to figure 1B MF patients' lymphocytes do not express mutated CALR. Can the authors comment on their observation that lymphocytes from homozygous CALR patients display upregulation of ER stress proteins. How can it be related with homozygous CALR when these cells do not express mutant protein? This relates to major comment #9.

18. Line 333: Strikingly, this study lack a functional evaluation of the drugs identified using the high throughput model developed by the authors. Can the authors evaluate the effect on proliferation and survival on sorted cells from MF patients? (e.g. CD34+ cells and/or CD14+ monocytes). I mean, would it be possible to isolate CD34+ cells and Monocytes from MF patients homozygous or heterozygous for CALR, treat them with an UPR inhibitor or ER stressors and evaluate the effect on cells proliferation, viability, apoptosis? This would be the first step for a subsequent in vivo testing. The same should be done for Vosaroxin and can be overlooked for molecules already adopted in clinical trials.

Minor

- Supplementary figure 1A and F: for continuous variables like VAF can the color scale be modified to make the intermediate color equivalent to the median value.
- Figure 5H: in box plot y-axis change uM with μ M, moreover the name of each group should be corrected by substituting MF with MCM.

Reviewer #3 (Remarks to the Author): expertise in proteomics

Wildschut et al. here present a study that has high clinical value for myelofibrosis patients and contains a wealth of data. Clinical and functional data has been integrated with proteomic data, with the latter being a major strength of the study.

The authors have in most cases successfully presented the data clearly (and beautifully), and managed to highlight the key conclusions.

I have a few comments for improvements. Most of these relate to the data that forms the basis for the rest of the manuscript. Nevertheless, the manuscript does already contain the validation of the conclusions, wherefore my comments can mostly be used to strengthen the conclusions further.

It is a major concern that the chosen isolation approaches for proteotyping are not high purity flow cytometry-based. It is likely a problem with low purity for especially the HSPC-enriched samples, because of the low frequency of CD34+ cells in the starting material (as the authors also show in Fig S3C). Therefore, it is desirable that the authors show the purity of each cell type and sample after isolation. The ranking plots of Suppl Fig 1 indicate there is high variability in purity, but a FACS plot would be more appropriate. If there is impurity in the isolation and large variability in purity between samples, the proteotype information does not accurately represent the cell type of interest, and the authors need to address this issue.

In addition, I do not find it correct to refer to it as "PBMCs were sorted", when in reality it is only an enrichment that has been done.

Somewhat related, the hemoglobin regression needs to be mentioned and explained in the main text.

It is potentially a problem that as many as 20 MF patients have been used in both cohorts. That is almost half of the patients from the first cohort, giving bias to high chances of finding the same pathways. How does the correlation in terms of enriched pathways look if these 20 patients are excluded from the second analysis?

Line 154: It would be desirable to show in the circos plot of Fig S1A which 3 samples/patients were excluded from drug response results.

Line 205: "Within MF, we found that their cellular proteotypes showed consistent pathway-level alterations with MF HSPC proteotypes (Figure S5A)."

I think the figure does not entirely justify this statement, and the figure is difficult to understand. In my view, the MF HSPCs have a few weakly enriched pathways that are overlapping with MF granulocytes, but it is a bit stretched to call it "consistent". That said, it is not totally unexpected that there would be many cell type-specific pathway alterations.

The authors can also help the reader with more details and explanations in the figure legend, especially by clarifying which input samples have been used, e.g. whether it is n=20 or n=40 for the HSPCs and T cells.

Are the outliers in Fig S1B, C and G filtered out? This is not clear in the methods, and it should be visible in Fig S1A and F which samples are outliers (and thus hopefully excluded from proteotype analysis).

Fig S1D shows that there is one T cell sample (rank 39) that has passed the outlier filtering despite a very low CD3 expression. Can the authors comment on the validity of the data from this sample?

Suppl Table S3 needs more description in title or legend for each tab - this data is a rich resource for the community and deserves better and more clear presentation. It is unclear for most tabs what values are shown. For example, why is there only one value per patient, and not one per drug?

Minor points:

The colour coding for mutation in the legend of Fig S1F seem to be incorrect (HD cohort has been coloured as JAK2 mutation).

Figure S2C: To say that the CALRm specifically results in sensitivity to JAK/STAT signaling inhibition, more than two replicates would make the statement better justified. Can the authors elaborate on why the KO responds like the MUT at 1 uM and like the WT at 10 uM? Alternatively, the figure can be shown as a response curve instead of bar graphs to show the differences between the three groups better (and by that justify the statement and significance)?

Figure S3F: Could the authors show this data as a scatter plot instead, to show the real correlation for CNN-based data and the counts for each patient?

Could the authors use the same colour coding for the different drug classes in Fig S4D and Fig 1E?

Abbreviations could be avoided in abstract.

With what approach was missing value imputation performed?

Reviewer #4 (Remarks to the Author): expertise in computational analysis of drug screening

The study by Snijder and coworkers is a well done and highly relevant report that reveals new methods and information of relevance to both basic and translational research.

The broader study of ex vivo analysis of cell types from myelofibrosis patients incorporates next-generation methods in precision medicine. It highlights several new concepts in disease classification based on the authors proteotype analysis. Finally, it associates several of the newly revealed disease proteotypes with potential therapies based on an ex vivo drug response signature.

The studies primary issue is presentation. The manuscript could easily be doubled in size to delve more deeply into the methods and data. The figures are rich with information that is often described in a brief sentence or two. The reader is left to sift through the methods and SI figures to better place key outcomes in the context that the authors intend. If word-count limits can be extended for this manuscript the authors should consider a more descriptive version of this paper be re-submitted.

Minimally, the main texts description (or figure or figure legend) of the drug exposure should include the drug concentration(s) and time of exposure.

The method is highly innovative and the authors deserve significant credit for working out key details that enable such a robust ex vivo scoring model. The use of 'sanity check' secondary methods like the CALRm and pSTAT5 imaging is also well done. The method still relies primarily on cell intrinsic factors to reveal drug sensitivities. For instance, the revelation that the 'MCM-high' proteotype is associated with cell cycle/DNA repair pathways and respond to inhibitor of those pathways/targets is straightforward and a highly actionable finding. The data around the JAK2 proteotype is possibly less straightforward. While activating mutations in JAK signaling is a hallmark of myelofibrosis, a purely cell intrinsic view of JAK signaling is probably insufficient to explain the findings in this study. For instance, only the more promiscuous JAK inhibitor pacritinib (misspelled in figure S2F) demonstrate a strongly response in the MF JAK2 proteotype while highly selective (and clinically valuable) ones like ruxolitinib are less active. Pacritinib is also a potent IRAK inhibitor. Could that be playing a role (particularly in the pSTAT5 intensity reduction)? The authors should review their consider and discuss.

The data and discussion around BET inhibitor and HDAC inhibitor signatures is interesting and potentially important. These mechanisms obviously play a role in multiple cancers and dozens of HDAC and BET inhibitors have entered human clinical evaluation. A reasonable therapeutic index for HDAC inhibitors has been found for multiple agents while this remains elusive for BET inhibitors which likely reflects the complex mechanistic consequences of these drugs. The data and discussion around the enrichments in this study are limited to a previously defined inverse correlation with RAS pathway activation. Overall, this wasn't the most impactful outcome from this study. Its completely up to the authors, but limiting the discussion around this finding to give room for expanding the method description and other findings would be fine.

The authors should discuss the lack of activity for key standards of care of myelofibrosis which are unlikely to demonstrate activity in a 24-hour ex vivo (in vitro) experiment. These include the 'imid' drugs like pomalidomide and lenolidomide and hedgehog and Smoothened modulators like glasdegib, vismodegib, and erismodegib. All precision medicine platforms will have to learn how to handle important drugs that rely on cell extrinsic factors which are often missing from ex vivo/in vitro environment and the authors should consider and discuss their methods and its limitations in predicting proteotypes which would benefit from these agents.

I was also intrigued that exportin inhibitors where not discussed. Selinexor (and Selinexor+JAK inhibitor) therapy is demonstrating promise in myelofibrosis. Did the Selinexor responses not show any enrichment?

Finally, the outcomes with the BCL-2 inhibitors (Navitoclax and Venetoclax) are of crucial importance. A deeper dive into the secondary signatures for responding versus non-responding

would greatly increase the impact of this work.

Overall, this is an excellent study and report with dual relevance to the myelofibrosis community and the precision medicine community.

Minor requests/comments

The authors should explain what they mean by BET and BCL-2 family inhibitors having a 'common mechanism' (page 5, line 167).

Figure 3 and the main text description of the data presented in this figure are sparse enough that it limits a broader understanding of the deeper dive into specific proteins/pathways revealed in this study. If the authors can expand on this it would be good.

Craig Thomas

Reviewer #5 (Remarks to the Author): expertise in single cell image and deep learning analysis

The authors presented drug screening and proteotype profiling to shed light on vulnerabilities of myelofibrosis (MF), a hematopoietic stem cell disorder belonging to myeloproliferative neoplasms or mutated blood progenitor cells. The technique was applied to patient blood using high-content imaging, mass spec-based protein profiling, and deep learning based single cell analysis. Biological findings included (1) CALR mutations linked to BET and HDAC inhibitor sensitivity, (2) MCM complex high proliferation linked to pro-survival signaling and DNA replication, and (3) homozygous CALR mutations linked to high ER stress. While the paper demonstrates interesting vulnerabilities in drug sensitivities in MF, there are multiple major concerns:

1) The novelty seems to be very limited. Pharmacoscopy (cited in 16, Lancet), multiplexed single cell imaging and CNN classification (cited in 21, now published in Sci Adv), and conventional DIA LC-MS/MS proteomics. The abstract and manuscript are a bit misleading: "We developed a pharmacoscopy drug screening approach". These claims should be toned down to reflect accurate prior art in this line of work. Brief and relevant details of each approach should be detailed for readers to follow the methodologies.

2) While image-based single cell analysis of "blood cells" has been utilized by the authors in recent papers (since 2017), side-by-side comparisons to IMAGE STREAM and multiplexed FLOW CYTOMETRY of the same or even more number of antibodies can be screened and used for drug response screening. Image-based multiplexing has been powerful for the spatial context of tissues or subcellular organization. However, imaging blood cells (no spatial context) is less interesting unless authors perform a spatial feature analysis (nucleus, cytosol, organelles, and other pixel-level features). Even Figure 6F indicates single cell intensities in x-y axes (that the same can be done with flow cytometry or image stream cytometry). I highly recommend subcellular spatial analysis for revisiting drug responses.

3) Image-based CNN quantification shows 90.7% accuracy but is this clinically reproducible in all the MF samples and benchmarked by a simple approach for cell phenotyping? I highly recommend a systematic approach to delineate the (1) technical errors from (2) biological variability in drug testing and quantification.

4) When integrating image-based drug targets with MS/MS based proteotype drug targets, what are the normalization steps involved to take into account dynamic range, protein specificity, and sensitivities? This is a concern beyond using z-scores for analysis.

5) After establishing the image-based assay with the abovementioned rigor and proper data integration with MS/MS data, the biological insights may need to be revisited as they were provided in (1) CALR mutations linked to BET and HDAC inhibitor sensitivity, (2) MCM complex high proliferation linked to pro-survival signaling and DNA replication, and (3) homozygous CALR mutations linked to high ER stress.

Overall, the proposed approach, after maturation and extensive validations, may be applicable to other disorders, including blood and solid tumors.

1 **RESPONSE TO REVIEWERS' COMMENTS**

2 **Reviewer #1 (Remarks to the Author): expertise in drug** 3 **screening methods**

Wildschut, et al. provide an in-depth analysis of the drivers of myelofibrosis (MF) stemming from a
combined molecular profiling and functional analysis of patient-derived blood cell populations and
CRISPR-edited cell lines. The authors identify three distinct vulnerabilities in MF patient samples, each
of which can be targeted by existing drugs currently in trial or clinically available. The novelty of this
manuscript lies within the application of an existing functional screening approach in tandem with
proteomic characterization and molecular profiling to develop new treatment paradigms for patients
with MF. Though this study focuses on a myeloproliferative neoplasm, the approach could be
generalized to other types of blood cancers.

Overall, this is an exciting work that provides novel therapeutic avenues and robust evidence that
functional drug screening approaches in tandem with -omics data can be leveraged to personalize
treatment selections for patients with cancer. The article nicely complements similar studies
demonstrating the utility of functional drug screening, particularly for blood cancers. Yet, it extends
well beyond published work by integrating several data types to uncover MF biology and devise
therapeutic strategies.

**Specific Comments and Questions:**

**Reviewer 1 | Major 1**

How are the drug concentrations selected for screening, do they match PK/PD clinical data?

Based on previous experience with the pharmacoscopy (PCY) method for drug sensitivity screening,
we have used standardized concentrations of 1 μ M and 10 μ M for small-molecule drugs and 0.1 μ g/ μ L,
1 μ g/ μ L, and 10 μ g/ μ L for biologicals. Since we screen for 79 compounds in parallel, we have to limit
the amount of concentrations to ensure we have sufficient viable cells across our sample cohort. While
we initially - in prior work when starting to screen in primary patient material - screened at drug
concentrations matching IC50 values in cell lines, we soon learned that primary patient samples
tolerate and resist considerably high drug concentrations. One important distinction between prior
drug screening campaigns in cell line models is that we do not score drugs that kill all cells, but we
score drugs based on their ability to specifically kill a target subset of cells (in this case, e.g. CALRm+
or CD34+ cells) while leaving alive the non-malignant healthy bystander cells. Critically, at exactly
these same concentrations and readout, we have now performed two *interventional clinical trials*
(EXALT-1 and DARTT-1) where lymphoma and AML patients received treatments based on exactly this
readout of ex vivo drug sensitivity at these drug concentrations and timepoints (Heinemann et al.,
2022; Kornauth et al., 2022; Schmid et al., 2023; Snijder et al., 2017). In both of these studies, we
observed improvements in clinical response rates, indicating that drug testing in primary patient

samples at these concentrations, timepoints, and with this readout, is predictive of clinical response
and helps guide the personalized treatment of individual patients. In the case of the pharmacoscopy-
guided treatment of AML patients (DARTT-1), the target population of CD34+ cells was identical to
that target population definition used in this study.

Furthermore, we have just published the results of a multi-year observational clinical trial on multiple
myeloma patients in Nature Cancer (Kropivsek et al., 2023). Here, again, we analyzed *ex vivo* drug
responses by pharmacoscopy at the aforementioned drug concentrations, timepoints, and relative
readout concept. Critically, we again find that also for this cohort, *ex vivo* drug responses at these
conditions are indicative of clinical responses (Figure 8), and show strong robustness to the *ex vivo*
drug responses measured (Extended Data Figure 6B), resulting from the relative readout (i.e. by not
scoring just total cell killing, as is the case for cell line assays).

Combined, this interventional and observational clinical trial data provides a very strong rationale for
the drug concentrations, timepoints, and readouts we have used in this study. The experimental
validation in various model systems (e.g. CRISPRed cell lines) further confirm that our *ex vivo* drug
response measurements lead to verifiable and validatable biological insights into the biology
underlying MF. We have now expanded the M&M section motivating the selection of assay conditions.

58 **Reviewer 1 | Major 2**

It is highly advisable to include some additional methodological details to favor reproducibility, such
as fixation and staining process, adjustments that are made to ensure cells are not lost etc

We thank the reviewer for the suggestion and now additionally describe steps to minimize cell loss in
the process. Please note that fixation and staining procedures were already detailed in the materials
and methods section, under the header “MF PBMC immunofluorescence and pharmacoscopy”.

66 **Reviewer 1 | Major 3**

Development of specific antibody-based readouts is integral to the paper and well described in the
results. However none of the data is included in the main figures – it would make sense to do so.

We agree with the reviewer and added data on the development and validation of the antibody-based
readouts to the main **Figure 1**.

73 **Reviewer 1 | Major 4**

How large was the dataset used for training the CNN to distinguish live and dead cells? What is the
accuracy and sensitivity of this system? Was manual labeling performed by semi-quantitative
measures or by an expert in evaluating cell fitness based on morphology?

We have added the requested details to the M&M section as well as below. However, we would first
like to provide some historical background. Firstly established in (Snijder et al., 2009, 2017), nuclear
and cell morphology can be used to quantitatively and reliably assess the viability of each cell by
automated confocal microscopy, and based on nuclear morphology obtained from DAPI imaging in
our workflow (reproduced below). Here (**below**, panel B from Figure 1 of (Snijder et al., 2017)) we
show that cell morphological analysis of cell death is highly reproducible to basing cell viability on a

cleaved-caspase 3 stain, leading to highly correlated drug response readouts (correlation = 0.99), with
 1 and 10 μM drug concentrations allowing to predict clinical response of AML patients to first-line
 chemotherapies (shown in the example **below** of Panel D, but more critically validated in both the
 EXALT-1 and DARTT-1 interventional clinical trials (Heinemann et al., 2022; Kornauth et al., 2022;
 Schmid et al., 2023; Snijder et al., 2017)).

Figure 1 of (Snijder et al., 2017)

 As a starting point for a live-dead classifier for MF cells, transfer learning of a previously established
 CNN trained for images of acute myeloid leukemia (AML) cells was utilized (e.g. used in (Shilts et al.,
 2022)). Single-cell crops of live- and dead-classified MF cells were saved and, based on DAPI and
 brightfield image layers, manually labeled by an expert, training a first CNN to classify MF images. This
 CNN was used to classify and save more cells per class that were again labeled by an expert, who
 labeled 1000 cells per patient per class (live/dead).

 To directly answer the three individual questions posed by the reviewer:

- 1. In total 86'000 single-cell annotated image crops (1000 live and 1000 dead cells for the 43
 patients) were included
- 2. This allowed for training of an optimized CNN-based classifier that detects live cells with an
 accuracy of 90.3% and sensitivity of 90.5% (included **below** and in **Figure S2C**)
- 3. Manual labeling was performed by transfer learning and expert labeling based on nuclear
 (DAPI) and cell (brightfield) morphologies

		True class		
Predicted class	Live	90.1	9.5	Live
	Dead	9.9	90.5	Dead
		Live	Dead	90.3

Figure S2C: Confusion matrix of the image-based CNN live/dead cell classifier. Diagonal numbers indicate prediction accuracies per cell type of the disjoint test set not used for training of the CNN, averaging to an overall 90.3% accuracy across all included MF PBMC samples.

**Reviewer 1 | Major 5**

It is unclear whether there are limitations related to sample collection that need to be considered,
such as minimum bounds for the prevalence of each cell type. It may be useful to include a description
of which sample criteria must be met for patient samples to undergo this analysis

We thank the reviewer for the suggestion and have added the following information to the relevant
materials and methods sections:

“MF and ET patients diagnosed according to the World Health Organization (WHO)
classification, who had signed informed consent, and visited the University Hospital Zürich in
the period of December 2012-May 2021 or the University Hospital Basel in the period March
2015-May 2019 were included in the study. In case of low viable cell counts after thawing of
PBMC samples, PCY was prioritized over proteotyping (< 4E6 viable PBMCs, 3/43 samples). An
HSPC fraction of >0.3% was required for PCY analysis (40/43 samples).”

As also requested by **Reviewer 3 in point 5**, we now indicate in the Circos plot (**Figure 2B**) which 3
samples were excluded for PCY drug response analysis after analyzing the cell population sizes by our
CNN cell classifier.

**Reviewer 1 | Major 6**

Which data was used to perform the t-SNE embedding in Figure 1B?

We thank the reviewer for raising this question and have expanded the corresponding figure legend.
For each of the 43 patients, 250 cells (or less if the population was smaller than 250 classified cells)
were randomly selected for each of the four defined cell populations (HSPC, T-cell, Monocyte, and
Others) within the DMSO-treated conditions. These make for the 41'286 cells that are shown in the
respective t-SNE plot. We now clarified this in the legend of **Figure 2D** (former Figure 1B) in the revised
manuscript. As already previously indicated in the figure legend, for each of these cells the CNN class
probabilities were analyzed by t-SNE, and the resulting embedding shown, colored by cell class.

**Reviewer 1 | Major 7**

In Figure 2D, low expression of Ras signaling pathway members is identified as a driver of sensitivity.
However, several other pathways show similar NES. Are these relevant? If not, what are the criteria
for exclusion

The Ras signaling pathway was highlighted as it was identified as a significant driver of sensitivity
(adjusted p-value <0.01) that was shared between the two drug classes investigated, BETi and HDACi.
However, as the reviewer pointed out other pathways are also relevant given their NES, particularly
for the BETi. Indeed, not all terms are relevant for HSPCs, such as the two other overlapping pathways,
“Platelet activation” and “Vascular smooth muscle contraction”, which also encompass various Ras
signaling pathway members and were given their lower relevance therefore not highlighted next to
the Ras signaling term. For BETi, most terms related to resistance reflect partially overlapping signaling
pathways with Ras, such as the MAPK, Chemokine, Rap1, VEGF, and PI3K-Akt signaling pathways. We
have added these and other labels to the volcanoes of **Figure 4D** (former Figure 2D).

**Reviewer 1 | Major 8**

The DNA replication fork speed of MCM-high MF patients remains similar to that in the HD population.
What is the relevance of this?

Indeed, despite being characterized by higher levels of proliferation, the speed of individually
measured DNA replication forks is not elevated in MCM-high MF patients. The investigation of this
presumed discrepancy is part of a large ongoing project that investigates DNA replication forks in MPN.
We postulate that alternative mechanisms, potentially including the increase in number of replication
forks, have enabled sustained high proliferation without occurrence of potentially detrimental effects
of high replicative stress and uncontrolled DNA replication forks in these cells. We discussed this in
more detail in the revised manuscript.

**Minor:**

**Reviewer 1 | Minor 1**

The first paragraph of the results section is missing a heading. Page 7, Line 241 – the text currently
reads Figure 5A and 3H. 3H appears to refer to 3G. Reference to Jacobs et al, submitted should be
eliminated – either a preprint is cited, or data is shown.

We thank the reviewer for pointing this out and have adapted the manuscript accordingly. The
manuscript by Jacobs et al. was accepted in Molecular Cell (Jacobs et al., 2022), and is now referenced
as such in the revised manuscript.

**Reviewer #2 (Remarks to the Author): expertise in**
**myelofibrosis genomics and treatment response**

In the present work the authors describe a powerful high-throughput in vitro screening system for the
evaluation of drug activity in PBMCs from myelofibrosis patients. The strength of this systems relies
on the capacity to discriminate between cells harboring CALR or JAK2 mutations. The presence of CALR
variant is evidenced by the detection of the mutated protein using a specific antibody, while the
presence of JAK2 mutation is determined by evaluating JAK/STAT pathway activation through
phospho-STAT5 antibody. The proposed tool allows to prioritize the selection of drugs based on their
effect on HSPCs and mutated cells. By coupling this system with parallel proteotyping of HSPCs and T-
cells and proteotyping of granulocytes from an independent patients' cohort authors were able to
describe specific therapeutic vulnerabilities correlated with patients' molecular profile.

Even if the present work is sound, interesting and well conducted, there are some criticisms that needs
to be addressed by the authors.

Major

Reviewer 2 | Major 1

1. In supplementary table 1 primary and secondary myelofibrosis cases are distinguished according to column “specific diagnosis”. It would be helpful for a general readership to specify the distinction between primary and secondary myelofibrosis progressing from ET and PV in the introduction. Furthermore, I did not find any caption for supplementary tables where abbreviations are defined, it would be helpful for comprehension.

We thank the reviewer for this comment and specified the distinction between primary myelofibrosis and post ET/PV myelofibrosis in the introduction. We have furthermore added a section (the second tab) in Supplementary Table 1 on all used abbreviations.

Reviewer 2 | Major 2

Line 111: what do the authors mean with homogeneously? Based on boxplots shown in Figure S2A single-cell mean intensity of mutant CALR signal display high dispersion. Does the antibody stain all CMK11-5 cells, which is the frequency of stained cells in each image? In reference 22, the antibody used was able to detect intracellular mutant CALR in white blood cells from MPN patients by means of flow cytometry. Can the authors evaluate its expression also by means of flow cytometry both in cell lines and patients’ samples to confirm antibody staining?

We agree with the reviewers point and have amended the phrasing “homogeneously”. We see throughout the images that most cells from mutated cell lines (CMK11-5 CALR MUT and MARIMO) are stained with the antibody. However, as the reviewer has noted, we see considerable variation in mutant CALR expression levels (measured as single-cell signal intensities); this coincides with an heterogeneous, rather than homogenous cellular staining pattern. Given this defined subcellular localization, the variation in quantified intensities might at least partly be explained by the variation in imaging planes that cross the cells and their subcellular organelles. We have, to address the reviewer’s question, thresholded the mean single cell CALRm intensities to determine frequencies of stained CMK11-5 cells. We would like to emphasize that we however, when analyzing the CMK11-5 panel drug responses, rather resort to a co-staining of the cells with CALRm and anti-CALR (wild type) followed by CNN-based cell classification, which we find to be a more robust method than mean single cell CALRm intensity thresholding.

As the reviewer points out, in (Mughal et al., 2022) we have used the CALRm antibody and shown that it is able to detect CALR-mutated MF patient cells by flow cytometry. As our cell line panel provides the cleanest model system for CALR mutant protein staining, we have now also evaluated CALRm staining by flow cytometry in this model. The data is shown below and included in **Figure 1A**, and indeed validates that the CALRm antibody specifically stains the CALR-mutated cell line similarly when read out by flow cytometry as it does in our imaging workflow.

 **Left:** Thresholded image-based quantification of CALRm intensities in the CMK cell line panel. CMK11-5 cells expressing wild
 type, mutant, or no CALR (CMK WT, MUT, and KO, respectively) were stained with CALRm. Data as shown in Figure S2A, yet
 instead of quantification of average intensities, CALRm-stained cells are quantified by thresholding of the single-cell
 intensities (stained = mean cellular CALRm intensity > 0.07, unstained =< 0.07). Frequencies of stained cells are determined
 239 per well and summarized as boxplots for n=4 replicate wells. p-values indicate Student's t-test significance: **** = p<0.0001.
 **Right (Figure 1A; right panel):** Flow cytometry-based quantification of CALRm intensities in the CMK cell line panel. CMK cells
 were fixed and permeabilized for intracellular staining with CALRm. Compensated APC intensities of live cells are summarized
 as boxplots. p-values indicate Student's t-test significance: **** = p<0.0001

**Reviewer 2 | Major 3**

Line 119: Frequency of mutated cells shown in Figure S2D does represent mean or median values of
 frequency readout of how many replicates/images per sample? Can the authors include this
 information in figure legend and rename bars accordingly to the related images?

The frequency is determined based on thresholding of the mean signal intensity values. For this and
 other quantifications, we quantify all cells detected in 100x images taken from 4x separate replicate
 wells. We have moved the figure to the main figures, and added this information to the figure legend
 of **Figure 1B**.

**Reviewer 2 | Major 4**

Line 120: Since the authors showed CALRm intensity for Cell lines it would be useful to represent
 patients' heterogeneity by showing signal intensity also for MF PBMCs shown in Figure S2D.

In the revised manuscript we now also show CALRm intensity for the MF PBMCs shown in **Figure 1B**
 (former Figure S2D). In contrast to the cell lines, in which we expect all cells to be of a similar cell type
 and have similar protein expression profiles, we see, as shown below, a large heterogeneity in CALRm
 intensities in MF PBMCs explained by the presence of diverse cell types including HSPCs, monocytes,
 T cells, etc. Notably, the fraction CALRm positive cells scales positively with the clinical diagnostic
 measurements of the VAF.

**Figure 1B:** Single-cell immunofluorescence imaging of MF PBMCs stained with CALRm. Representative images of different
 MF PBMC samples are shown (left panels; blue: DAPI; red: CALRm). Single-cell mean CALRm intensities of 4x replicate wells
 with 25x images per well are shown as a violin plot with percentages of thresholded CALRm-positive cells annotated as text
 (right panel). Violin colors indicate the *CALR* VAF of the corresponding MF patient.

 **Reviewer 2 | Major 5**

Line 150: Did the authors look for a direct correlation between the size of image-based HSPC
 fractions and blast percentage detected in patients? Can they perform a correlation analysis (e.g.
 Pearson or spearman correlation)? Did the authors evaluate blast immunophenotype for the studied
 patients? Did the blasts express CD34 antigen?

Correlation plots of PB blast and CNN-determined HSPC population sizes. X-axis shows clinically determined blast percentages in peripheral blood. Y-axis shows CNN-based HSPC population size for the DMSO-treated conditions of the pharmacoscopy data.

We thank the reviewer for the suggestion, and have correlated the “whole blood blast percentages” as determined in the clinic (by imaging, not immunophenotyping) with the image- and CNN-based quantification of CD34+ HSPC in PBMCs. While the exact percentages of both cannot be expected to be identical (different target cell population definitions in different total cell populations),

we are encouraged to see highly significant positive correlations both by Pearson and Spearman
correlation (see above).

A notable difference between the two readouts: In routine clinical practice MF patient blasts are
determined using imaging and not immunophenotyping, thus the blast CD34 antigen expression is not
available for us to compare to our PCY data and the clinically determined blast population could
include CD34-negative cells.

**Reviewer 2 | Major 6**

Figure 1D: CALR mutant patients were classified as heterozygous or homozygous according to CALR
VAF. According to Table S1, CALR VAF was established only in 9/16 patients, nevertheless it seems that
patients with unknown VAF was included in heterozygous group. Why did the authors make this
decision? Would it be possible to perform the analysis by including only patients with known VAF?

Our group has previously published that MPN patients with homozygous CALR mutations develop a
deficiency in myeloperoxidase (MPO) (Theocharides et al., 2016). MPO activity can be measured with
an ADVIA cytometer and clearly identify MPN patients with MPO deficiency. In our published work
MPN patients with a heterozygous CALR mutation all have normal MPO activity, while all patients with
a homozygous CALR mutation are MPO deficient. We therefore used MPO as a surrogate to confirm
the heterozygous CALR status of patients not analyzed by VAF. We now mention this in the M&M
section under VAF quantification.

Importantly, homozygous CALR mutations are relatively rare, and all homozygous CALR patients had
been confirmed by mutant allele burden analysis. As is also evident from **Figure 2F** (former Figure 1D),
leaving out a subset of heterozygous patients does not alter the conclusion of the analysis.

**Reviewer 2 | Major 7**

Why z-score normalization was performed per patient in Figure 1C and across patients in Figure 1D.
Would it be possible to harmonize z-score normalization, or, if not, why does the authors believe that
the different normalization does not impact the results?

We thank the reviewer for the opportunity to clarify this further. The reviewer is correct in the
difference in normalization, and we have clarified this further in the figure legends of **Figures 2E and**
**F** (former Figure 1C and 1D). The difference in normalization is motivated by the fact that we want to
show two different characteristics of the CALRm/pSTAT5 expression patterns identified:

- In **Figure 2E**, we focus on the expression pattern per patient, highlighting which cell types have
high or low CALRm/pSTAT5 intensities for each individual patient (intrapatient heterogeneity).
In order to remove differences in overall intensities and elucidate per patient patterns, we
need to normalize all intensities per patient.

- In **Figure 2F**, we want to show the differences in CALRm intensities across the MF CALR cohort,
between different patients rather than within an individual patient (interpatient
heterogeneity). In order to do this analysis, we should in contrast to Figure 2E not normalize

out the interpatient variation in overall CALRm staining. Therefore we normalize across the
cohort instead of per patient.

**Reviewer 2 | Major 8**

Is there any correlation between pSTAT5 levels and JAK2 VAF?

This is an interesting question. We have now correlated pSTAT5 levels with the JAK2 VAF, and do not
see a significant correlation.

While the CALRm levels and CALRm VAF correlate as they measure the genetic burden and protein
expression of the same underlying phenomenon (CALR mutations), JAK2 VAF and pSTAT5 levels are
two measurements of less directly related phenomena. Notably, pSTAT5 levels are not expected to
only occur in JAK2 mutant cells, and pSTAT5 levels also capture JAK-STAT signaling in non-malignant
cells resulting from e.g. inflammatory signals that are commonly present in MF. Indeed, we find that
pSTAT5 levels are consistently higher in MF samples in comparison to healthy donors (**Supplementary
Figure S1E**). We had already explicitly addressed this in the main text, by referring to the pSTAT5
readout as a proxy, not a direct readout, of hyperactivated JAK/STAT signaling. Thus, a drug-induced
reduction of pSTAT5 levels has a slightly different interpretation (and can also capture reduced
inflammation) that a reduction in CALRm levels.

Correlation plots of MF CALR and JAK2 VAF to concurrent average CALRm and pSTAT5 staining intensities, respectively.
Intensities are plotted and correlated for all four CNN-classified cell types. Pearson correlation coefficients and significance
levels are annotated.

**Reviewer 2 | Major 9**

Figure 1B: CALRm and pSTAT5 signals seems to be detected only in monocytes and HSPC, not in
lymphocytes and other cells. Would it be possible to evaluate whether lymphocytes from analyzed
patients harbor CALR or JAK2 mutations?

This is an interesting question that has already been performed in multiple independent and
previously published studies (El-Khoury et al., 2020; Larsen et al., 2007). While we did not perform
genotyping of purified lymphocytes from JAK2 and CALR mutant patients again, the previous

investigations have shown that the VAF of JAK2 and CALR is significantly lower - or even entirely absent
- in the lymphocytes of most patients analyzed (El-Khoury et al., 2020; Larsen et al., 2007). Therefore,
the data generated by PCY is congruent with previous data.

A small side-note: we would further like to clarify that a negative z-score intensity (as we show in
**Figure 2D** (former Figure 1B) and **Figure 2E-F**), does not per se mean the same as “no expression”, but
rather means “a lower than average expression”.

**Reviewer 2 | Major 10**

Figure 2C: it is not clear the comparison to which the p-value refers to? In this panel two different drug
concentration were used but only one p-value is indicated for each drug. Moreover, Student’s t-test
is used to compare the mean values of two groups, in each sub-panel there are represented the mean
values of three groups (CALR WT, CALR MUT, CALR KO). Can the Student’s t test be applied when each
group is composed by only two replicates? Does each group represent a normal distribution? The
authors should be more precise in describing the statistical methods they adopted for their analysis
including an appropriate methods section and providing more specific information in figures captions.

We agree with the reviewer and now describe the statistical method in more detail in the revised
manuscript. The p-value in **Figure 4C** (former Figure 2C) refers to the significance of the drug-treated
CMK CALR MUT conditions to the DMSO condition. Thus, the two replicates of the 1 μ M and the two
replicates of the 10 μ M drug-treated condition, four in total, are compared to the 25 DMSO control
data points. Normality of the complete drug response dataset was confirmed by qq-plot.

**Reviewer 2 | Major 11**

Figure 2: Did the authors evaluated the effect of HDAC inhibitors on CMK11-5 cell line as it is shown
for BET inhibitors in Figure 2C?

We performed the very same experiment with HDAC inhibitors as shown for BET inhibitors, and
observed the same trend of HDACi resistance of the CALR MUT cell line. However, HDACi toxicity was
significantly higher to all CMK11-5 lines, independent of the CALR status, likely due to the too high
toxicity of these drugs, even at 1 μ M. Because of this high overall toxicity we chose not to include the
data and do not make any claims in this regard.

**Reviewer 2 | Major 12**

Figure S5B: in main text authors say that MF represents the condition with the highest dissimilarity
with HD, nevertheless, in figure S5B they show a network where the color of each node represents
the fold change between MF and ET. Are these proteins differentially expressed also when considering
the comparison between MF and HDs? Why did they decided to show the comparison between MF
and ET? The maximum FC detected is only 1.6 (Log₂FC= 0.68), considering tSNE representation in
Figure 5B it is not surprising since MF and ET samples are almost superimposable. It is expected that
differences between MF and HDs are more pronounced. Does this difference involve the same
proteins and pathways?

The analysis in **Figure S5A** (former Figure S5B) attempts to visualize the full extent of MPN-deregulated
protein networks. We agree with the author that it requires more extensive explanation and have
adjusted this in the revised manuscript figure legend. Step-wise, we have performed the following:

- 1. Assess the significant proteins between ET and HD and between MF and HD, with significance
defined by an adjusted T-test p-value < 0.01
- 2. Select only proteins that reach a significant upregulation for both the ET vs HD and MF vs HD
comparisons, thus being significantly upregulated in both MPN diseases
- 3. Make a network visualization based on a stringent STRING interaction score cutoff of > 0.7
- 4. Highlight within this network whether the deregulation is similar in MF patients as in ET
patients, by color coding of the nodes
 - a. White nodes represent a log₂FC of MF vs ET of zero, thus being similarly upregulated
in MF and ET patients
 - b. Green nodes represent a positive log₂FC of MF vs ET, thus being more upregulated in
MF than in ET patients
 - c. Red nodes represent a negative log₂FC of MF vs ET, thus despite being commonly
upregulated in both MF and ET when compared to HD, they are less upregulated in
MF than in ET patients

We now clarify this better in the figure legend of **Figure S5A**.

432 **Reviewer 2 | Major 13**

Figure 3G: It is not clear why, for the mutation-specific proteins only MF CALR and MF JAK2 samples
are included, and it is not shown the expression in ET CALR and ET JAK2 samples. In methods sections
authors says that they evaluated differential protein expression using ANOVA analysis. Why are there
shown Student's t-test results?

We thank the reviewer for the opportunity to clarify this point. We have focused on MF-specific
mutational influences to stay within the MF-focused scope of the paper. However, we now include
the results of ET patients in **Figure S5C**. As can be seen below, the mutation-specific protein expression
patterns are consistently present in both ET and MF.

**Figure S5C (only mutation-specific proteins):** Boxplots of expression levels of the 15-protein signature grouped by MPN
general, disease-, and mutation-specific expression patterns. Protein levels are normalized by subtracting the median HD
expression.

We had evaluated differential protein abundances using ANOVA analyses, to separate the effect of for
example general MPN from MF patient status. T-test significance of a protein upregulated in all MPN
patients vs HD, will also lead to a significance finding when comparing MF vs HD (or ET vs HD). To filter

out these effects from general to specific, we performed a Type I ANOVA: first all variance that could
 be contributed to MPN vs HD was tested, then the variance that could within MPN be contributed to
 MF vs ET, and finally the variance that could within MF be contributed to CALR or JAK2 mutation status.
 To not only show the ANOVA significances for the aforementioned specific comparisons, but be able
 to for example for the general MPN vs HD comparison also show the differences in protein levels of
 ET vs HD and MF vs HD, we performed subsequent Student's t-tests and used this to define the
 significance levels visualized in **Figure 6A**.

**Figure 6A:** Volcano plots of comparisons underlying the classification of the RFE-selected protein signature (n=15). Volcanoes
 indicate $\log_2\text{FC}$ (x-axis) and significance (y-axis) of Student's t-test results comparing granulocyte protein levels for the
 indicated groups. The proteins belonging to the 15-protein signature are highlighted across all comparisons. Proteins are
 classified by reaching significant differences only in the HD vs MPN comparison ("MPN general" alterations), also in the MF
 vs ET comparison ("Disease-specific" alterations), or also in the MF CALR vs MF JAK2 comparison ("Mutation-specific"
 alterations).

 **Reviewer 2 | Major 14**

Figure 4: represented factors were selected only based on adjusted p-value or FC was used to select
 significantly deregulated proteins? Would it be possible to represent the results of ANOVA analysis, in
 a supplementary figure showing a volcano plot for each analysis performed, or in a supplementary
 table? It is not clear how ANOVA analysis was used to determine the correlation between protein
 expression and mutation VAF, why a Pearson or Spearman correlation analysis wasn't used instead of
 ANOVA?

We have now included volcano plots for the key comparisons in **Figure 6A**. However, ANOVA is
 sometimes required to account for the presence of extensive 'interactions' between the clinical
 factors across this complex patient cohort, making a simple pairwise comparison or correlation
 analysis sometimes difficult to correctly interpret. For the comparisons of the clinical factors
 underlying the granulocyte proteotype network analysis, a single ANOVA has been performed to
 simultaneously assess the influence of MPN vs HD, MF vs ET, and CALR vs JAK2.

An example of such a complex interaction: MF and ET patients tend to have highly differential VAF,
 which we observe in our cohort, and has been reported before in literature. If we just did a Pearson
 or Spearman correlation with VAF, these ET-MF VAF differences would overrule the analyses. Consider
 the MCM proteins, which are elevated in MF. As MF patients have a higher VAF, there would be a very

significantly positive correlation between MCM and VAF for both CALR and JAK2. However, when
 assessing the influence of VAF within ET patients or within MF patients, there is no correlation in which
 MF patients with higher VAF have higher MCM protein levels than those with low VAF.

The ANOVA results can be compared to a (more laborious) Pearson correlation that accounts for
 differences in VAF between MF and ET by linear regression. We show this through the following steps:
 a) create a linear model of ET/MF vs protein levels, b) take the residuals of this model (i.e. protein
 levels from which the influence of ET/MF has been removed), and c) perform regular linear Pearson
 correlations. As can be seen in the figure below, this yields the same prioritization of proteins as the
 ANOVA does.

Pearson correlations of granulocyte protein levels versus *CALR/JAK2* VAF after regressing out the influence of ET vs MF. For
 every protein, a linear model has been used to model protein levels to MF/ET disease status. The residuals of this model
 have been correlated to the respective VAF across the cohort. The x-axis shows correlation coefficients, and the y-axis their
 significance as $-\log_{10}(\text{p-value})$. Highlighted are the top-20 proteins correlating to *CALR* and *JAK2* VAF, which are the same as
 shown in Figure 4.

Reviewer 2 | Major 15

Line 230: according to authors results, MF diagnosis was associated with the expression of proteins
 involved in DNA replication. This result came from the proteotype analysis of granulocytes which
 represent terminally differentiated post-mitotic cells. Can the authors comment on this? Did the
 authors evaluate granulocyte purification efficiency (e.g. morphologic evaluation or flow cytometry)?

Indeed, as indicated by the reviewer, granulocytes were used for comparisons of the proteotypes of
 MF and ET patients and HD, and highlighted the specificity of proteins involved in DNA replication,
 including the MCMs, in MF patients. We next found that this could largely be attributed to a subgroup
 of MF patients. We also expected in general low replication protein levels in differentiated
 granulocytes and T-cells. However, by comparison of MF HSPC (and T-cell) proteotyping we could see

that the same high protein levels that were readily present in the less differentiated HSPCs seemed to
be maintained upon differentiation to granulocyte/T-cells within a subset of patients (Figure 7F).

To confirm this, we included quantitative immunofluorescence-based imaging data of MF PBMCs co-
stained with MCM7 and CD34 (see example image in Figure S6C). This shows that despite overall levels
of MCM7-high expressing cells indeed being higher in the CD34+ (immature) compared to the CD34-
(mature) compartment of PBMCs as indicated by the percentages on the axes, high MCM7-expressing
cells are not exclusive to CD34+ cells yet can also be found in the CD34- population (see below).
Furthermore, the trends of MCM levels across MF patients as described in Figure 7G and correlated
to defined drug responses hold true in both compartments. Regarding the implication of this we can
currently only speculate, yet it might be that these terminally differentiated post-mitotic cells have
high MCM expression as a “memory” from the progenitor cells they differentiated from. As mentioned
in the response to Reviewer #1, the investigation of DNA replication in MPN is part of a large ongoing
project, which will among others try to address the important question raised by the reviewer.

MCM7 staining quantification as shown in Figure 7G (left panel) split up for the CD34+ (left) and CD34- (right) cell populations.
p-values indicate Student's t-test significance: **** = p<0.0001, *** = p<0.001, * = p<0.05.

Regarding the reviewer's question of granulocyte purification efficiency, this has indeed been
evaluated during isolation for one of our cohorts (University Hospital Basel cohort, data available for
34/36 samples), We can show that the purification method yields an almost pure granulocyte fraction
with a median purity of 92.7% (see below).

Granulocyte isolation purities as determined by flow cytometry. Samples are grouped by disease and mutation status and
 the overall median purity is annotated.

**Reviewer 2 | Major 16**

Would it be possible to validate MCMs expression in granulocytes, T-cells and/or HSPCs by means of
 western blot analysis and at transcript level by means of real time qRT-PCR? Authors should validate
 the expression of deregulated proteins, like 15-protein signature and HSPA5 by means of other
 techniques such as western blot analysis or real time qRT-PCR?

 We thank the reviewers comment, and now validate our findings and simultaneously assess whether
 these are transcriptionally regulated by investigating mRNA levels. Rather than qPCRs for individual
 genes, we chose the strategy to query publicly available datasets, thereby at the same time validating
 our findings in independent external cohorts.

 We sought to identify ideal validations for our key findings, which we classified as follows:

Key finding	Core protein set	Validation dataset	Brief description
Proteotype alterations directly related to the CALR mutation: ER stress proteins	ER stress proteins: CALR, CANX, HSPA5, HSP90B1, HYOU1, CNPY3, PPIB, DNAJC3, DNAJB11, PDIA3, PDIA4, PDIA6,	Nam et al. , Nature 2019 (single-cell HSPCs, n=5 ET CALR, n=4 MF CALR)	A single-cell method was developed to correlate CALR mutation status to mRNA expression within individual ET and MF patients
15-protein signature correlated to disease status	STYX, NEK5, COPS8, MCM4, RFC2, MCM7, FH, CD59, ZNF735	Rampal et al. Blood 2014 (granulocytes, CALR/JAK2 selected; n=1 healthy, n=41 ET, n=13 MF)	Large-scale array-based transcriptomics was performed on granulocytes isolated from MPN patients

			and healthy donors
15-protein signature correlated to MF mutation status	ALPL, ACSL1, PSAT1, CALR, PBEF1, NAMPT, S100A9	1) Rampal et al. Blood 2014 2) Rontautoli et al. Blood Adv 2021 (MF granulocytes, CALR/JAK2 selected; n=21 CALR, n=43 JAK2)	1) See above 2) Large-scale array-based transcriptomics was performed on granulocytes isolated from MF patients
MCM-protein upregulation in a subset of MF patients	MCM2, MCM3, MCM4, MCM5, MCM6, MCM7	Rampal et al. Blood 2014	See above

As for the ER stress signature correlated to the *CALR* mutation, we can in the dataset from Nam *et al.* (Nam *et al.*, 2019), indeed find that this upregulated specifically in the mutated cells of ET and MF patients (see **panels below**). We have found the same by qPCR profiling of our CRISPRed cell line panel (data not shown) and want to highlight that we have for HSPA5 the single-cell antibody-based quantification (i.e. similar as WB) included in the manuscript in **Figure 8F**. This validates our finding of the *CALR* mutation being directly causative of the ER stress upregulation, and shows that this is transcriptionally regulated. Of note, the only protein behaving oppositely as expected from the proteotype data is *CALR* (transcriptionally upregulated, down on protein-level). This coincides with the post-transcriptional degradation and/or secretion of *CALR* protein as discussed in our manuscript and elsewhere.

Nam *et al.* 2019 Genotyping of Transcriptomes
Single-cell mRNA expression *CALR* mutant vs WT cells

Analysis of the publicly available genotyping of transcriptomes data of Nam *et al.* (Nam *et al.*, 2019). Within single-cell HSPC and MkP transcriptomes, mRNA expression of mutated cells of n=5 ET and n=4 MF patients are compared to their corresponding wild type transcriptomes. Indicated are genes belonging to the *CALR* VAF ER stress correlation signature of **Figure 8B**.

As for proteotype alterations related to disease status (i.e. either MPN vs HD or MF vs ET), we see that the MCM signature is regulated on the transcriptional level, with mRNA levels correlating very well with the proteotype data with levels being lowest in HD, intermediate in ET, and highest in MF (Rampal *et al.*, 2014). Interestingly, this could correlate to the increased transcriptional cell cycle module found in MF cells in Nam *et al.* (Nam *et al.*, 2019). The concurrence between proteotype and transcriptional regulation is not present for all disease-related proteins, indicating a large influence of post-transcriptional regulation. Exemplary, CD59, a protein found in our data and various other protein-based datasets to be significantly downregulated, is upregulated in MF on a transcriptional level in the data of Rampal *et al.* This warrants the importance of proteotyping for assessment of levels of the functional (protein) molecule.

Analysis of the publicly available MPN granulocyte transcriptome data of Rampal *et al.* (Rampal *et al.*, 2014). Transcriptomes of MPN patients and healthy donor controls are queried for the disease-related proteins of the 15-protein and MCM signatures. The genes of the 15-protein signature not included here (STYX, NEK5, and ZNF735) were not present in the dataset. Array-based mRNA expression is summarized per gene by selection of the max probe. Log2 values are normalized to the healthy controls per gene. p-values indicate Student's t-test significance: *** = $p < 0.001$, ** = $p < 0.01$, * = $p < 0.05$.

As for the proteotype alterations related to MF mutation status (i.e. MF CALR vs MF JAK2), we combine the finding from Rampal *et al.* (Rampal *et al.*, 2014) with the MF-specific granulocyte transcriptome dataset of Rontauroli *et al.* (Rontauroli *et al.*, 2021) to increase validation power. We see across these datasets that indeed the proteotype trends, in which the six proteins were higher in MF JAK2 than MF CALR, hold true on a transcriptional level, indicating their transcriptional regulation. Once more, CALR is the single exception, which has readily been described above.

Analysis of the publicly available MF granulocyte transcriptome data of Rampal *et al.* (Rampal *et al.*, 2014) and Rontauroli *et al.* (Rontauroli *et al.*, 2021). Transcriptomes of MF patients are queried for the MF mutation-related proteins of the 15-protein signature. Array-based mRNA expression is summarized per gene by selection of the max probe and expressed as log2 values. p-values indicate Student's t-test significance: *** = $p < 0.001$, ** = $p < 0.01$, * = $p < 0.05$.

**Reviewer 2 | Major 17**

Line 288: According to figure 1B MF patients' lymphocytes do not express mutated CALR. Can the authors comment on their observation that lymphocytes from homozygous CALR patients display upregulation of ER stress proteins. How can it be related with homozygous CALR when these cells do not express mutant protein? This relates to major comment #9.

This is an interesting observation for which there are different possible explanations. However, please note that negative scaled intensities shown in **Figure 2D-E** (former Figure 1B-C) do not necessarily mean that there is no expression at all. However, we have seen by manual assessment of images that indeed, particularly for most heterozygous MF CALR patients, we cannot detect CALRm signal above background levels in T-cells. Interestingly, one can see that two of the homozygous patients have a slightly higher CALRm signal, thus not ruling out they do have low but significant mutant CALR expression. Secondly, in **Figure 2E** we show the intensities averaged across the population. Within the population of T-cells, there could be a subpopulation that has higher CALRm levels and leads to an increased averaged ER stress protein abundance in our bulk T-cell proteotyping. Thirdly, another (non-exclusive) hypothesis is that T cells could be ER-stressed due to the presence of secreted mutant CALR (Garbati *et al.*, 2016).

**Reviewer 2 | Major 18**

Line 333: Strikingly, this study lack a functional evaluation of the drugs identified using the high throughput model developed by the authors. Can the authors evaluate the effect on proliferation and survival on sorted cells from MF patients? (e.g. CD34+ cells and/or CD14+ monocytes). I mean, would it be possible to isolate CD34+ cells and Monocytes from MF patients homozygous or heterozygous for CALR, treat them with an UPR inhibitor or ER stressors and evaluate the effect on cells proliferation,

viability, apoptosis? This would be the first step for a subsequent in vivo testing. The same should be
done for Vosaroxin and can be overlooked for molecules already adopted in clinical trials.

We have followed the reviewers suggestion and now provide functional evaluation of drugs identified
by PCY in sorted cell populations from MF patients (**Figures 7K and 8J**; both also reproduced **below**).

In order to validate and get more insight into the effect of vosaroxin on HSPCs, we have FACS-sorted
CD34+ HSPCs from MF PBMC samples. We have selected three patient samples that were found to be
sensitive for HSPC depletion in PCY to assess how vosaroxin exerts its effect. After 20h of 10uM
treatment, the most effective concentration in PCY, we indeed confirm reduced HSPC counts across
all samples. In order to follow up on possible reasons for HSPC depletion, we assessed apoptosis and
proliferation by cleaved caspase-3 and Ki-67 stains, respectively. Here, we found that in all three
samples both an increase in apoptosis and reduction in proliferation could be quantified (**Figure 7K**).

**Figure 7K:** Vosaroxin responses of isolated CD34+ MF patient HSPCs. HSPCs were isolated by FACS from MF patient PBMCs
and exposed to 10µM vosaroxin. Boxplots represent log₂FC differences between DMSO- and drug-treated conditions in
viable cell counts and cleaved caspase-3 and Ki-67 positivity for three vosaroxin-sensitive patients.

To further follow up on the effect of ER stressors on CALR-mutated cells, we furthermore exposed
FACS-sorted CD34+ HSPCs to carfilzomib. Having selected one homozygous and two heterozygous MF
CALR patients, we indeed find the highest reduction of CALR mutant-expressing cells in the
homozygous patient HSPCs, corresponding to our PCY data. As a reduction in CALRm staining could
either indicate a) killing of CALR mutant-expressing cells or b) a reduction of expression in otherwise
viable cells, we followed up by assessing cell viability and apoptosis with a cleaved caspase-3 apoptosis
stain. Here, we found indeed that the higher correlation of CALRm reduction coincided with a lower
number of viable cells and higher induction of apoptosis by carfilzomib treatment. These results point
towards the HSPC depletion discovered by PCY being a result of carfilzomib-mediated killing of CALRm-
positive cells (**Figure 8J**).

**Figure 8J:** Carfilzomib responses of isolated CD34+ MF patient HSPCs. HSPCs were isolated by FACS from MF patient PBMCs
 and exposed to 1 μ M and 10 μ M carfilzomib. Boxplots represent log₂FC differences between DMSO- and drug-treated
 conditions in CALRm single-cell mean intensities, viable cell counts, and cleaved caspase-3 positivity for homozygous and
 heterozygous MF CALR patients.

**Minor**

 **Reviewer 2 | Minor 2**

Supplementary figure 1A and F: for continuous variables like VAF can the color scale be modified to
 make the intermediate color equivalent to the median value.

We tried modifying the color of the scale in **Figure 2B and 5B** (former Figure S1A and S1F) to make the
 intermediate color equivalent to the median value. However, as the median of the cohort is close to
 the lower bound (indicated in the legends) it did not work well to our liking. We recommend keeping
 it as is.

**Reviewer 2 | Minor 2**

Figure 5H: in box plot y-axis change uM with um, moreover the name of each group should be
 corrected by substituting MF with MCM.

We thank the reviewer for these corrections and have adapted **Figure 7H** (former Figure 5H)
 accordingly.

**Reviewer #3 (Remarks to the Author): expertise in**
**proteomics**

Wildschut et al. here present a study that has high clinical value for myelofibrosis patients and contains
a wealth of data. Clinical and functional data has been integrated with proteomic data, with the latter
being a major strength of the study.

The authors have in most cases successfully presented the data clearly (and beautifully), and managed
to highlight the key conclusions.

I have a few comments for improvements. Most of these relate to the data that forms the basis for
the rest of the manuscript. Nevertheless, the manuscript does already contain the validation of the
conclusions, wherefore my comments can mostly be used to strengthen the conclusions further.

**Major**

**Reviewer 3 | Major 1**

It is a major concern that the chosen isolation approaches for proteotyping are not high purity flow
cytometry-based. It is likely a problem with low purity for especially the HSPC-enriched samples,
because of the low frequency of CD34+ cells in the starting material (as the authors also show in Fig
S3C). Therefore, it is desirable that the authors show the purity of each cell type and sample after
isolation. The ranking plots of Suppl Fig 1 indicate there is high variability in purity, but a FACS plot
would be more appropriate. If there is impurity in the isolation and large variability in purity between
samples, the proteotype information does not accurately represent the cell type of interest, and the
authors need to address this issue.

We agree with the reviewer that a FACS-based purification can be more stringent than MACS-based
enrichment, and welcome the opportunity to address this point. Our laboratory has a long-standing
experience with the purification of CD34+ cells from PBMCs by MACS. Below, we now show the purity
data for MACS-based CD34+ isolations from our patient cohort. Across data on the purity of 73
independent isolations, median CD34+ purity is **98%**. To account for the rare outliers in purity, we
performed subsequent stringent bioinformatic approaches for definition of outliers (already
previously detailed in the M&M section).

We now report these statistics in the manuscript under the header “MF PBMC isolation” in the M&M
section: “We have found this protocol to yield a median CD34+ enrichment purity of 98% across 73
independent isolations for the included MF patients”.

CD34+ purities after MACS purification of MF patient samples. Purity data from previous isolations of MF patients included in this study are shown. Enriched CD34+ cells are stained with CD3, CD19, and CD34, and percentage positive cells as determined by flow cytometry are shown on the y-axis.

Reviewer 3 | Major 2

In addition, I do not find it correct to refer to it as "PBMCs were sorted", when in reality it is only an enrichment that has been done.

We agree with the reviewer and adapted the phrasing in the "MF T-cell and HSPC proteotyping" methods section accordingly.

Reviewer 3 | Major 3

Somewhat related, the hemoglobin regression needs to be mentioned and explained in the main text.

We agree with the reviewer and mention and now explain the hemoglobin regression in the main text of the revised manuscript.

Reviewer 3 | Major 4

It is potentially a problem that as many as 20 MF patients have been used in both cohorts. That is almost half of the patients from the first cohort, giving bias to high chances of finding the same pathways. How does the correlation in terms of enriched pathways look if these 20 patients are excluded from the second analysis?

We used the integration of the HSPC/T-cell and the granulocyte dataset not to present a validation of findings from granulocytes in a second cohort, which would, as we agree with the reviewer, given the overlap in patients included not be valid to present as independent data. In contrast, we made use of the overlap in patients as a means to compare two independently generated datasets measuring results from distinct cell types. We utilized it to show that, even in the context of different isolation, processing, and quantification methods, findings from granulocyte proteotypes are partially translatable to more difficult to access cell types, such as HSPCs. Therefore, the pathway enrichment

analysis mentioned by the reviewer was performed on only the 20 overlapping patients: this is the
optimal cohort to assess cross-cell type translatability in.

**Reviewer 3 | Major 5**

Line 154: It would be desirable to show in the circos plot of Fig S1A which 3 samples/patients were
excluded from drug response results.

We agree with the reviewer and have now highlighted the 3 samples/patients in the circos plot of
**Figure 2B** (former Figure S1A) of the revised manuscript as suggested.

**Figure 2B:** Circos plot of clinical annotations of the MF PBMC cohort. Legend indicates color codes for the included variables.
For discrete variables, patient numbers are included, and for continuous variables, the range and median are represented.
Proteotyping outliers are annotated by H and T for HSPC and T-cell proteotypes, respectively.

**Reviewer 3 | Major 6**

Line 205: "Within MF, we found that their cellular proteotypes showed consistent pathway-level
alterations with MF HSPC proteotypes (Figure S5A)."

I think the figure does not entirely justify this statement, and the figure is difficult to understand. In
my view, the MF HSPCs have a few weakly enriched pathways that are overlapping with MF
granulocytes, but it is a bit stretched to call it "consistent". That said, it is not totally unexpected that
there would be many cell type-specific pathway alterations.

The authors can also help the reader with more details and explanations in the figure legend,
especially by clarifying which input samples have been used, e.g. whether it is n=20 or n=40 for the
HSPCs and T cells.

We thank the reviewer for pointing this out, and agree. For simplicity, we have removed the specific
analysis, while still pointing out enrichments in selected key pathways (ER stress & replication)
observed between both HSPCs and granulocytes.

**Reviewer 3 | Major 7**

Are the outliers in Fig S1B, C and G filtered out? This is not clear in the methods, and it should be
visible in Fig S1A and F which samples are outliers (and thus hopefully excluded from proteotype
analysis).

The outliers as defined in **Figure S4A-D** (former Figures S1B, S1C, and S1G) have indeed been filtered
out of the respective proteotype analyses. For the MPN granulocyte cohort, this led merely to removal
of replicates or could be accounted for by repeated MS injections. For the MF PBMC cohort, in which
we had only a single replicate and injection per patient, outlier detection led to removal of specified
patients from the subsequent analyses. We have annotated these patients in the Circos plots of **Figure**
**2B** (former Figure S1A; see also **above**).

**Reviewer 3 | Major 8**

Fig S1D shows that there is one T cell sample (rank 39) that has passed the outlier filtering despite a
very low CD3 expression. Can the authors comment on the validity of the data from this sample?

We decided to rather than using the quantification of a single “marker” protein as an exclusion criteria,
use cutoffs that could be used in a similar fashion for the three different cell types (HSPCs, T-cells, and
granulocytes, of which the latter had not been subjected to marker-based MACS). We decided to
define samples to be removed based on a) outliers, determined by proteotype dissimilarity to all other
samples measured, and b) data quality, with as a proxy the amount of missing proteins (NA values
instead of protein quantities) per sample. As an example, for the T-cell samples, three samples had
very dissimilar overall proteotypes, of which one was furthermore characterized by a low amount of
quantified proteins. These three samples were thus removed.

Indeed, as for HSPCs and T-cells the CD34 and CD3 proteins are used for enrichment, a low abundance
of these proteins in the data could indicate a less pure enrichment. We noticed that most outliers
indeed had low marker proteins, however for the HSPCs in particular, outlier samples were also found
in the middle of CD34 abundance ranks. For the particular T-cell sample mentioned by the reviewer,
we noticed the opposite: despite being across the entire quantified proteome similar to all other
samples, the CD3E protein levels were relatively low. We can only hypothesize that this either has a
technical or biological background, meaning either due to the MS detection or the T-cells of this
patient having reduced their CD3E abundance relating to for example their activation state, despite
similar global proteotypes.

**Reviewer 3 | Major 9**

Suppl Table S3 needs more description in title or legend for each tab - this data is a rich resource for
the community and deserves better and more clear presentation. It is unclear for most tabs what
values are shown. For example, why is there only one value per patient, and not one per drug?

We thank the reviewer for pointing this out and have added description tabs to our supplementary
tables, including **Table S3**, in the revised manuscript for further clarification.

More specifically for **Table S3**, we realize that there was a mistake in the naming, which was “The
processed proteotype drug response matrices” instead of the correct name “The processed
proteotype matrices”. Thus, there is only one data point per patient in this table because in contrast
to **Table S2**, which shows the pharmacoscopy data of patient cells treated with many different drug
conditions, in the proteotyping summarized in **Table S3** we have only measured protein levels under
basal conditions. The data points thus reflect abundances of a single protein for a single patient. The
range of values is different from usual on the first tab (“Granulocyte”), as these represent residuals
from the hemoglobin regression instead of log2 intensity values commonly represented for
proteotyping data (which are added in the tab “Granulocyte_noResiduals”). We hope our added
descriptions enable the community to better interpret the values represented in our data tables.

**Minor points:**

**Reviewer 3 | Minor 1**

The colour coding for mutation in the legend of Fig S1F seem to be incorrect (HD cohort has been
coloured as JAK2 mutation).

We thank the reviewer for pointing this out and adapted the color coding in **Figure 5B** (former Figure
S1F).

**Reviewer 3 | Minor 2**

Figure S2C: To say that the CALRm specifically results in sensitivity to JAK/STAT signaling inhibition,
more than two replicates would make the statement better justified. Can the authors elaborate on
why the KO responds like the MUT at 1 uM and like the WT at 10 uM? Alternatively, the figure can be
shown as a response curve instead of bar graphs to show the differences between the three groups
better (and by that justify the statement and significance)?

The significance statement of **Figure S1D** (former Figure S2C) is based on comparing the 4 CALR MUT
cell line drug-treated conditions (all replicates and concentrations) to the 25 DMSO-treated
conditions. For the CALR MUT line, this identifies a significant reduction in viability for both tested
JAK/STAT inhibitors. These are the significance levels shown in the drug headers of **** $p < 0.0001$
and *** $p < 0.001$ for Fedratinib and Ruxolitinib, respectively. In contrast, both the WT and KO do not
show a significant reduction in viability upon either JAK/STAT inhibitor. The fluctuations of the KO the
reviewer points out are centered around 0 and with a significantly reduced effect size compared to
that of the MUT line, and not significant.

**Reviewer 3 | Minor 3**

Figure S3F: Could the authors show this data as a scatter plot instead, to show the real correlation for
CNN-based data and the counts for each patient?

We have included the data visualized as a scatterplot below. We believe that directly correlating the
two values against each other can be confusing, as the measures are not quantifying exactly the same:
blood counts in PB are different from cell population fractions in isolated PBMCs. Therefore we prefer
to show the general correlating trend, in which those samples characterized by high PB counts also
have high PBMC fractions, less directly using the boxplots in **Figure S2F** (former Figure S3F).

Correlation plot of clinical peripheral blood (PB) monocyte counts versus CNN-based monocyte cell populations sizes. The x-
 axis indicates the population size as determined by the 4-class cell type image-based CNN in the DMSO-treated condition.
 The y-axis indicates PB monocyte counts in whole blood as determined by ADVIA. Colors represent MF CALR and MF JAK2
 patients. Spearman correlation coefficient and significance are annotated.

**Reviewer 3 | Minor 4**

Could the authors use the same colour coding for the different drug classes in Fig S4D and Fig 1E?

We thank the reviewer for highlighting this inconsistency and have used the same color coding in our
 revised manuscript for **Figure S3D and 3A** (former Figure S4D and 1E).

**Reviewer 3 | Minor 5**

Abbreviations could be avoided in abstract.

We adapted the abstract accordingly and reduced the use of abbreviations.

**Reviewer 3 | Minor 6**

With what approach was missing value imputation performed?

We would like to refer the reviewer to the end of Materials & Methods section “Protein-level
 identification and quantification”. After removing all proteins from the data matrix that were not
 detected in 90% of all patients of at least one group, for the remaining proteins missing values were
 imputed with the lowest detected value for that protein and up to five percent added noise.

**Reviewer #4 (Remarks to the Author): expertise in**
**computational analysis of drug screening**

The study by Snijder and coworkers is a well done and highly relevant report that reveals new methods
and information of relevance to both basic and translational research.

The broader study of ex vivo analysis of cell types from myelofibrosis patients incorporates next-
generation methods in precision medicine. It highlights several new concepts in disease classification
based on the authors proteotype analysis. Finally, it associates several of the newly revealed disease
proteotypes with potential therapies based on an ex vivo drug response signature.

**Major**

**Reviewer 4 | Major 1**

The studies primary issue is presentation. The manuscript could easily be doubled in size to delve more
deeply into the methods and data. The figures are rich with information that is often described in a
brief sentence or two. The reader is left to sift through the methods and SI figures to better place key
outcomes in the context that the authors intend. If word-count limits can be extended for this
manuscript the authors should consider a more descriptive version of this paper be re-submitted.

Minimally, the main texts description (or figure or figure legend) of the drug exposure should include
the drug concentration(s) and time of exposure.

We have indeed extended the revised manuscript, primarily adding information to the main figures
and methods sections.

We had already in the main text and figure legends mentioned the time point of drug exposure.
However, concentrations were not explicitly mentioned in the main text (only in the methods section),
leading us to include the information in the main text of the revised manuscript as follows:

“In short, after 20 hours of drug treatment, cells were fixed and stained with a multiplexed
antibody panel and imaged by high-content automated microscopy. Small compound drugs
were tested at 1 and 10 μ M, while biologics were tested at 0.1, 1, and 10 μ g/ml.”

**Methods section:**

“All drugs were randomized across the plate layout with duplicates of 1 μ M and 10 μ M final
concentrations, except for antibodies that had three replicates and 0.1 μ g/ml, 1 μ g/ml, and
10 μ g/ml final concentrations. Cells were incubated with drugs for 20h at 37°C and 5% CO₂.”

**Reviewer 4 | Major 2**

The method is highly innovative and the authors deserve significant credit for working out key details
that enable such a robust ex vivo scoring model. The use of ‘sanity check’ secondary methods like the

CALRm and pSTAT5 imaging is also well done. The method still relies primarily on cell intrinsic factors
to reveal drug sensitivities. For instance, the revelation that the 'MCM-high' proteotype is associated
with cell cycle/DNA repair pathways and respond to inhibitor of those pathways/targets is
straightforward and a highly actionable finding. The data around the JAK2 proteotype is possibly less
straightforward. While activating mutations in JAK signaling is a hallmark of myelofibrosis, a purely cell
intrinsic view of JAK signaling is probably insufficient to explain the findings in this study. For instance,
only the more promiscuous JAK inhibitor pacritinib (misspelled in figure S2F) demonstrate a strongly
response in the MF JAK2 proteotype while highly selective (and clinically valuable) ones like ruxolitinib
are less active. Pacritinib is also a potent IRAK inhibitor. Could that be playing a role (particularly in the
pSTAT5 intensity reduction)? The authors should review their consider and discuss.

We thank the reviewer for pointing this out, and have now expanded on the discussion on the
differences between the multiple tested JAKi and cite relevant prior studies. As can be noted from the
drug panel (full panel in **Figure S3A and Table S2**), we have purposely selected a large variety of drugs
from this class within our drug library (n=8/79 drugs) to be screened across the full patient cohort.
Despite being regarded as drugs with a similar method of action, their chemical properties, clinical
responses, and the clinical experiences with these drugs are highly variable as well (see e.g. (Bose and
Verstovsek, 2020)): For example, Ruxolitinib-resistant patients can often still be successfully treated
with fedratinib or pacritinib. Therefore, we are of the opinion that, as the reviewer mentions, other
drug targets that include IRAK, as well as differences in their affinity for JAK1 and JAK2, can all
contribute to these apparent response differences.

**Reviewer 4 | Major 3**

The data and discussion around BET inhibitor and HDAC inhibitor signatures is interesting and
potentially important. These mechanisms obviously play a role in multiple cancers and dozens of HDAC
and BET inhibitors have entered human clinical evaluation. A reasonable therapeutic index for HDAC
inhibitors has been found for multiple agents while this remains elusive for BET inhibitors which likely
reflects the complex mechanistic consequences of these drugs. The data and discussion around the
enrichments in this study are limited to a previously defined inverse correlation with RAS pathway
activation. Overall, this wasn't the most impactful outcome from this study. Its completely up to the
authors, but limiting the discussion around this finding to give room for expanding the method
description and other findings would be fine.

We thank the reviewer for the suggestions. We no longer mention the RAS pathway in the discussion,
while expanding on additional (new) results and on methodological details in both the main text and
methods section. Given the current interest for BET inhibition specifically for the treatment of MF, as
identified from ongoing clinical trials, its inclusion may be of potential interest and relevance for the
MF community, for which we chose to keep it.

**Reviewer 4 | Major 4**

The authors should discuss the lack of activity for key standards of care of myelofibrosis which are
unlikely to demonstrate activity in a 24-hour ex vivo (in vitro) experiment. These include the 'imid'
drugs like pomalidomide and lenolidomide and hedgehod and Smoothened modulators like glasdegib,
vismodegib, and erismodegib. All precision medicine platforms will have to learn how to handle
important drugs that rely on cell extrinsic factors which are often missing from ex vivo/in vitro

environment and the authors should consider and discuss their methods and its limitations in
predicting proteotypes which would benefit from these agents.

We thank the reviewer for pointing this out. Indeed, we tried to include as many potentially clinically
relevant drugs in our MF drug library, including the standards of care, while simultaneously realizing
that our drug platform will not be able to pick up all the clinically relevant effects that these drugs
could exert. However, we would like to note that IMiDs and Smoothed modulators are not standard
of care for MF. For immunomodulatory drugs, rather than the readouts used within the scope of this
study (i.e. cell killing and mutant readout reduction) that as expected show little effect in the time
window of 20h, our newly developed platform could be used to investigate cell-cell interactions,
potentially reflecting their clinical effects better. As pharmacoscopy includes the actual immune
components of the individual patient's blood inside the drug-treated wells, it has readily been proven
as a powerful method to elucidate these cell-extrinsic effects (Severin et al., 2022; Shilts et al., 2022;
Vladimer et al., 2017), and ex vivo sensitivity to immunotherapies of multiple myeloma patients has
been found to correctly identify patients with prolonged clinical responses to the matched therapies
(Kropivsek et al., 2023). Follow-up analyses focusing on this in the context of MPNs/MF could elucidate
the therapeutic window for drugs such as the included imiDs, and could, in combination with the here
developed and investigated readouts, lead to an improved understanding of the proteotype
signatures underlying response and resistance to immunomodulatory agents as well as
immunotherapies in MF. We now mention this opportunity in the discussion.

**Reviewer 4 | Major 5**

I was also intrigued that exportin inhibitors were not discussed. Selinexor (and Selinexor+JAK
inhibitor) therapy is demonstrating promise in myelofibrosis. Did the Selinexor responses not show
any enrichment?

We thank the reviewer for the suggestion. In this study, we have limited ourselves to single agent drug
exposure and still see interesting single-agent effects of Selinexor. As can be seen in **Figure 3A and**
**S3D**, we find a particular HSPC-targeting effect of Selinexor in MF CALR patients.

**Reviewer 4 | Major 6**

Finally, the outcomes with the BCL-2 inhibitors (Navitoclax and Venetoclax) are of crucial importance.
A deeper dive into the secondary signatures for responding versus non-responding would greatly
increase the impact of this work.

We agree with the importance of BCL-2i and upon suggestion of the reviewer repeated the analyses
presented in this paper for BETi and HDACi for this drug class.

Firstly, we correlated HSPC BCL-2i drug responses to clinical factors, similar as in performed in **Figure**
**4B** for the BETi and HDACi. For BCL-2i, did not find significant independent associations to the selected
factors.

ANOVA of BCL-2 inhibitor drug responses to clinical parameters. This analysis is the same as performed in **Figure 4B** for BET and HDAC inhibitors. Heatmap colors indicate signed significance of positive (red) and negative (blue) associations. Only positive associations that do not reach significance are found for Bcl-2 inhibitors.

Next, we correlated the drug responses to HSPC proteotypes. Interestingly, we find strong signatures of resistance and response in this analysis. Most strikingly, HSPC with high levels of proteins involved in oxidative phosphorylation and various terms associated with this, are resistant to BCL-2 inhibition. This

corresponds to the findings obtained in CLL samples and cell lines of Guièze *et al.*, in which resistance
 to BCL-2 inhibition was functionally shown to be driven by increased oxidative phosphorylation
 (Guièze et al., 2019). MF HSPCs with higher JAK-STAT signaling pathway protein levels are more
 sensitive to Bcl-2 inhibition, potentially reflecting the targeted effect of this drug class.

Figure 4F: Analysis of protein processes significantly up- or downregulated in BCL-2 inhibitor-responding patients. This analysis is the same as performed in **Figure 4D** for BET and HDAC inhibitors. T-tests were performed for each protein comparing protein levels in HSPCs of responding MF patients compared to the non-responding and resistant patients. GSEA was performed on proteins ranked according to signed t-test significance. GSEA normalized enrichment scores (NES; x-axis) and enrichment significance (y-axis) are shown.

We have now added this analysis to the manuscript as **Figure 4F** and thank the reviewer for the suggestion.

Overall, this is an excellent study and report with dual relevance to the myelofibrosis community and
 the precision medicine community.

**Minor requests/comments**

**Reviewer 4 | Minor 1**

The authors should explain what they mean by BET and BCL-2 family inhibitors having a ‘common
mechanism’ (page 5, line 167).

We realize that the sentence was ambiguous and meant to indicate that within the same drug family
there is a common mode of action, rather than comparing the BET to the BCL-2 family inhibitors. We
thank the reviewer for highlighting this ambiguity and have rephrased the sentence.

**Reviewer 4 | Minor 2**

Figure 3 and the main text description of the data presented in this figure are sparse enough that it
limits a broader understanding of the deeper dive into specific proteins/pathways revealed in this
study. If the authors can expand on this it would be good.

To address this point we have reorganized selected results and expanded on key observations.
Specifically, we have separated the data condensed in the former Figure 3 to one figure dedicated on
proteotype-based patient classification (**Figure 5**) and a second on the biology underlying these
proteotype differences (**Figure 6**). We further simplified the visualization of the associations of these
15 proteins in the series of volcano plots in **Figure 6A**. The MCM complex members and ER stress
signature identified by the approach are of course subject to additional analysis, particularly in **Figure**
**7 and Figure 8**. In all, we hope this expanded discussion and visualization of these results are
satisfactory.

**Craig Thomas**

Reviewer #5 (Remarks to the Author): expertise in single cell image and deep learning analysis

The authors presented drug screening and proteotype profiling to shed light on vulnerabilities of myelofibrosis (MF), a hematopoietic stem cell disorder belonging to myeloproliferative neoplasms or mutated blood progenitor cells. The technique was applied to patient blood using high-content imaging, mass spec-based protein profiling, and deep learning based single cell analysis. Biological findings included (1) CALR mutations linked to BET and HDAC inhibitor sensitivity, (2) MCM complex high proliferation linked to pro-survival signaling and DNA replication, and (3) homozygous CALR mutations linked to high ER stress. While the paper demonstrates interesting vulnerabilities in drug sensitivities in MF, there are multiple major concerns:

Reviewer 5 | Major 1

The novelty seems to be very limited. Pharmacoscopy (cited in 16, Lancet), multiplexed single cell imaging and CNN classification (cited in 21, now published in Sci Adv), and conventional DIA LC-MS/MS proteomics. The abstract and manuscript are a bit misleading: “We developed a pharmacoscopy drug screening approach”. These claims should be toned down to reflect accurate prior art in this line of work. Brief and relevant details of each approach should be detailed for readers to follow the methodologies.

We thank the reviewer for the suggestion. However, the quote “We developed a pharmacoscopy drug screening approach” appears nowhere in our manuscript. The text that comes closest to the quote reads as follows (from the introduction):

“Utilizing pharmacoscopy, previously shown to be clinically predictive across a variety of hematological malignancies^{16,17}, we developed a single-cell multiplexed and deep learning-based MF drug response profiling platform that captures patient heterogeneity and quantifies drug efficacy on the malignant MF clone.”

And in the results section:

“Building on recent work²¹, we established a multiplexed single-cell pharmacoscopy drug screening strategy that allows us to measure by deep learning the drug response of CD34⁺ HSPCs, CD3⁺ T-cells, CD14⁺ monocytes, and other (marker negative) cells in primary patient blood samples. We complement this cell identity-based analysis with single-cell level readouts reflecting the presence of either CALR or JAK2 driver mutations.”

Both sentences fully acknowledge the prior existence with pharmacoscopy in different contexts, while highlighting the specific novelty and advance of the work presented in this study here. In this manuscript, we develop a pharmacoscopy drug screening approach specifically tailored to myelofibrosis, and for the first time, explore integrative pan-cohort ex vivo drug screening results with sample-matched DIA-SWATH proteomics on MPNs and MF. While pharmacoscopy and proteomics indeed have been

applied previously in other studies and looking at different diseases, we see ample novelty of our
study, including:

- 1. First-time use of a mutant-specific antibody to systematically detect expression of the driver
mutation at the single-cell level in MF patient samples. Previous pharmacoscopy-based
studies relied on surface marker expression for detection of diseased cell populations.
However, these can have overlapping expression with healthy cells. We here use a readout
that directly recognizes the oncogenic driver of disease. This is solely present in diseased cells,
thus depletion of the CALRm-positive cells is the most direct readout of drug specificity
possible for MF CALR patients.
- 2. First-time use of a mutant-induced pathway readout. While realizing that in contrast to the
CALRm pSTAT5 can also be detected in healthy cells, measuring pathway activation of the
JAK/STAT pathway enables prioritization of drugs that reduce this known driver of disease.
- 3. Proteotyping of sorted cell population combined with sample-matched drug response
profiling enables elucidation of novel insights of disease. We know that particularly for CALR
mutations, that heavily disturb the protein homeostasis, a protein-level interrogation of
disease pathways is critical for elucidation of molecular driving events that can be absent or
even oppositely regulated on a transcriptomic level.

1135 **Reviewer 5 | Major 2**

While image-based single cell analysis of “blood cells” has been utilized by the authors in recent papers
(since 2017), side-by-side comparisons to IMAGE STREAM and multiplexed FLOW CYTOMETRY of the
same or even more number of antibodies can be screened and used for drug response screening.
Image-based multiplexing has been powerful for the spatial context of tissues or subcellular
organization. However, imaging blood cells (no spatial context) is less interesting unless authors
perform a spatial feature analysis (nucleus, cytosol, organelles, and other pixel-level features). Even
Figure 6F indicates single cell intensities in x-y axes (that the same can be done with flow cytometry
or image stream cytometry). I highly recommend subcellular spatial analysis for revisiting drug
responses.

We thank the reviewer for these critical comments. However, we would like to clarify that the reviewer
is incorrect when stating that subcellular and spatial information do not factor into our results.

In fact, our method is indeed taking subcellular and spatial pixel information into account, namely
through the use of so called convolutional neural networks (CNNs), which are a deep learning method
to analyze and classify objects in image data that explicitly make use of spatial pixel pattern
information (https://en.wikipedia.org/wiki/Convolutional_neural_network). We use these CNNs to
classify cells as being either alive or dead, as well as to assign each cell into distinct cell classes (HSPCs,
monocytes, T-cells, and other cells).

In order to show the benefit we experience in our use of CNNs, we have repeated our 4-class cell type
classification by training a fully-connected neural network (NN) classifier that lacks the convolutional
layers characteristic of CNNs, and thus do not make use of spatial pixel information. Instead of training
the CNN on the raw images, we trained the alternative NN on averaged single-cell features extracted
by conventional image analysis (CellProfiler). As can be seen below (and now added to the
**Supplementary Figure S2A-B**), the subcellular information significantly affects CNN performance:

whereas our image-trained CNN has an accuracy of 90.7%, this drops to 75.2% upon loss of subcellular
 and spatial pixel-level information. In other words, cell morphology greatly contributes to the
 classification accuracy of our image-based drug screening approach.

**Figure S2A and S2B:** Confusion matrix of the image-based (left) and feature-based (right) CNN cell type classifiers. Networks
 were trained on the same single-cell crop input images. Diagonal numbers indicate prediction accuracies per cell type of the
 disjoint test set not used for training of the CNN. Network accuracies across cell types average to an overall 90.7% and 75.2%
 accuracy across all included MF PBMC samples for image- and feature-based classification, respectively.

 Image Stream and flow cytometry indeed might be alternative methods to pharmacoscopy when
 applied to drug discovery in patient samples, however, we are not aware of current studies that have
 applied these methods for drug discovery in MF or MPN, emphasizing the novelty of our study. Direct
 comparisons of these alternative methods for drug response measurements is outside of the scope of
 our study.

 Lastly, despite being outside of the scope of this study, we have shown that pharmacoscopy provides
 opportunities to study spatial interactions among blood and bone marrow cells. See for example
 (Kropivsek et al., 2023; Severin et al., 2022; Shilts et al., 2022; Vladimer et al., 2017). Building on the
 first pharmacoscopy approach and study we present here for MF, it could indeed be interesting to
 explore multicellular spatial phenotypes in response to for example immunotherapies, especially if
 and once those become clinically available for MPNs and MF - which is currently not the case. We now
 highlight the opportunity in the discussion of the manuscript.

 **Reviewer 5 | Major 3**

Image-based CNN quantification shows 90.7% accuracy but is this clinically reproducible in all the MF
 samples and benchmarked by a simple approach for cell phenotyping? I highly recommend a
 systematic approach to delineate the (1) technical errors from (2) biological variability in drug testing
 and quantification.

 We have analyzed the CNN accuracy per sample, and find that, remarkably, the mean accuracy in
 detecting CD34+ cells (the key readout of our drug screens resulting from this CNN) across all patients

is **96%** (see **panel below**). Thus our technical error is well below the biological variability. This is also
 evident from the discovery of key independently validatable molecular and clinical associations with
 drug sensitivity (see **Figures 7 and 8**) that we identify across the cohort.

 CNN accuracies per cell type for each of the included MF PBMC patient samples. Percentages indicate prediction accuracies
 1199 per cell type of the disjoint test set not used for training of the CNN.

 **Reviewer 5 | Major 4**

When integrating image-based drug targets with MS/MS based proteotype drug targets, what are the
 normalization steps involved to take into account dynamic range, protein specificity, and sensitivities?
 This is a concern beyond using z-scores for analysis.

 The reviewer seems to allude to the possibility that our data integration between drug responses and
 proteotype measurements might be sensitive to differences in dynamic range of the different
 datasets. In order to show robustness of our results to potential influences of for example dynamic
 range and normalizations used, we have repeated our analyses using a purely rank-based approach.
 For every protein of the replicative signature (n=50) and for every screened drug (n=79) we performed
 a rank- rather than linear-based Spearman correlation analysis of protein levels to HSPC drug
 responses. As shown below, the conclusions corresponding to the correlations of **Figure 7I** still hold,
 and are thus robust to differences in dynamic range and chosen normalization methods. We hope this
 alleviates the reviewers' concern.

Spearman correlations of HSPC drug sensitivities to the 50-protein signature. The same data has been used as for **Figure 7I**.
 Yet instead of a Pearson correlation with averaged z-scored protein values, which could theoretically be influenced by factors
 such as dynamic range of protein quantification, for every protein (n=50) and every drug (n=79) a Spearman rank-based
 correlation was performed. Correlation coefficients and signed $-\log_{10}(\text{correlation p-values})$ were averaged per drug. The x-
 axis shows the averaged Spearman correlation coefficients and the y-axis the corresponding FDR-adjusted $-\log_{10}(\text{p-values})$.

**Reviewer 5 | Major 5**

After establishing the image-based assay with the abovementioned rigor and proper data integration
 with MS/MS data, the biological insights may need to be revisited as they were provided in (1) CALR
 mutations linked to BET and HDAC inhibitor sensitivity, (2) MCM complex high proliferation linked to
 pro-survival signaling and DNA replication, and (3) homozygous CALR mutations linked to high ER
 stress.

We thank the reviewer for sharing their concern. As we now show above in our reply to this Reviewer's
 Major point 4, our integration results are robust and reproduced when using rank-based approaches
 that are independent of the scale of the individual measurements.

Overall, the proposed approach, after maturation and extensive validations, may be applicable to
 other disorders, including blood and solid tumors.

We thank the reviewer for these encouraging words. We have now performed two *interventional*
 *clinical trials* (EXALT-1 and DARTT-1) where lymphoma and AML patients received treatments based
 on our pharmacoscopy platform (Heinemann et al., 2022; Kornauth et al., 2022; Schmid et al., 2023;
 Snijder et al., 2017). In both of these interventional clinical trials, we observed improvements in clinical
 response rates, indicating that drug testing in primary patient samples at these concentrations,
 timepoints, and with this platform, is predictive of clinical response and helps guide the personalized
 treatment of individual patients. Furthermore, we have just published the results of a multi-year
 observational clinical trial on multiple myeloma patients in Nature Cancer (Kropivsek et al., 2023).
 Here, again, we analyzed *ex vivo* drug responses by pharmacoscopy at the same drug concentrations,
 timepoints, and relative readout concept. Critically, we again find that also for this cohort, *ex vivo* drug

responses at these conditions are indicative of clinical responses. We are currently developing the
platform further to the study of immunology (Severin et al., 2022; Shilts et al., 2022; Vladimer et al.,
2017) as well as the *ex vivo* drug responses of solid tumors, including melanoma, ovarian cancer, and
brain cancers (Conde et al., 2021; Eichhoff et al., 2023; Irmisch et al., 2021; Lee et al., 2022; Meister
et al., 2022). Building on the first-time development and extensive use of pharmacoscopy for the study
of MPNs that we present in this work, we now intend to take these developments as a basis to explore
the use of pharmacoscopy to guide clinical decision making for the management of MPN.

References

Bose P and Verstovsek S (2020) JAK Inhibition for the Treatment of Myelofibrosis: Limitations and
Future Perspectives. *HemaSphere* 4(4): e424.

Conde J, Pumroy RA, Baker C, et al. (2021) Allosteric Antagonist Modulation of TRPV2 by
Piperlongumine Impairs Glioblastoma Progression. *ACS central science* 7(5): 868–881.

Eichhoff OM, Stoffel CI, Käsler J, et al. (2023) ROS Induction Targets Persister Cancer Cells with Low
Metabolic Activity in NRAS-Mutated Melanoma. *Cancer research*: OF1–OF19.

El-Khoury M, Cabagnols X, Mosca M, et al. (2020) Different impact of calreticulin mutations on
human hematopoiesis in myeloproliferative neoplasms. *Oncogene* 39(31): 5323–5337.

Garbati MR, Welgan CA, Landefeld SH, et al. (2016) Mutant calreticulin-expressing cells induce
monocyte hyperreactivity through a paracrine mechanism. *American journal of hematology*
91(2): 211–219.

Guièze R, Liu VM, Rosebrock D, et al. (2019) Mitochondrial Reprogramming Underlies Resistance to
BCL-2 Inhibition in Lymphoid Malignancies. *Cancer cell* 36(4): 369–384.e13.

Heinemann T, Kornauth C, Severin Y, et al. (2022) Deep morphology learning enhances ex vivo drug
profiling-based precision medicine. *Blood cancer discovery*. DOI: 10.1158/2643-3230.BCD-21-
0219.

Irmisch A, Bonilla X, Chevrier S, et al. (2021) The Tumor Profiler Study: integrated, multi-omic,
functional tumor profiling for clinical decision support. *Cancer cell* 39(3): 288–293.

Jacobs K, Doerdelmann C, Krietsch J, et al. (2022) Stress-triggered hematopoietic stem cell
proliferation relies on PrimPol-mediated repriming. *Molecular cell* 82(21): 4176–4188.e8.

Kornauth C, Pemovska T, Vladimer GI, et al. (2022) Functional Precision Medicine Provides Clinical
Benefit in Advanced Aggressive Hematological Cancers and Identifies Exceptional Responders.
*Cancer discovery* 12(2): 372–387.

Kropivsek K, Kachel P, Goetze S, et al. (2023) Ex vivo drug response heterogeneity reveals
personalized therapeutic strategies for patients with multiple myeloma. *Nature cancer* 4(5):
734–753.

Larsen TS, Christensen JH, Hasselbalch HC, et al. (2007) The JAK2 V617F mutation involves B- and T-
lymphocyte lineages in a subgroup of patients with Philadelphia-chromosome negative chronic
myeloproliferative disorders. *British journal of haematology* 136(5): 745–751.

Lee S, Weiss T, Bühler M, et al. (2022) Targeting tumour-intrinsic neural vulnerabilities of
glioblastoma. *bioRxiv*. DOI: 10.1101/2022.10.07.511321.

Meister H, Look T, Roth P, et al. (2022) Multifunctional mRNA-Based CAR T Cells Display Promising
Antitumor Activity Against Glioblastoma. *Clinical cancer research: an official journal of the*
*American Association for Cancer Research* 28(21): 4747–4756.

Mughal FP, Bergmann AC, Huynh HUB, et al. (2022) Production and Characterization of Peptide
Antibodies to the C-Terminal of Frameshifted Calreticulin Associated with Myeloproliferative
Diseases. *International journal of molecular sciences* 23(12). DOI: 10.3390/ijms23126803.

Nam AS, Kim K-T, Chaligne R, et al. (2019) Somatic mutations and cell identity linked by Genotyping
of Transcriptomes. *Nature* 571(7765): 355–360.

Rampal R, Al-Shahrour F, Abdel-Wahab O, et al. (2014) Integrated genomic analysis illustrates the
central role of JAK-STAT pathway activation in myeloproliferative neoplasm pathogenesis. *Blood*
123(22): e123–33.

Rontautoli S, Castellano S, Guglielmelli P, et al. (2021) Gene expression profile correlates with
molecular and clinical features in patients with myelofibrosis. *Blood advances* 5(5): 1452–1462.

Schmid JA, Festl Y, Severin Y, et al. (2023) Efficacy and feasibility of Pharmacoscopy-guided
treatment for acute myeloid leukemia patients that exhausted all registered therapeutic
options. *medRxiv*. DOI: 10.1101/2023.03.28.23287745.

Severin Y, Hale BD, Mena J, et al. (2022) Multiplexed high-throughput immune cell imaging reveals
molecular health-associated phenotypes. *Science advances* 8(44): eabn5631.

Shilts J, Severin Y, Galaway F, et al. (2022) A physical wiring diagram for the human immune system.
*Nature*. DOI: 10.1038/s41586-022-05028-x.

Snijder B, Sacher R, Rämö P, et al. (2009) Population context determines cell-to-cell variability in
endocytosis and virus infection. *Nature* 461(7263): 520–523.

Snijder B, Vladimer GI, Krall N, et al. (2017) Image-based ex-vivo drug screening for patients with
aggressive haematological malignancies: interim results from a single-arm, open-label, pilot
study. *The Lancet. Haematology* 4(12): e595–e606.

Theocharides APA, Lundberg P, Lakkaraju AKK, et al. (2016) Homozygous calreticulin mutations in
patients with myelofibrosis lead to acquired myeloperoxidase deficiency. *Blood* 127(25): 3253–
3259.

Vladimer GI, Snijder B, Krall N, et al. (2017) Global survey of the immunomodulatory potential of
common drugs. *Nature chemical biology* 13(6): 681–690.

REVIEWERS' COMMENTS

Reviewer #1 (Remarks to the Author):

The authors addressd all points raised, I have no futher comment.

Reviewer #2 (Remarks to the Author):

The authors answered all the major comments thoroughly and included some of the obtained results in the final version of the manuscript.

I have only some minor concerns.

1: in Figure 1B the legend says that image magnification is 25x, while in answer to major3 you said magnification is 100x. Which is the correct one? Please correct the paper if necessary

2: Figure 2E and F. I understand authors' answer to comment Major7. If the authors want to emphasize intrapatient heterogeneity I would suggest to change box plot visualization in Figure 2E by grouping them according to driver mutation and compare CALRm or pSTAT5 intensities among cell types in order to support their assumption "levels are highest in monocytes and HSPCs".

Moreover, in Figure 2F they used Student's t-test to compare CALRm intensities, nevertheless, since heterozygous CALR group is composed by only 2 samples I believe a non parametric test should be more indicated.

3: Authors answered Major11 and said that they evaluated the effect of HDACi in CMK11-5 cells. Surprisingly they observed HDACi resistance in CALR MUT cell line, maybe there was a mistake and they would have said "HDACi sensitivity". They decided not to present these data because of high overall toxicity, I would suggest to include this observation in the manuscript at least in a supplementary figure because otherwise it is lacking of an important functional validation.

4: As regards Major comment 16, authors validated proteomic results at the transcriptional level in independent cohorts retrieved by literature. I strongly suggest the inclusion of these validations in the final version of the manuscript since they demonstrate that results obtained by the authors in the present work relates to fundamental molecular mechanisms involved in disease pathogenesis.

5: line 275 substitute brown with grey

Reviewer #3 (Remarks to the Author):

I thank the authors for responding to my questions and concerns, and for improving the manuscript by elaborating on some of the things that had been unclear to me. I have two remaining concerns:

Originally Major 1

I am still concerned about the purity of some of the samples that have been included and hope the authors can make this more transparent. Median purity of 98% is great, however that is still only the median and doesn't tell about variability in purity and whether some of the samples have a very low purity. Can the authors include the data on purity of each sample and cell type? If the median purity of 98% has been calculated from 73 isolations, why don't they show the purity for all 73?

Originally Major 9

In Table S3, tabs "Granulocyte" and "Granulocyte_noResidual", the gene names that have been exchanged with 'Mar-02' and 'Mar-03' in Excel need to be fixed.

Reviewer #4 (Remarks to the Author):

The authors clearly worked hard to address the concerns and comments from all 5 reviewers. The paper is improved and of high value to the community.

Reviewer #5 (Remarks to the Author):

The authors provided satisfactory responses to the concerns raised before:

- 1 . Subcellular inclusion in CNN models and comparisons of image features.
2. Showing technical reproducibility.
3. Benefits over ImageStream in terms of application.
4. Rank approach when comparing image-data and MS/MS results.

While the authors' assertive language is not suitable, it is a fine revision and the answers are acceptable.

RESPONSE TO REVIEWERS' COMMENTS

Reviewer #1 (Remarks to the Author):

The authors addressd all points raised, I have no futher comment.

Reviewer #2 (Remarks to the Author):

The authors answered all the major comments thoroughly and included some of the obtained results in the final version of the manuscript.

I have only some minor concerns.

Reviewer 2 | Comment 1

In Figure 1B the legend says that image magnification is 25x, while in answer to major3 you said magnification is 100x. Which is the correct one? Please correct the paper if necessary

We thank the reviewer for pointing out this ambiguity and apologise for the mistake. Throughout, we now indicate image magnification with a capital X, while indicating the number of images (or wells) without an x, thus:

“[...] at 20x magnification with 25x imaged sites per well” now becomes

“[...] at 20X magnification with 25 imaged sites per well”.

The images in question are taken at 20X magnification.

Reviewer 2 | Comment 2

Figure 2E and F. I understand authors' answer to comment Major7. If the authors want to emphasize inpatient heterogeneity I would suggest to change box plot visualization in Figure 2E by grouping them according to driver mutation and compare CALRm or pSTAT5 intensities among cell types in order to support their assumption “levels are highest in monocytes and HSPCs”. Moreover, in Figure 2F they used Student's t-test to compare CALRm intensities, nevertheless, since heterozygous CALR group is composed by only 2 samples I believe a non parametric test should be more indicated.

By grouping Figure 2E first by cell type, and then by mutation status, we highlight the fact that the observed inpatient heterogeneity is in part cell-type dependent and consistent between mutations. We have tried the alternative proposed grouping, i.e. first by mutation and then by cell type, but the plot gets very confusing as cell type is the more dominant factor.

Regarding the reviewer's concern about the Student's t-test with limited patient numbers in the homozygous (not heterozygous) MF CALR group: we have made it clearer visible that we have here 3 and not, as it looks from the figure, 2 patients included, and all n's are now better reported in the figure legends

throughout. Furthermore, the data underlying this and all other figures can now be accessed in our Source Data.

Reviewer 2 | Comment 3

Authors answered Major11 and said that they evaluated the effect of HDACi in CMK11-5 cells. Surprisingly they observed HDACi resistance in CALR MUT cell line, maybe there was a mistake and they would have said “HDACi sensitivity”. They decided not to present these data because of high overall toxicity, I would suggest to include this observation in the manuscript at least in a supplementary figure because otherwise it is lacking of an important functional validation.

Indeed, in our original response to Major 11 we erroneously wrote “HDACi resistance” instead of “HDACi sensitivity”. We apologize for the mistake and thank the reviewer for spotting it.

Below we show the data for the CMK11-5 cell line panel treated with HDACi. In line with our MF patient cohort data, the CALR MUT cell line is the most sensitive to the tested HDAC inhibitors in 5 of the 6 conditions measured. However, the HDAC inhibitors are overall more cytotoxic than the BETi, and the resulting trends are therefor less clear as for the BETi data. For this reason, we have decided not to include these results and not to make any claims about them in the manuscript.

Reviewer 2 | Comment 4

As regards Major comment 16, authors validated proteomic results at the transcriptional level in independent cohorts retrieved by literature. I strongly suggest the inclusion of these validations in the final version of the manuscript since they demonstrate that results obtained by the authors in the present work relates to fundamental molecular mechanisms involved in disease pathogenesis.

We have now added these supporting transcriptional analyses from publicly available data to Supplementary Figures S6A (focus on MCM complex) and S7D (ER stress signature).

Supplementary Figures S6A:

Supplementary Figures S7D:

Reviewer 2 | Comment 5

line 275 substitute brown with grey

We thank the reviewer for the critical look and have corrected this in our manuscript.

Reviewer #3 (Remarks to the Author):

I thank the authors for responding to my questions and concerns, and for improving the manuscript by elaborating on some of the things that had been unclear to me. I have two remaining concerns:

Reviewer 3 | Originally Major 1

I am still concerned about the purity of some of the samples that have been included and hope the authors can make this more transparent. Median purity of 98% is great, however that is still only the

median and doesn't tell about variability in purity and whether some of the samples have a very low purity. Can the authors include the data on purity of each sample and cell type? If the median purity of 98% has been calculated from 73 isolations, why don't they show the purity for all 73?

There appears a slight miscommunication: From 11/40 patients of our cohort we have historical FACS-based purity data of CD34+ MACS-enriched biopsies. As we repeatedly collect patient biopsies, we have for those 11 patients in total 73 MACS-based enrichments for which we assessed the purity by FACS. The median CD34+ purity across all these MACS-enrichments is very good, i.e. 98% pure. The per-sample data is now shown in the below graph (as individual dots). However, we do not have this data for all samples analyzed by proteotyping in our cohort, which is why we cannot include the FACS-based purity of all samples analyzed in our cohort.

Reviewer 3 | Originally Major 9

In Table S3, tabs "Granulocyte" and "Granulocyte_noResidual", the gene names that have been exchanged with 'Mar-02' and 'Mar-03' in Excel need to be fixed.

We thank the reviewer for carefully checking our tables and have corrected the gene names in our Excel tables.

Reviewer #4 (Remarks to the Author):

The authors clearly worked hard to address the concerns and comments from all 5 reviewers. The paper is improved and of high value to the community.

Reviewer #5 (Remarks to the Author):

The authors provided satisfactory responses to the concerns raised before:

- 1 . Subcellular inclusion in CNN models and comparisons of image features.
2. Showing technical reproducibility.
3. Benefits over ImageStream in terms of application.
4. Rank approach when comparing image-data and MS/MS results.

While the authors' assertive language is not suitable, it is a fine revision and the answers are acceptable.